# Higher-Order Certified Robustness for Regression

**Jie Zhang** [1]   **Natalie Frank** [1]

## Abstract

Randomized smoothing has emerged as a scalable technique for certifying the adversarial robustness of classifiers. However, its application to regression remains under-explored and faces unique challenges. Existing regression certificates rely on probabilistic acceptance regions and fail to exploit the local geometry of the function. In this work, we present a novel framework for certified robust regression that addresses these limitations. We derive a prediction-centered certificate that guarantees the stability of the smoothed model's prediction and ensures practical computability at test time. We investigate several alternatives for constructing these certificates by explicitly incorporating means, variances, and gradients. In particular we demonstrate on the MNIST rotation task that utilizing gradient information yields significantly tighter robustness certificates compared to the current state-of-the-art, $\alpha$-smoothing.

## 1. Introduction

Deep neural networks have demonstrated remarkable success across a wide range of applications, yet their vulnerability to adversarial perturbations remains a significant obstacle to their deployment in safety-critical systems. These perturbations, often imperceptible to humans, can cause dramatic failures in model predictions—a phenomenon extensively documented in classification tasks (Szegedy et al., 2013; Biggio et al., 2013; Goodfellow et al., 2014). Consequently, the field has pivoted toward defenses with formal guarantees, with randomized smoothing emerging as a leading scalable method for certifying robustness without making strong assumptions about the model architecture (Cohen et al., 2019). At a high level, randomized smoothing constructs a smoothed predictor by averaging the base model's predictions under random perturbations of the input, and

---

[1]University of Washington, USA. Correspondence to: Jie Zhang <claizhan@cs.washington.edu>, Natalie Frank <natalief@uw.edu>.

*Proceedings of the 43rd International Conference on Machine Learning*, Seoul, South Korea. PMLR 306, 2026. Copyright 2026 by the author(s).

then uses the induced smoothness to certify that predictions remain stable within an input neighborhood. However, while certification for classification is well-established, the robustness of regression models—which predict continuous, real-valued outputs—remains less explored. This gap is critical; in domains like autonomous navigation and medical imaging, certifying that a continuous prediction will not deviate beyond a safe margin is a fundamental safety requirement.

Prominent recent approaches for regression, most notably $\alpha$-smoothing and RS-Reg (Rekavandi et al., 2024; 2025), operate by defining a probabilistic "safe" region and estimating the probability mass $p_A$ that the base model's outputs fall within this region under noise. This formulation is advantageous for its flexibility, as it allows for certification over arbitrary output distributions without assuming parametric forms. However, these approaches effectively reduces the complex behavior of the function to a simple binary outcome (inside or outside the region), discarding critical information regarding the local geometry of the smoothed landscape. Consequently, it often yields loose certificates that do not fully exploit the smoothness properties induced by the noise.

Furthermore, the certification criteria in $\alpha$-smoothing and RS-Reg are typically defined by ensuring that for any perturbed input, the smoothed model's output remains within a neighborhood of the *base* model's output (Rekavandi et al., 2024; 2025). This formulation creates a misalignment between certification and deployment, as the actual model used for inference is the smoothed regressor, not the base model. While it is empirically possible to configure such methods to center on the smoothed prediction, doing so invalidates their theoretical guarantees.

In this work, we introduce a framework that addresses these limitations by leveraging *higher-order information* to derive certificates explicitly centered on the smoothed prediction. By incorporating the gradient norm of the smoothed regressor alongside the variance of the base output evaluated with smoothing noise, we characterize the "worst-case" base function analytically. This approach yields closed-form analytical bounds on the adversarial shift. Unlike methods that rely on certifying probabilistic acceptance regions, our theoretical bounds are deterministic given the population statis-

tics, though practical certification utilizes high-probability estimates of these values. This distinction allows for significantly tighter radii by exploiting the function's geometry, while simultaneously resolving the anchor inconsistency by guaranteeing the stability of the actual deployed prediction $g(x)$. We empirically validate our framework on the MNIST rotation angle prediction task, demonstrating that incorporating gradient information yields significantly tighter robustness radii compared to baselines relying on variance or quantile-based inclusion bounds alone.

## 2. Related Work

**Randomized smoothing and higher-order certification.** Randomized smoothing has established itself as a scalable standard for certifying $\ell_2$ robustness in deep classifiers (Lecuyer et al., 2019; Cohen et al., 2019), often relying on the Neyman-Pearson lemma to derive optimal certification regions from zeroth-order statistics. While subsequent work investigated incorporating first-order information (Mohapatra et al., 2020; Levine et al., 2020), adapting these insights to regression requires a fundamental shift in methodology. Unlike the classification setting, which focuses on decision boundary geometry or likelihood ratios, regression necessitates bounding the magnitude of continuous function variation. To achieve this, we depart from the Neyman-Pearson framework and instead employ a variational analysis to characterize the worst-case base function. This methodological shift is crucial: it allows us to make gradient information practically useful by explicitly incorporating the gradient as a constraint on the worst-case output shift.

**Approaches to robust regression.** Prior to randomized smoothing, certification for regression was primarily explored through Lipschitz continuity analysis (Tanielian & Biau, 2021). However, these techniques typically require white-box access to model parameters and scale poorly to deep architectures. In the black-box setting, recent works have attempted to bridge the gap by reducing regression to classification. For instance, Chiang et al. (2020) discretize the output space for object detection, while Hammoudeh & Lowd (2023) reduce regression poisoning to a classification task.

Most relevant to our work, Rekavandi et al. (2025) and Rekavandi et al. (2024) developed the first direct smoothing frameworks for continuous regression. However, these methods rely on *statistical* inclusion probabilities rather than function geometry. First, as evidenced by the proof of Theorem 1 in Rekavandi et al. (2025), they treat certification as a binary problem: determining whether an output falls within a probabilistic acceptance region based on zeroth-order counts. Second, regarding $\alpha$-smoothing (Rekavandi et al., 2024), while they successfully utilize $\alpha$-trimming to

stabilize predictions—quantifying this benefit via the regularized beta function—their certificate characterizes local stability solely through the distribution of perturbed evaluations. It exploits the trimming to increase the probability of inclusion, but it does not utilize the gradient to analytically bound the worst-case magnitude of the output shift. In contrast, our approach is *geometric*: we utilize the gradient norm to directly constrain the variation of the smoothed function itself.

## 3. Preliminaries

**Randomized Smoothing** Randomized smoothing is a post-processing method for improving the robustness of a given predictor. The method is motivated by the observation that large local variation can make a predictor vulnerable to small input perturbations: nearby inputs may receive substantially different predictions. Randomized smoothing reduces this local variation by replacing the predictor at each input with an average of its values over a distribution concentrated near that input. This averaging yields a smoother predictor whose change over a prescribed neighborhood can be precisely quantified, leading to certified robustness guarantees. Prior work has developed this framework primarily for classification; in this paper, we formulate a corresponding smoothing procedure for regression.

**Notation and Setup** We consider a base regression model (e.g., a neural network) $f : \mathbb{R}^d \to \mathbb{R}$ that maps an input $\boldsymbol{x} \in \mathbb{R}^d$ to a 1-dimensional output. Our analysis focuses on the certification of a smoothed regressor $g : \mathbb{R}^d \to \mathbb{R}$, which is defined as the expectation of the base regressor $f$ under isotropic Gaussian noise:

$$g(\boldsymbol{x}) = \mathbb{E}[f(\boldsymbol{x} + \boldsymbol{e})], \quad \text{where } \boldsymbol{e} \sim \mathcal{N}(\boldsymbol{0}, \sigma^2 \boldsymbol{I}) \quad (1)$$

We denote the $\ell_2$ norm by $\| \cdot \|_2$. We use $\mathbb{E}[\cdot]$ and $\text{Var}(\cdot)$ to denote the expectation and variance operators respectively. We use $y \in \mathbb{R}$ to represent the (potentially unknown) ground-truth value associated with an input $\boldsymbol{x}$.

**Threat Model and Certification Goal** We operate under the standard $\ell_2$-norm threat model. Consider a fixed nominal input $\boldsymbol{z}$. An adversary may perturb $\boldsymbol{z}$ by adding any perturbation $\boldsymbol{\delta} \in \mathbb{R}^d$ as long as the perturbation is bounded, $\|\boldsymbol{\delta}\|_2 \leq R$. The resulting adversarial example is $\boldsymbol{z} + \boldsymbol{\delta}$.

Our objective is to certify the local stability of the smoothed regressor $g$ around $\boldsymbol{z}$. We aim to find the largest possible radius $R > 0$ such that for any perturbation $\boldsymbol{\delta}$ satisfying $\|\boldsymbol{\delta}\|_2 \leq R$, the output of the smoothed function is guaranteed to remain within an acceptable distance, $\epsilon$, of its prediction at clean input:

$$|g(\boldsymbol{z} + \boldsymbol{\delta}) - g(\boldsymbol{z})| \leq \epsilon \quad (2)$$

This formulation provides a practical guarantee, as the certificate is centered on the model's own prediction $g(z)$, which is computable at test time.

Formally, certifying this radius requires bounding the *worst-case shift* of the smoothed function output under perturbation. Denote the worst-case shift as $\Delta_{\max}$. When underlying base function is assumed to come from a function class $\mathcal{F}$, $\Delta_{\max}$ can be seen as the maximizing objective value achieved by this nested maximization problem:

$$\Delta_{\max}(R) = \max_{\|\boldsymbol{\delta}\|_2 \leq R} \max_{f \in \mathcal{F}} |\mathbb{E}[f(z + \boldsymbol{\delta} + e)] - \mathbb{E}[f(z + e)]|$$
(3)

To constrain $\mathcal{F}$, we compute fundamental summary statistics describing $f$, such as its mean and variance under gaussian noise. We then optimize over all base functions consistent with our observed statistics (e.g., variance $C$ and gradient $\mathbf{G}$). We certify a radius by finding the largest $R$ for which $\Delta_{\max}(R) < \epsilon$.

## 4. Certification via Worst-Case Analysis

To derive the certified radius, we solve the nested maximization problem defined in Equation (3) to bound worst case shift $\Delta_{\max}(R)$. To facilitate the analysis using variational calculus, we switch from the noise variable $e$ to the input variable $x$. Let $x = z + e$. Then, under the nominal setting, $x \sim \mathcal{N}(z, \sigma^2 I)$. Similarly, for the perturbed setting, $x \sim \mathcal{N}(z + \boldsymbol{\delta}, \sigma^2 I)$. We define the Gaussian densities centered at the nominal and perturbed points as $p_z = \mathcal{N}(z, \sigma^2 I)$ and $p_{z+\delta} = \mathcal{N}(z + \boldsymbol{\delta}, \sigma^2 I)$ respectively. With this notation, we can express the expectations in problem (3) as integrals over the input space $x$:

$$\Delta_{\max}(R) = \max_{\|\boldsymbol{\delta}\|_2 \leq R} \max_{f \in \mathcal{F}} |\mathbb{E}_{p_{z+\delta}}[f(x)] - \mathbb{E}_{p_z}[f(x)]| \quad (4)$$

To obtain a certificate that holds for *any* base regressor consistent with our observations, we maximize this objective over the space of functions $f$ satisfying specific empirical constraints.

### 4.1. Constraint Specifications

To make the optimization well-posed, we impose constraints based on properties of the base model $f$ that can be empirically estimated.

We distinguish between two regimes regarding the mean of the function. In the *unbounded* setting (where the output $f(x)$ can take any real value), the objective function (4) is invariant to constant shifts: replacing $f(x)$ with $f(x) + c$ changes both expectations by $c$, leaving the difference $\Delta(\boldsymbol{\delta})$ unchanged (see Lemma D.1 in Appendix D). Consequently, we do not require knowledge of the specific mean value

$g(z)$ to certify the radius in unbounded cases. However, in the *bounded* setting, the specific value of the mean determines the relative position of the feasible bounds, breaking this invariance. We define optimization problems for both settings below.

**Unbounded Setting: Variance Constraint.** Since the mean is uninformative in the unbounded setting due to shift invariance, we rely on the variance of the base function to constrain its scale. We assume the variance under the nominal distribution is bounded by an empirically measurable constant $C$. This yields the following optimization problem:

$$\begin{aligned} \underset{\|\boldsymbol{\delta}\| \leq R, f}{\text{maximize}} \quad & \left|\mathbb{E}_{p_{z+\delta}}[f(x)] - \mathbb{E}_{p_z}[f(x)]\right| \quad (5) \\ \text{subject to} \quad & \text{Var}_{p_z}(f(x)) \leq C \end{aligned}$$

**Unbounded Setting: Variance and Gradient Constraints.** To capture local sensitivity, we additionally constrain the gradient of the smoothed function. We assume we have an estimate of the gradient vector $\mathbf{G}$, which fixes both the magnitude and direction of the local trend. We refer to this setting as $(C, G)$ in our experiments.

$$\begin{aligned} \underset{\|\boldsymbol{\delta}\| \leq R, f}{\text{maximize}} \quad & \left|\mathbb{E}_{p_{z+\delta}}[f(x)] - \mathbb{E}_{p_z}[f(x)]\right| \quad (6) \\ \text{subject to} \quad & \text{Var}_{p_z}(f(x)) \leq C \\ & \nabla\mathbb{E}_{p_z}[f(x)] = \mathbf{G} \end{aligned}$$

Our results later reduce the certified robustness bounds to depend only on $\|\mathbf{G}\|$. This reduction is crucial, as empirically estimating a full high-dimensional gradient would require prohibitively many samples.

**Bounded Setting: The Role of the Mean.** In many regression tasks, the output is naturally bounded (e.g., steering angles). We impose a pointwise bound constraint $f(x) \in [-M, M]$. As noted above, the specific value of the mean $E = \mathbb{E}_{p_z}[f(x)]$ determines where the feasible "box" $[-M, M]$ is located relative to the distribution's center (see Appendix D). Therefore, we must explicitly constrain the mean $E$.

We consider two variations in this setting. As a baseline, we define the $(E, C) + M$ setting (detailed in Appendix G), which constrains the mean, variance, and bounds. However, our primary focus is the tightest setting which utilizes all available statistics, denoted as $(E, C, G) + M$:

**Mean, Variance, Gradient, and Boundedness.** We formulate the full optimization problem over functions $f(x)$ as

follows:

$$
\begin{aligned}
\underset{\|\boldsymbol{\delta}\| \leq R, f}{\text{maximize}} \quad & \left| \mathbb{E}_{p_{\boldsymbol{z}+\boldsymbol{\delta}}}[f(\boldsymbol{x})] - \mathbb{E}_{p_{\boldsymbol{z}}}[f(\boldsymbol{x})] \right| \\
\text{subject to} \quad & \mathbb{E}_{p_{\boldsymbol{z}}}[f(\boldsymbol{x})] = E \\
& \text{Var}_{p_{\boldsymbol{z}}}(f(\boldsymbol{x})) \leq C \\
& \nabla \mathbb{E}_{p_{\boldsymbol{z}}}[f(\boldsymbol{x})] = \mathbf{G} \\
& - M \leq f(\boldsymbol{x}) \leq M
\end{aligned}
\tag{7}
$$

## 4.2. Methodology: Calculus of Variations

The optimization defined above requires finding a function $f^*$ from an infinite-dimensional space. This is the domain of the *Calculus of Variations*, which generalizes standard optimization to find functions that extremize a functional (an integral depending on the function).

This connects directly to the certification techniques used in classification. The seminal work by Cohen et al. (2019) relies on the Neyman-Pearson Lemma to find the worst-case base classifier. The Neyman-Pearson Lemma is fundamentally a variational result: it solves for the optimal *indicator* function (a 0/1 test) that maximizes power subject to a constraint on size (Casella & Berger, 2024).

In regression, we generalize this principle. We are not restricted to boolean 0/1 functions; our base function $f$ is real-valued. We employ the method of Lagrange multipliers within the Calculus of Variations framework (see Appendix C for the detailed variational setup). We construct a Lagrangian functional $\mathcal{L}[f]$ that incorporates our observed constraints and solve the Euler-Lagrange equations $\frac{\delta \mathcal{L}}{\delta f} = 0$ to derive the analytical form of the worst-case function $f^*$.

## 5. Worst Case Function

### 5.1. Warm-up: Worst Case Function with Variance Constraint

We begin by deriving the certified radius under the simplest assumption: that the base function $f$ has bounded variance under the smoothing distribution. This serves as a baseline for our method and illustrates the variational approach in an isotropic setting.

**Proposition 5.1.** *[Certified Radius with Variance Estimate] For fixed nominal point $\boldsymbol{z}$, solution to the optimization problem with variance constraint (5) satisfies*

$$
f^*(\boldsymbol{x}) - g(\boldsymbol{z}) = k \left( \frac{p_{\boldsymbol{z}+\boldsymbol{\delta}}(\boldsymbol{x})}{p_{\boldsymbol{z}}(\boldsymbol{x})} - 1 \right)
$$

*where $k = \sqrt{\frac{C}{\text{Var}_{p_{\boldsymbol{z}}}(p_{\boldsymbol{z}+\boldsymbol{\delta}}/p_{\boldsymbol{z}})}}$, and $g(\boldsymbol{z}) = \mathbb{E}_{p_{\boldsymbol{z}}} f(\boldsymbol{x})$. In this case, certified radius to ensure $|g(\boldsymbol{z} + \boldsymbol{\delta}) - g(\boldsymbol{z})| < \epsilon$ can be calculated as*

$$
R = \sigma \sqrt{\log \left( 1 + \frac{\epsilon^2}{C} \right)}.
$$

*Proof Sketch.* (See Appendix E for full derivation). We construct the Lagrangian functional $\mathcal{L}(f, \lambda)$ combining the objective and the variance constraint $\mathcal{L}(f, \lambda) = \int f(\boldsymbol{x})(p_{\boldsymbol{z}+\boldsymbol{\delta}}(\boldsymbol{x}) - p_{\boldsymbol{z}}(\boldsymbol{x}))d\boldsymbol{x} - \lambda \left( \int (f(\boldsymbol{x}) - g(\boldsymbol{z}))^2 p_{\boldsymbol{z}}(\boldsymbol{x})d\boldsymbol{x} - C \right)$. The stationary condition simplifies directly to the worst-case function form $f^*$ presented in the proposition. Plugging this optimal form back into the objective function and solving for the radius $R$ yields the closed-form formula. □

We note that the form of the worst case function identified in Proposition 5.1 is the direct analogue of the Neyman-Pearson lemma: the optimal function is determined by the likelihood ratio of the two distributions. And the closed form formula for $R$ shows how $R$ varies with $\epsilon, C$ and $\sigma$.

While Proposition 5.1 provides a valid certificate, it generates an isotropic bound, relying solely on the scalar variance $C$. Since variance is a rotationally invariant statistic, this certificate cannot distinguish between directions of low and high sensitivity. In the next section, we tighten this bound by incorporating the gradient.

### 5.2. Incorporating Gradient Estimate

We remain in the *unbounded* setting. We now tighten the variance-only bound by incorporating the gradient $\mathbf{G} = \nabla g(\boldsymbol{z})$, solving the problem defined in (6).

To derive the certified radius, we decouple the nested maximization. We first characterize the worst-case base function $f$ for a *fixed* perturbation $\boldsymbol{\delta}$.

**Proposition 5.2.** *[Worst-Case Function for Fixed Perturbation] Consider the inner maximization of Problem (6) with a fixed perturbation vector $\boldsymbol{\delta}$. The base function $f^*$ that maximizes the shift $\mathbb{E}_{p_{\boldsymbol{z}+\boldsymbol{\delta}}}[f(\boldsymbol{x})] - \mathbb{E}_{p_{\boldsymbol{z}}}[f(\boldsymbol{x})]$ subject to the variance constraint $\text{Var}_{p_{\boldsymbol{z}}}(f(\boldsymbol{x})) \leq C$ and gradient constraint $\nabla \mathbb{E}_{p_{\boldsymbol{z}}}[f(\boldsymbol{x})] = \mathbf{G}$ is a function of the form:*

$$
\begin{aligned}
f^*(\boldsymbol{x}) - g(\boldsymbol{z}) = k(\boldsymbol{\delta}) \Bigg[ & \left( e^{\frac{\boldsymbol{\delta}^\top (\boldsymbol{x}-\boldsymbol{z})}{\sigma^2} - \frac{\|\boldsymbol{\delta}\|_2^2}{2\sigma^2}} - 1 \right) \\
& - \left( \boldsymbol{\delta} - \frac{\sigma^2}{k(\boldsymbol{\delta})} \mathbf{G} \right)^\top \frac{\boldsymbol{x} - \boldsymbol{z}}{\sigma^2} \Bigg]
\end{aligned}
\tag{8}
$$

*where $k(\boldsymbol{\delta}) = \sqrt{\frac{C - \sigma^2 \|\mathbf{G}\|_2^2}{e^{\|\boldsymbol{\delta}\|_2^2/\sigma^2} - 1 - \frac{\|\boldsymbol{\delta}\|_2^2}{\sigma^2}}}$ and $g(\boldsymbol{z}) = \mathbb{E}_{p_{\boldsymbol{z}}} f(\boldsymbol{x})$.*

---

**Algorithm 1** Calculating Certified Radius

---

1: **Input:** noise level $\sigma$, failure probability $\alpha$, output shift tolerance $\epsilon$, search limit $r_{\max}$.
2: **Output:** Certified Radius $R$.
3: Estimate variance bound $C$ and gradient norm $\|\mathbf{G}\|_2$, each with error probability $1 - \alpha/2$.
  ▷ Define worst-case shift function (Corollary 5.4)
4: **function** $\Delta_{\max}(r)$
5:   $V_r \leftarrow e^{r^2/\sigma^2} - 1 - r^2/\sigma^2$
6:   **return** $\sqrt{C - \sigma^2\|\mathbf{G}\|_2^2} \cdot \sqrt{V_r} + r \cdot \|\mathbf{G}\|_2$
7: Perform bisection search on $r \in [0, r_{\max}]$ to find maximum $R$ such that $\Delta_{\max}(R) \leq \epsilon$.
8: **Return** $R$

---

The constant $k(\boldsymbol{\delta})$ is guaranteed to be real-valued by the variance lower bound $C \geq \sigma^2\|\mathbf{G}\|_2^2$ established in Proposition B.3 in Appendix B.

Having characterized the worst-case function $f^*$ for any fixed $\boldsymbol{\delta}$, we now solve the outer optimization problem over the perturbation. The following proposition establishes that the worst-case perturbation is necessarily aligned with the gradient direction $\mathbf{G}$, reducing the multi-dimensional search to a single dimension.

**Proposition 5.3.** *Let $\Delta_{\max}(\boldsymbol{\delta})$ denote the optimal value achieved of objective function in (6). For any radius budget $R > 0$, the maximum of $\Delta_{\max}(\boldsymbol{\delta})$ over all perturbations with norm $R$ is achieved when $\boldsymbol{\delta}$ is aligned with the gradient $\mathbf{G}$. That is:*

$$\underset{\boldsymbol{\delta}:\|\boldsymbol{\delta}\|_2=R}{\arg\max} \Delta_{\max}(\boldsymbol{\delta}) = R\frac{\mathbf{G}}{\|\mathbf{G}\|_2}$$

Combining these results yields the closed-form worst-case shift.

**Corollary 5.4.** *For the worst-case perturbation $\boldsymbol{\delta} = R\frac{\mathbf{G}}{\|\mathbf{G}\|_2}$, the worst case shift is:*

$$\Delta_{\max}(R) = \sqrt{C - \sigma^2\|\mathbf{G}\|_2^2} \cdot \sqrt{e^{R^2/\sigma^2} - 1 - R^2/\sigma^2} + R\|\mathbf{G}\|_2$$

As with the previous proposition, the variance lower bound $C \geq \sigma^2\|\mathbf{G}\|_2^2$ (Proposition B.3) ensures this value is real. Notably, this bound depends only on the norm of the gradient $\|\mathbf{G}\|_2$, not its direction.

We observe that $\Delta_{\max}(R)$ is strictly monotonically increasing with respect to $R$ for $R \geq 0$. The first term is monotonic because the function $e^x - 1 - x$ is strictly increasing for $x > 0$ and the second term $R\|G\|_2$ is linear in $R$. This monotonicity guarantees that the equation $\Delta_{\max}(R) = \epsilon$

has a unique solution. We calculate the certified radius for this setting using the bisection search procedure, detailed in Algorithm 1. Notably, the bound explicitly depends on the norm $\|\mathbf{G}\|_2$, confirming that incorporating gradient information tightens the certificate.

The optimal function $f^*$ reveals a fundamental structure: it is a superposition of exponential and linear ramps determined by the projection of the input onto the gradient direction. This generalizes the discrete "slab" constructions of Levine et al. (2020) to continuous regression.

### 5.3. Bounded Function Value Setting

We now address the bounded setting, where $f(\boldsymbol{x}) \in [-M, M]$. As formulated in Section 4.1, we consider two cases: utilizing the Mean and Variance as constraints $((E, C) + M)$, and full set of constraints: Mean, Variance, and Gradient $((E, C, \mathbf{G}) + M)$.

**Case 1: Mean and Variance Constraints** $((E, C) + M)$**.** As derived in Appendix G, we simplify the search for the worst-case $d$-dimensional function $f(\boldsymbol{x})$ by showing it is equivalent to finding a worst-case *univariate* function $\phi(t)$. Here, the scalar variable $t \sim \mathcal{N}(0, \sigma^2)$ represents the projection of the Gaussian noise along the perturbation direction. Consequently, the optimal univariate function $\phi^*(t)$ takes the form of a clipped affine transformation of the likelihood ratio. While structurally related to Proposition 5.1, the bounded solution is distinct in two ways: the pointwise bounds $[-M, M]$ induce clipping in the optimal function form, and the explicit mean constraint introduces an additional Lagrange multiplier. We numerically solve for the mean and variance multipliers to satisfy the integral constraints, while the clipping explicitly enforces the pointwise bounds (Algorithm 2 in Appendix G.3).

**Case 2: Mean, Variance, and Gradient Constraints** $((E, C, \mathbf{G}) + M)$**.** We now consider the fully constrained setting, denoted as $(E, C, \mathbf{G}) + M$, which yields the tightest certification bounds by utilizing all available statistics. Directly solving the variational problem (7) over the space of multivariate functions $f : \mathbb{R}^d \to \mathbb{R}$ is computationally intractable. A primary theoretical contribution of this work is proving that this high-dimensional optimization admits an *exact* reduction to a tractable optimization over univariate functions.

**Theorem 5.5** (Alignment and Reduction (Informal))**.** *Consider the bounded optimization problem (7). This high-dimensional optimization admits two key structural simplifications (proven in Appendix H, Theorem H.1 and Proposition H.2):*

*1. **Directional Alignment:** There exists a worst-case perturbation $\boldsymbol{\delta}^*$ that is perfectly aligned with the gradient*

*direction* $\mathbf{G}$ *(i.e.,* $\boldsymbol{\delta}^* = R \cdot \mathbf{G}/\|\mathbf{G}\|_2$*).*

2. ***Dimensionality Reduction:*** *The worst-case function* $f^*(\boldsymbol{x})$ *varies only along the direction of the gradient. Consequently, the optimization over* $f(\boldsymbol{x})$ *is equivalent to optimizing a univariate function* $\phi(t)$*, where* $t = \frac{\mathbf{G}^\top(\boldsymbol{x}-\boldsymbol{z})}{\|\mathbf{G}\|_2}$ *represents the scalar projection of the Gaussian noise onto the gradient direction.*

This reduction renders the variational problem tractable: instead of optimizing over multivariate functions, we optimize over a univariate function $\phi(t)$. The problem thus depends only on the scalar shift magnitude $\|\boldsymbol{\delta}\|_2$, with the pointwise bounds inducing a clipped solution structure.

**Corollary 5.6.** *[Optimal Function Form] For a fixed shift magnitude* $\alpha$ *(where* $\alpha = R$ *in the worst case), the optimal univariate function* $\phi^*(t)$ *is a clipped affine transformation of the likelihood ratio:*

$$\phi^*(t) = \mathrm{clip}_{[-M-E, M-E]} \left( \frac{w(t) - \mu t - \nu}{2\lambda} \right) \quad (9)$$

*where* $w(t) = \exp(\frac{\alpha t}{\sigma^2} - \frac{\alpha^2}{2\sigma^2}) - 1$ *denotes the centered likelihood ratio, and* $\lambda, \mu, \nu$ *are the Lagrange multipliers for the variance, gradient, and mean constraints, respectively.*

**Algorithmic Solution.** While clipping precludes an analytical solution for the multipliers, the dual problem remains convex. We efficiently solve for $(\lambda, \mu, \nu)$ using a dual optimization procedure (Algorithm 4 in Appendix H) to evaluate the exact worst-case shift $\Delta(R)$, enabling the computation of the certified radius via bisection search (Algorithm 3).

## 6. Estimation of Variance and Gradient Norm

To compute the certified radius, we estimate the variance $C$ and squared gradient norm $\|\nabla g(z)\|_2^2$ using U-statistics on $n$ independent Gaussian samples (Hoeffding, 1992; van der Vaart, 2000). U-statistics provide minimum-variance unbiased estimators for these quantities.

To ensure the joint validity of our certificate with probability at least $1 - \alpha$, we apply a union bound, allocating failure probability $\alpha/2$ to the variance estimation and $\alpha/2$ to the gradient norm estimation. We construct valid two-sided confidence intervals for both parameters at significance level $\alpha/2$ based on their asymptotic normality. For certification, we utilize the conservative upper bound of the variance interval and optimize the worst-case gradient norm over its respective valid interval. Detailed definitions of the kernels and the derivation of these confidence intervals are provided in Appendix I.

A key distinction between our framework and prior work in classification certification (e.g., Cohen et al. (2019)) is

the nature of the statistical guarantees. While classification often relies on exact Clopper-Pearson bounds for Binomial counts, our regression setting requires estimating continuous moments. Deriving tight, exact finite-sample bounds for these quantities without making strong distributional assumptions is often intractable. Consequently, we rely on the asymptotic normality of U-statistics. This represents a limitation of our current work, as strict finite-sample validity is not guaranteed. However, our convergence analysis (Appendix J.2) indicates that for practical sample sizes ($n \geq 10,000$), the asymptotic approximation is highly accurate and the certificates remain empirically sound.

## 7. Experiments

### 7.1. Validating Certificate Tightness on Synthetic Data

We evaluate the tightness and soundness of our unbounded $(C, G)$ certifier on synthetic functions where the true worst-case radius is computable, comparing performance against $\alpha$-smoothing. We test on three synthetic functions: a smooth quadratic bowl, a one-sided slice that is flat on one side and increases linearly beyond a threshold, and a "sandwich" function with lower and upper plateaus separated by a narrow transition band. For each function and each output tolerance $\epsilon \in \{0.2, 0.5\}$, we cross-validate over noise levels $\sigma \in \{0.1, 0.2, 0.5\}$ to find the optimal $\sigma$ that maximizes mean certified radius. For $\alpha$-smoothing, we cross-validate over both $\sigma \in \{0.1, 0.2, 0.5\}$ and trimming rates $\alpha \in \{0.35, 0.49\}$ to find the best combination. The optimal $\alpha$ value is $0.49$ for all cases. We then compare both methods at their respective optimal parameter settings. All experiments use $N = 5,000$ samples, $P = 0.9$ success probability for both methods (denoting the target acceptance probability for $\alpha$-smoothing vs. statistical confidence for ours), and certify on the same 10 randomly sampled test points per function. Detailed function definitions and ground truth computation methodology are provided in Appendix J.1.

Table 1 shows the comparison at optimal parameter settings. Our method consistently achieves larger mean certified radii across all functions and tolerance levels.

### 7.2. MNIST Rotation Certification and Comparison

We evaluate our certification methods on the MNIST rotation prediction task, a practical high-dimensional regression problem where the goal is to predict the rotation angle of a rotated handwritten digit. We use an E(2)-equivariant neural network (E2CNN) (Weiler & Cesa, 2019) trained to predict rotation angles in the range $[-\pi, \pi]$ radians ($[-180°, 180°]$ degrees). The task is well-suited for certification evaluation because: (1) the output is naturally bounded (angles are periodic), (2) the input space is high-dimensional (784 pixels),

*Table 1.* Comparison on synthetic functions after cross-validating $\sigma$ (and $(\sigma, \alpha)$ for $\alpha$-smoothing) on a fixed grid; selected $\alpha$-smoothing rows use $\alpha = 0.49$. Mean certified radius is averaged over ten test points at the selected configuration. Soundness shown in parentheses.

| Function | $\varepsilon$ | Method | Best $\sigma$ | Mean Cert. | Soundness |
|---|---|---|---|---|---|
| Quadratic | 0.2 | $(C, G)$ | 0.20 | 0.114 | (100%) |
| | | $\alpha$-smoothing | 0.10 | 0.073 | (100%) |
| | 0.5 | $(C, G)$ | 0.50 | 0.250 | (100%) |
| | | $\alpha$-smoothing | 0.10 | 0.231 | (100%) |
| Slice | 0.2 | $(C, G)$ | 0.50 | 0.351 | (100%) |
| | | $\alpha$-smoothing | 0.20 | 0.233 | (100%) |
| | 0.5 | $(C, G)$ | 0.50 | 0.603 | (100%) |
| | | $\alpha$-smoothing | 0.50 | 0.489 | (100%) |
| Sandwich | 0.2 | $(C, G)$ | 0.50 | 0.241 | (100%) |
| | | $\alpha$-smoothing | 0.50 | 0.102 | (100%) |
| | 0.5 | $(C, G)$ | 0.50 | 0.448 | (100%) |
| | | $\alpha$-smoothing | 0.20 | 0.186 | (100%) |

and (3) small pixel perturbations can induce large changes in continuous prediction (Zhou et al., 2020; Wang et al., 2021), making robustness certification critical for deployment. We evaluate on 100 stratified test samples (10 per digit class) from the rotated MNIST test set, using multiple noise levels $\sigma \in \{0.06, 0.12, 0.25, 0.50, 0.75\}$ and output tolerance $\epsilon = 10°$ (approximately 0.175 radians). All methods are evaluated at a success probability of 0.9 (denoting the target acceptance probability for $\alpha$-smoothing vs. statistical confidence for ours) to ensure a fair comparison . Additional experimental details are provided in Appendix J.3.

Figure 1(a) shows the cumulative distribution functions (CDFs) of certified radii for our two methods—$(E, C) + M$ and $(E, C, G) + M$—and the $\alpha$-smoothing baseline, using the optimal noise level $\sigma$ for each method (determined by maximizing mean certified radius). At their respective best $\sigma$ values, our $(E, C, G) + M$ method achieves a mean certified radius of 0.210 pixels with a median of 0.208 pixels at $\sigma = 0.75$, certifying all 100 samples (0% failure rate at $\epsilon = 10°$). In contrast, the $(E, C) + M$ method achieves a mean radius of 0.090 pixels at its optimal $\sigma = 0.06$, demonstrating that gradient constraints provide substantial improvement (2.33× mean improvement, 99/100 samples). The $\alpha$-smoothing baseline achieves a mean radius of 0.120 pixels at $\sigma = 0.06$ with $P = 0.9$, but fails to certify 6 out of 100 samples (6% abstain rate, certifying zero radius). Our $(E, C, G) + M$ method outperforms $\alpha$-smoothing by 1.76× in mean certified radius and achieves a larger certified radius on 94 out of 100 samples (including all 6 samples where $\alpha$-smoothing fails to certify).

Our method demonstrates robust performance across a wide range of noise levels, while $\alpha$-smoothing exhibits catas-

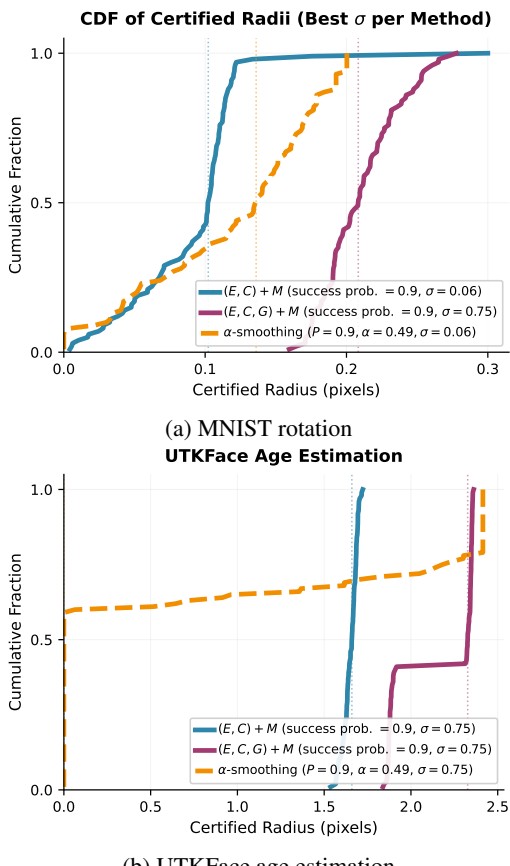

(a) MNIST rotation

(b) UTKFace age estimation

*Figure 1.* CDFs of certified radii using the best fixed configuration for each method. Panel (a) compares methods on MNIST rotation; panel (b) evaluates the aperiodic UTKFace age-estimation task. In both panels, the x-axis reports certified radius in pixels, meaning the $\ell_2$ perturbation norm in normalized pixel space. All methods use success probability 0.9.

trophic failure at large $\sigma$ values (see Appendix J.3 for detailed per-sigma results). At $\sigma \geq 0.5$, $\alpha$-smoothing certifies zero radius on 100% of test points (failing to certify any robustness within $\epsilon = 10°$), while our method maintains robust performance with mean radii of 0.156 and 0.210 pixels at $\sigma = 0.5$ and $\sigma = 0.75$, respectively. This robustness stems from our method's ability to leverage variance and gradient information simultaneously, whereas $\alpha$-smoothing relies solely on probability mass estimation which becomes unreliable when the probability mass $p_A$ becomes too small at large noise levels. The complete failure of $\alpha$-smoothing at moderate-to-large noise levels ($\sigma \geq 0.5$) limits its practical utility, as it restricts the method to a narrow range of noise levels and makes it fragile to hyperparameter selection, whereas our method can leverage larger noise levels to achieve better certified radii.

Although MNIST rotation has a periodic output space, our certificate does not exploit this periodicity: it treats the predictor as a bounded scalar-valued function and uses only

moment and gradient information. To verify that the improvement is not specific to periodic targets, we next evaluate an aperiodic scalar regression task.

### 7.3. Aperiodic Age Estimation

To check that the improvement is not specific to periodic angle prediction, we also certify a face age-estimation task with scalar, aperiodic outputs. We use the UTKFace dataset and evaluate on 100 fixed test images. The base predictor is a pretrained MiVOLO-v2 face-age regressor. We use the model in inference mode without retraining or fine-tuning. We resize each image to $64 \times 64$ RGB, scale pixel values to $[0, 1]$, and add Gaussian smoothing noise in this normalized pixel space. Each noisy image is then bilinearly resized to the model's $384 \times 384$ input resolution and normalized channel-wise using the pretrained model's image normalization constants before inference. We use the known valid age range $[0, 116]$ as the output bound for our $(E, C) + M$ and $(E, C, G) + M$ certificates, with $M = 116$.

We use the same fixed-grid comparison protocol as for MNIST rotation. For our methods, we sweep $\sigma \in \{0.06, 0.12, 0.25, 0.5, 0.75\}$; for $\alpha$-smoothing, we additionally sweep trimming levels $\alpha \in \{0.35, 0.49\}$. The output tolerance is $\epsilon = 6$ years. All methods are evaluated at success probability 0.9, using 5,000 Monte Carlo samples for our statistical estimates and 5,000 samples for the $\alpha$-smoothing probability estimate. Appendix J.4 gives the full details.

Figure 1(b) shows the CDFs using the best fixed configuration for each method, selected by mean certified radius. Both $(E, C) + M$ and $(E, C, G) + M$ select $\sigma = 0.75$, achieving mean certified radii 1.651 and 2.152, respectively, while $\alpha$-smoothing selects $(\sigma, \alpha) = (0.75, 0.49)$ and achieves mean radius 0.817. As in the MNIST rotation task, $\alpha$-smoothing often fails to certify a nonzero radius: it returns radius zero on 59/100 UTKFace test images at its best fixed configuration. Our $(E, C, G) + M$ certificate is larger than $\alpha$-smoothing on 76/100 points. On those points, the average improvement is 1.830; on the 24 points where $\alpha$-smoothing is larger, its average advantage is only 0.234, giving a total gain/loss ratio (the sum of our positive radius gains divided by the sum of $\alpha$-smoothing's positive gains) of 24.74. The appendix reports the corresponding radius–accuracy trade-off for the smoothed age predictor.

### 7.4. Certified Accuracy Analysis

To evaluate the practical utility of our certificates, we employ three complementary metrics that disentangle the quantity (coverage) from the quality (precision) of the certified predictions.

First, following standard protocols (Cohen et al., 2019), we define *absolute certified accuracy* (ACR) at radius $R$. This measures the overall utility of the system: what fraction of the test set is both robust and correct?

$$\text{cert\_acc}_{\text{abs}}(R) = \frac{1}{n} \sum_{i=1}^{n} \mathbf{1}\{d(\hat{y}_i, y_i) \leq \epsilon_{\text{corr}} \wedge r_i \geq R\},$$
(10)

where $n$ is the total number of samples, $d(\cdot, \cdot)$ is the circular distance, and $\epsilon_{\text{corr}} = 10°$ is the correctness tolerance.

Second, to assess the reliability of valid certificates, we measure *conditional certified accuracy*. This asks: among the subset of samples $\mathcal{S}_R = \{i : r_i \geq R\}$ that represent valid certificates, what fraction are actually correct?

$$\text{cert\_acc}_{\text{cond}}(R) = \frac{1}{|\mathcal{S}_R|} \sum_{i \in \mathcal{S}_R} \mathbf{1}\{d(\hat{y}_i, y_i) \leq \epsilon_{\text{corr}}\}. \quad (11)$$

Third, to capture the continuous nature of regression beyond binary correctness, we measure the *certified mean distance*. This quantifies the average precision of samples that meet the robustness requirement:

$$\text{cert\_mean\_dist}(R) = \frac{1}{|\mathcal{S}_R|} \sum_{i \in \mathcal{S}_R} d(\hat{y}_i, y_i). \quad (12)$$

Table 2 presents these metrics on the MNIST rotation task. To ensure a fair comparison across radii, for each method we select a single fixed noise level $\sigma$ that maximizes the area under the certified accuracy curve (or minimizes mean distance error).

**Results and Analysis.** Our $(E, C, G) + M$ method demonstrates superior performance across all metrics. In terms of **utility** (Absolute Accuracy), it maintains 95% accuracy up to $R = 0.15$ pixels, whereas the baseline $(E, C) + M$ drops to 2% and $\alpha$-smoothing drops to 39%. In terms of **reliability** (Conditional Accuracy), all methods achieve high scores (near 100%), indicating that when a certificate is issued, the prediction is reliably within the 10° tolerance. Finally, regarding **precision** (Mean Distance), our method achieves a mean error of 3.90° at $R = 0.15$, significantly lower than the 4.65° of $\alpha$-smoothing and 5.80° of $(E, C) + M$. Notably, at the largest radius threshold $R = 0.25$, our method not only certifies 5% of the data (where others certify roughly 0%) but does so with exceptional precision, achieving a mean distance of just 1.94°.

### 7.5. Certificate Validation: Tightness and Soundness

We validate our certificates through two complementary analyses: tightness (comparing certified radii to optimization-based radii) and soundness (verifying that certified radii do not exceed empirically-found worst-case radii).

*Table 2.* Certified metrics at different radius thresholds $R$ (in pixels) on MNIST rotation task. For each method, we select a single $\sigma$ value that maximizes the sum of certified accuracies (for absolute and conditional accuracy) or minimizes the sum of mean distances (for mean distance) across all thresholds, then report metrics at that fixed $\sigma$ for all $R$ values. Metrics are defined in Equations (10), (11), and (12), respectively. $(E,C)+M$ uses $\sigma = 0.06$ for accuracy and $\sigma = 0.06$ for distance; $(E,C,G)+M$ uses $\sigma = 0.75$ for accuracy and $\sigma = 0.75$ for distance; $\alpha$-smoothing uses $\sigma = 0.06$ for absolute accuracy, $\sigma = 0.12$ for conditional accuracy, and $\sigma = 0.12$ for distance.

| Method | Radius Threshold $R$ (pixels) | | | | |
|---|---|---|---|---|---|
| | 0.05 | 0.10 | 0.15 | 0.20 | 0.25 |
| *Absolute Accuracy (%)* | | | | | |
| $(E,C)+M$ | 81 | 57 | 2 | 1 | 1 |
| $(E,C,G)+M$ | 95 | 95 | 95 | 54 | 5 |
| $\alpha$-smoothing $(P=0.9)$ | 80 | 65 | 39 | 7 | 0 |
| *Conditional Accuracy (%)* | | | | | |
| $(E,C)+M$ | 99 | 98 | 100 | 100 | 100 |
| $(E,C,G)+M$ | 98 | 100 | 100 | 100 | 100 |
| $\alpha$-smoothing $(P=0.9)$ | 100 | 100 | 100 | 100 | 100 |
| *Mean Distance (degrees)* | | | | | |
| $(E,C)+M$ | 3.49 | 3.58 | 5.80 | 7.36 | 7.36 |
| $(E,C,G)+M$ | 4.36 | 4.39 | 3.90 | 4.09 | 1.94 |
| $\alpha$-smoothing $(P=0.9)$ | 3.56 | 3.86 | 4.65 | 5.33 | 5.65 |

*Table 3.* Tightness analysis for 82 samples (18 excluded for hitting search bound). Mean ratio: $3.41\times$, median: $3.48\times$. Two samples (2%) have ratio $< 1.0$.

| Method | $\sigma$ | Mean Cert. | Mean Opt. | Mean Ratio | Median Ratio | % Capped |
|---|---|---|---|---|---|---|
| $(E,C,G)+M$ | 0.5 | 0.146 | 0.504 | $3.41\times$ | $3.48\times$ | 18.0% |

**Tightness Analysis.** We compare certified radii with optimization-based radii computed using projected gradient descent (PGD) (Madry et al., 2018) to find worst-case adversarial perturbations. We evaluate on 100 MNIST rotation samples at $\sigma = 0.5$ using $(E,C,G)+M$. Of these, 18 samples hit the PGD search bound $R_{\max} = 5.0$ pixels and are excluded from ratio analysis (these samples have optimization-based radius $\geq 5.0$ but unknown exact value). For the remaining 82 samples, Table 3 and Figure 2 show a mean ratio of $3.41\times$ (median $3.48\times$), indicating optimization-based radii are $3.4\times$ larger than certified radii on average. Most samples have ratios between $2\times$ and $5\times$, demonstrating consistent conservatism. The $3.4\times$ average gap is comparable to the discrepancy between certified and empirical radii observed in robust classification, where true robustness often exceeds certified bounds by factors of two or more (Cohen et al., 2019). Detailed methodology is in Appendix J.6.

**Soundness Validation.** Samples with ratio $\geq 1.0$ are empirically sound (certified radius $\leq$ optimization-based radius). Of 100 samples, 98 (98%) are sound: 80 of the 82 an-

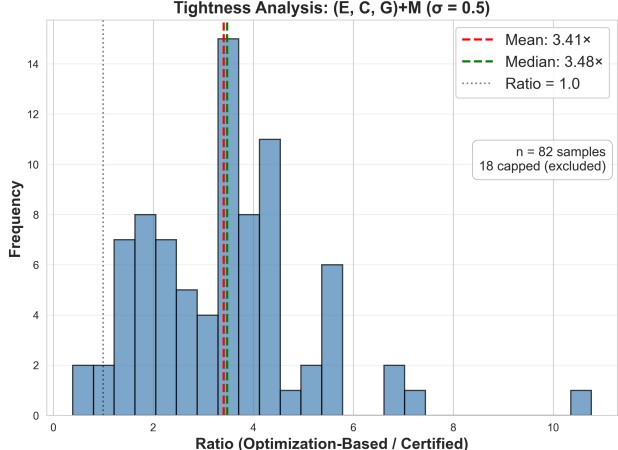

*Figure 2.* Tightness ratio distribution for 82 samples at $\sigma = 0.5$. Dotted line marks ratio $= 1.0$ (soundness threshold); 2 samples fall below.

alyzed samples have ratio $\geq 1.0$, and all 18 samples hitting the search bound are definitely sound (optimization-based $\geq 5.0 >$ any certified radius). The 2 samples (2%) with ratio $< 1.0$ (smallest: $0.39\times$) reflect finite-sample estimation error, expected with 95% confidence intervals (theoretical 5% failure rate vs. observed 2%). This confirms our method behaves as theoretically predicted. See Appendix J.6 for detailed boundary analysis.

## Impact Statement

This paper presents work whose goal is to advance the field of Machine Learning, specifically by enhancing the reliability and safety of regression models. Our framework for certifying robustness has potential positive societal impacts in safety-critical domains such as autonomous navigation and medical imaging, where guaranteeing that continuous predictions remain within safe margins is essential. We do not feel there are specific negative ethical or societal consequences that must be highlighted here.

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

The Appendix is organized as follows. Appendix A provides an extended discussion of related work, placing our contributions in the context of existing adversarial defense and certification literature. Appendix B establishes fundamental lemmas and propositions, including the application of Stein's Lemma and variance lower bounds, which serve as building blocks for our proofs. Appendix C introduces the Calculus of Variations framework and details how our certification objectives are formulated as variational problems. Appendix D clarifies the role of the mean value constraint in bounded versus unbounded settings. We then provide the complete derivations for our certified radii: Appendix E covers the unbounded variance-only setting (Section 5.1), Appendix F details the unbounded gradient-constrained setting (Section 5.2), Appendix G derives the solution for the bounded setting with mean and variance constraints, and Appendix H presents the proofs for our fully constrained setting (Section 5.3), including the dimensionality reduction theorem. Finally, Appendix I details the statistical estimation procedures using U-statistics, and Appendix J provides comprehensive experimental details, including model architectures, hyperparameters, and additional validation results.

## A. Extended Related Work

**Historical context: From attacks to certification.** Early work on adversarial robustness focused on empirical defenses and attack strategies (Chakraborty et al., 2018; Costa et al., 2024). As adaptive attacks systematically defeated many empirical defenses, the field pivoted toward provable robustness. This produced a spectrum of certification paradigms, including mixed-integer formulations and convex relaxations. However, randomized smoothing (Cohen et al., 2019) eventually emerged as the standard for scalable $\ell_2$ certification in modern deep networks, trading off the tightness of exact verification for the ability to scale to high-dimensional datasets like ImageNet.

**Regularization and local sensitivity.** Complementary to certification, several works use input-gradient regularization to control local sensitivity during training. Finlay & Oberman (2021) derives per-example robustness bounds from the loss gradient norm and a modulus-of-continuity term. While not a smoothing method per se, its objective—shrinking local change by limiting first-order behavior—aligns with our use of gradient constraints. Our work differs by providing post-hoc certification for smoothed models rather than a training-time penalty, though the geometric intuition is similar.

**Detailed comparison with existing regression methods.** Recent works by Rekavandi et al. (2025) and Rekavandi et al. (2024) formulate robustness as a probability of inclusion. Specifically, Rekavandi et al. (2025) defines a neighborhood around the ground-truth value and estimates a lower confidence bound on the probability that the noisy observation falls within this box. Rekavandi et al. (2024) improves upon this for unbounded outputs using $\alpha$-trimming. A theoretical limitation of these approaches is their reliance on a fixed anchor (such as the base prediction $f(x)$ or ground truth $y$) to define the "safe" region. While it is possible to experimentally center these regions on the smoothed prediction, doing so violates the independence assumptions required for their guarantees, as the center becomes a stochastic variable. By contrast, our method analytically certifies the stability of the value $g(x)$ itself, ensuring the certificate is theoretically valid for the actual deployed prediction.

**Theoretical limitations of smoothing.** A parallel body of work maps out the inherent limitations of smoothing-based approaches. Kumar et al. (2020) and Wu et al. (2021) show that for $\ell_2$ certification, the certified radius must inevitably decrease as $\sigma/\sqrt{d}$ in high dimensions to maintain accuracy. Similarly, Blum et al. (2020) argue that for $\ell_\infty$ perturbations, randomized smoothing may be fundamentally incapable of nontrivial certification for natural image distributions. Our work focuses on tightening the constant factor of these bounds via gradient information, but is subject to the same fundamental dimensional scaling laws inherent to the smoothing framework.

## B. Useful Lemmas and Propositions

**Lemma B.1** (Integrability of the Gradient). *Let $\boldsymbol{e} \sim \mathcal{N}\left(0, \sigma^2 I\right)$. If the function $f$ has bounded second moment $\mathbb{E}_{\boldsymbol{e}}\left[f(\boldsymbol{z} + \boldsymbol{e})^2\right] < \infty$, then the expectation of the gradient of the likelihood ratio is finite*

$$\int \left|f(\boldsymbol{z} + \boldsymbol{e}) \frac{\boldsymbol{e}}{\sigma^2} p(\boldsymbol{e})\right| d\boldsymbol{e} < \infty$$

*Proof.* We apply the Cauchy-Schwarz inequality for integrals. We can view the integral above as the inner product of

$|f(z + e)|$ and $|\frac{e}{\sigma^2}|$ weighted by the density $p(e)$. Applying the inequality yields:

$$\int \left| f(z + e)\frac{e}{\sigma^2} \right| p(e)\, de \leq \left( \int f(z + e)^2 p(e)\, de \right)^{1/2} \left( \int \left\| \frac{e}{\sigma^2} \right\|^2 p(e) de \right)^{1/2}$$

The first factor on the right-hand side is finite by our assumption. The second factor is the square root of the second moment of the Gaussian distribution, which is known to be finite. Since both factors are finite, their product is finite. $\square$

**Lemma B.2.** *Under the conditions of Lemma B.1, the smoothed function $g(z)$ is differentiable, and its gradient is given by Stein's Lemma:*

$$\nabla g(z) = \frac{1}{\sigma^2}\mathbb{E}_e[f(\boldsymbol{x} + e)e]$$

*Proof.* We first write out $g(z)$ in integral from. That is, $g(z) = \int f(z + e)p(e)\, de$ where $p(e)$ is gaussian measure centered at 0. We apply the change of variable $\boldsymbol{x} = z + e$, which implies $e = \boldsymbol{x} - z$, and $de = d\boldsymbol{x}$. The integral becomes: $g(z) = \int f(\boldsymbol{x})p(\boldsymbol{x} - z)\, d\boldsymbol{x}$.

We compute the gradient with respect to $z$ by differentiating under the integral sign:

$$\nabla g(z) = \int f(\boldsymbol{x})\nabla_z p(\boldsymbol{x} - z)\, d\boldsymbol{x}$$

This interchange of derivative and integral is justified by the Dominated Convergence Theorem, as Lemma B.1 establishes that the gradient of the integrand is absolutely integrable.

Substituting the gradient of the Gaussian density, $\nabla_z p(\boldsymbol{x} - z) = p(\boldsymbol{x} - z)\frac{\boldsymbol{x} - z}{\sigma^2}$, we obtain:

$$\nabla g(z) = \int f(\boldsymbol{x})\frac{\boldsymbol{x} - z}{\sigma^2}p(\boldsymbol{x} - z)\, d\boldsymbol{x}.$$

Reverting the change of variables with $\boldsymbol{x} = z + e$, we obtain:

$$\nabla g(z) = \int f(z + e)\frac{e}{\sigma^2}p(e)\, de = \frac{1}{\sigma^2}\mathbb{E}[f(z + e)e].$$

$\square$

**Proposition B.3** (Variance Lower Bound). *Let $f : \mathbb{R}^d \to \mathbb{R}$ be a square-integrable function with respect to the Gaussian measure $\mathcal{N}(\mathbf{z}, \sigma^2 I)$. Let $C = Var(f)$ be the variance and let $\mathbf{G} = \frac{1}{\sigma^2}\mathbb{E}[f(\mathbf{x})(\mathbf{x} - \mathbf{z})]$ be the gradient vector defined by Stein's Lemma. Then, the following inequality holds:*

$$C \geq \sigma^2\|\mathbf{G}\|_2^2. \tag{13}$$

*Equality holds if and only if $f(\mathbf{x})$ is an affine function, i.e., $f(\mathbf{x}) = \mathbf{G}^\top(\mathbf{x} - \mathbf{z}) + \mathbb{E}[f(\boldsymbol{x})]$.*

*Proof.* Let $\mathbf{e} = \mathbf{x} - \mathbf{z} \sim \mathcal{N}(0, \sigma^2 I)$. The gradient constraint is given by $\mathbb{E}[f(\mathbf{x})\mathbf{e}] = \sigma^2\mathbf{G}$. Consider the scalar random variable $Y$ defined as the projection of the noise along the gradient direction:

$$Y = \mathbf{G}^\top \mathbf{e}.$$

The variance of this projection is:

$$\text{Var}(Y) = \mathbf{G}^\top \text{Cov}(\mathbf{e})\mathbf{G} = \mathbf{G}^\top(\sigma^2 I)\mathbf{G} = \sigma^2\|\mathbf{G}\|_2^2.$$

Next, we compute the covariance between the function $f(\mathbf{x})$ and the projection $Y$. Since $\mathbb{E}[Y] = 0$, we have:

$$\begin{aligned}
\text{Cov}(f(\mathbf{x}), Y) &= \mathbb{E}[(f(\mathbf{x}) - \mathbb{E}[f])Y] \\
&= \mathbb{E}[f(\mathbf{x})\mathbf{e}^\top\mathbf{G}] \\
&= (\mathbb{E}[f(\mathbf{x})\mathbf{e}])^\top\mathbf{G} \\
&= (\sigma^2\mathbf{G})^\top\mathbf{G} \\
&= \sigma^2\|\mathbf{G}\|_2^2.
\end{aligned}$$

Applying the Cauchy-Schwarz inequality to the covariance, we have $\text{Cov}(f, Y)^2 \leq \text{Var}(f)\text{Var}(Y)$. Substituting our terms:

$$(\sigma^2 \|\mathbf{G}\|_2^2)^2 \leq C \cdot (\sigma^2 \|\mathbf{G}\|_2^2).$$

Assuming $\|\mathbf{G}\|_2 > 0$ (otherwise the statement is trivial), we divide by $\sigma^2 \|\mathbf{G}\|_2^2$ to obtain $C \geq \sigma^2 \|\mathbf{G}\|_2^2$. Equality in the Cauchy-Schwarz inequality holds if and only if $f(\mathbf{x})$ is linearly related to $Y$ almost everywhere. That is, $f(\mathbf{x}) - \mathbb{E}[f(\boldsymbol{x})] = aY = a\mathbf{G}^\top(\mathbf{x} - \mathbf{z})$ for some constant $a$.

Plug the affine form into the gradient constraint.

$$\begin{aligned}
\mathbb{E}[f(\mathbf{x})\mathbf{e}] &= \mathbb{E}\left[\left(a\mathbf{G}^\top\mathbf{e} + \mathbb{E}[f]\right)\mathbf{e}\right] \\
&= a\mathbb{E}\left[(\mathbf{G}^\top\mathbf{e})\mathbf{e}\right] + \mathbb{E}[f]\underbrace{\mathbb{E}[\mathbf{e}]}_{0}
\end{aligned}$$

Matching the gradient constraint, and given $\mathbb{E}[\boldsymbol{e}\boldsymbol{e}^\top] = \sigma^2 I$, we have $a = 1$.

$\square$

# C. Our Problem as Calculus of Variations Problem

The most basic problem in the calculus of variations is to find a function, let's call it $y(\boldsymbol{x})$, that extremizes (maximizes or minimizes) a functional $I[y]$ of the form:

$$I[y] = \int L(\boldsymbol{x}, y, y') \, dx \tag{14}$$

where $L$ is some function of the independent variable $x$, the function $y(\boldsymbol{x})$, and its derivative $y'(\boldsymbol{x})$.

When we have constraints, we use the method of Lagrange multipliers. For a problem of the form:

- **Extremize:** $I[y] = \int L(\boldsymbol{x}, y, y') \, d\boldsymbol{x}$
- **Subject to:** $J[y] = \int K(\boldsymbol{x}, y, y') \, d\boldsymbol{x} = \text{Constant}$

We form a new Lagrangian functional, $\mathcal{L}[y] = I[y] - \lambda J[y]$, and find the function $y(\boldsymbol{x})$ that makes it stationary by solving its Euler-Lagrange equation.

$$\frac{\partial L}{\partial f} - \frac{d}{d\boldsymbol{x}}\frac{\partial L}{\partial f'} = 0$$

### C.1. Mapping Our Problem to the Standard Form

In our specific problem, the unknown function we are trying to find is $f(\boldsymbol{x})$. We now map our objective and constraints to the standard form described above.

### C.2. The Functional to Maximize, $I[f]$

Our objective is to maximize the change in the smoothed function $g(\boldsymbol{z})$, which is given by:

$$I[f] = \int f(\boldsymbol{x})p_{z+\delta}(\boldsymbol{x}) \, d\boldsymbol{x} - \int f(\boldsymbol{x})p_z(\boldsymbol{x}) \, d\boldsymbol{x}$$

We can combine this into a single integral to identify the integrand $L$:

$$I[f] = \int \underbrace{f(\boldsymbol{x})\left(p_{\boldsymbol{z}+\delta}(\boldsymbol{x}) - p_{\boldsymbol{z}}(\boldsymbol{x})\right)}_{L(\boldsymbol{x}, f(\boldsymbol{x}))} \, d\boldsymbol{x}$$

So, the integrand of our main functional is $L(\boldsymbol{x}, f(\boldsymbol{x})) = f(\boldsymbol{x})(p_{\boldsymbol{z}+\delta}(\boldsymbol{x}) - p_{\boldsymbol{z}}(\boldsymbol{x}))$. Notice that this function **does not depend on the derivative,** $f'(\boldsymbol{x})$.

## C.3. The Constraint Functionals, $J_i[f]$

We have two types of constraints that we can express as functionals.

- **Variance Constraint:** The constraint is $\text{Var}_{p_z}(f(\boldsymbol{x})) \leq C$. We can write this functional as:

$$J_1[f] = \int \underbrace{(f(\boldsymbol{x}) - g(\boldsymbol{z}))^2 p_{\boldsymbol{z}}(\boldsymbol{x})}_{K_1(\boldsymbol{x}, f(\boldsymbol{x}))} \, d\boldsymbol{x} \leq C,$$

  where $g(\boldsymbol{z}) = \int f(\boldsymbol{x}) p_{\boldsymbol{z}}(\boldsymbol{x}) d\boldsymbol{x}$ and depends on $f$. The integrand for this constraint is $K_1(\boldsymbol{x}, f(\boldsymbol{x})) = (f(\boldsymbol{x}) - g(\boldsymbol{z}))^2 p_{\boldsymbol{z}}(\boldsymbol{x})$. Again, this does not depend on $f'(\boldsymbol{x})$.

- **Gradient Constraint:** This is a vector of $d$ constraints, where the $i$-th component is:

$$J_{2,i}[f] = \int \underbrace{f(\boldsymbol{x}) \frac{\boldsymbol{x}_i - \boldsymbol{z}_i}{\sigma^2} p_{\boldsymbol{z}}(\boldsymbol{x})}_{K_{2,i}(\boldsymbol{x}, f(\boldsymbol{x}))} \, d\boldsymbol{x} = G_i$$

  The integrand is $K_{2,i}(\boldsymbol{x}, f(\boldsymbol{x})) = f(\boldsymbol{x}) \frac{\boldsymbol{x}_i - \boldsymbol{z}_i}{\sigma^2} p_{\boldsymbol{z}}(\boldsymbol{x})$. This also does not depend on $f'(\boldsymbol{x})$.

## C.4. The Full Lagrangian Functional

We combine the objective and constraints using our Lagrange multipliers, the scalar $\lambda$ and the vector $\boldsymbol{\mu}$:

$$\mathcal{L}[f] = I[f] - \lambda J_1[f] - \boldsymbol{\mu}^T \mathbf{J_2}[f]$$

Writing this with a single integrand , $L_{\text{total}}$, we get:

$$\mathcal{L}[f] = \int L_{\text{total}}(\boldsymbol{x}, f(\boldsymbol{x}) \, d\boldsymbol{x} = \int \left( L(\boldsymbol{x}, f) - \lambda K_1(\boldsymbol{x}, f) - \boldsymbol{\mu}^T \mathbf{K_2}(\boldsymbol{x}, f) \right) \, d\boldsymbol{x}$$

## C.5. The Simplified Euler-Lagrange Equation

The full Euler-Lagrange equation for the functional $\mathcal{L}[f]$ is:

$$\frac{\partial L_{\text{total}}}{\partial f} - \frac{d}{d\boldsymbol{x}} \frac{\partial L_{\text{total}}}{\partial f'} = 0$$

Since none of our integrands ($L$, $K_1$, or $\mathbf{K_2}$) depend on the derivative $f'(\boldsymbol{x})$, the term $\frac{\partial L_{\text{total}}}{\partial f'}$ is identically zero. This makes the entire second part of the Euler-Lagrange equation vanish.

Therefore, the necessary condition for finding the optimal function $f^*(\boldsymbol{x})$ simplifies from a potentially complex differential equation to a simple algebraic equation:

$$\frac{\partial L_{\text{total}}}{\partial f} = 0$$

This final equation is precisely the "functional derivative" step to find the analytical form of the worst-case function $f^*(\boldsymbol{x})$.

## C.6. Details of Derivation of Stationary Condition from Lagrangian Functional

In this section, we show how to derive the stationary condition from the Lagrangian functional, given that $g(\boldsymbol{z})$ is dependent on $f$. We use the optimization problem with only variance constraint (5) to exemplify this. Note other optimization problems which have their stationary condition's corresponding term follow similarly.

Recall the Lagrangian for optimization problem (5) is

$$\mathcal{L}(f, \lambda) = \underbrace{\int f(\boldsymbol{x})(p_{\boldsymbol{z}+\boldsymbol{\delta}}(\boldsymbol{x}) - p_{\boldsymbol{z}}(\boldsymbol{x})) d\boldsymbol{x}}_{\text{Term A}} - \lambda \underbrace{\left( \int (f(\boldsymbol{x}) - g(\boldsymbol{z}))^2 p_{\boldsymbol{z}}(\boldsymbol{x}) d\boldsymbol{x} - C \right)}_{\text{Term B}}$$

where $g(z) = \int f(x)p_z(x)dx$, and $p_z$ is defined as in Section 4 to be $\mathcal{N}(z, \sigma^2 I)$. We focus on the variance term. The functional derivative of Term A is straightforward:

$$\frac{\delta}{\delta f(x)}(\text{Term A}) = p_{z+\delta}(x) - p_z(x)$$

For Term B, we utilize the fact that for a distribution centered at $z$:

$$\text{Var}_e(f) = \frac{1}{2}\int\int(f(z + e') - f(z + e))^2 dP(e)dP(e')$$

where $e$ and $e'$ are independent copies of the same random variable, that is $e, e' \sim \mathcal{N}(0, \sigma^2 I)$.

We calculate the functional derivative by perturbing $f$ to $f + \epsilon h$:

$$\frac{d}{d\epsilon}\text{Var}[f + \epsilon h]\bigg|_{\epsilon=0} = \int\int(f(z + e') - f(z + e))(h(z + e') - h(z + e))dP(e)dP(e')$$

$$= 2\int f(z + e')h(z + e')dP(e') - 2\int\int f(z + e)h(z + e')dP(e)dP(e')$$

$$= 2\int\left[f(z + e') - \underbrace{\int f(z + e)dP(e)}_{g(z)}\right]h(z + e')dP(e')$$

To recover the density form used in the Lagrangian, we perform a change of variables for the outer integral. Let $x = z + e'$. Then $dP(e') = p_z(x)dx$. This yields:

$$\int 2(f(x) - g(z))h(x)p_z(x)dx$$

Thus, variance term's functional derivative with respect to $f(x)$ is $2(f(x) - g(z))p_z(x)$.

Putting functional derivative from Term A and Term B together we get the stationary condition shown in Equation (15):

$$(p_{z+\delta}(x) - p_x(x)) - 2\lambda(f(x) - g(z))p_x(x).$$

## D. Mean Value Constraint

We compare the role of the mean function $g(z) = \mathbb{E}_{p_z}[f]$ in the unbounded and bounded settings. In both cases, the objective function (the harm) $\Delta(\delta) = \mathbb{E}_{p_{z+\delta}}[f] - \mathbb{E}_{p_z}[f]$ is invariant under adding constants to $f$. However, the impact of the mean on the feasible set differs significantly.

In the unbounded setting, the constraints are shift-invariant; thus, the specific value of $g(z)$ is immaterial. In the bounded setting, the pointwise box constraint couples the admissible centered function to the mean $g(z)$ via a shift of the feasible interval. Consequently, an unknown $g(z)$ forces us to consider the worst-case shift, leading to a looser (more conservative) certificate.

**Lemma D.1** (Shift invariance in the unbounded setting). *Let $p_z = \mathcal{N}(z, \sigma^2 I)$. Consider an optimization problem where the objective is*

$$\Delta(\delta; f) = \mathbb{E}_{p_{z+\delta}}[f] - \mathbb{E}_{p_z}[f],$$

*and the constraints depend on $f$ only through the variance $\text{Var}_{p_z}(f)$ and the gradient of the mean $\nabla_z\mathbb{E}_{p_z}[f]$. Then, for any constant $c \in \mathbb{R}$, the objective and constraints are invariant under the transformation $f \mapsto f + c$. Consequently, optimizing over an unknown mean (allowing arbitrary constant shifts) yields the same worst-case value as fixing the mean to any specific value.*

*Proof.* For any constant shift $c$, the objective is unchanged:

$$\Delta(\boldsymbol{\delta}; f + c) = (\mathbb{E}_{p_{\boldsymbol{z}+\boldsymbol{\delta}}}[f] + c) - (\mathbb{E}_{p_{\boldsymbol{z}}}[f] + c) = \Delta(\boldsymbol{\delta}; f).$$

Similarly, the variance is shift-invariant: $\text{Var}_{p_{\boldsymbol{z}}}(f + c) = \text{Var}_{p_{\boldsymbol{z}}}(f)$. Finally, the gradient constraint is preserved because the gradient of a constant is zero:

$$\nabla_{\boldsymbol{z}} \mathbb{E}_{p_{\boldsymbol{z}}}[f + c] = \nabla_{\boldsymbol{z}}(\mathbb{E}_{p_{\boldsymbol{z}}}[f] + c) = \nabla_{\boldsymbol{z}} \mathbb{E}_{p_{\boldsymbol{z}}}[f].$$

Since both the objective and the feasible set are invariant to constant shifts, the optimum is independent of the scalar value of the mean. $\square$

**Lemma D.2** (Mean determines feasible box shift in the bounded setting). *Assume $f(\boldsymbol{x}) \in [-M, M]$ pointwise almost everywhere, and let $E = \mathbb{E}_{p_{\boldsymbol{z}}}[f]$ be a fixed mean value. Define the centered function $\tilde{f}(\boldsymbol{x}) = f(\boldsymbol{x}) - E$. Then:*

1. *The objective is unchanged: $\Delta(\boldsymbol{\delta}; f) = \Delta(\boldsymbol{\delta}; \tilde{f})$.*

2. *The moment constraints transform to: $\mathbb{E}_{p_{\boldsymbol{z}}}[\tilde{f}] = 0$, $\mathbb{E}_{p_{\boldsymbol{z}}}[\tilde{f}^2] = \text{Var}_{p_{\boldsymbol{z}}}(f)$, and $\nabla_{\boldsymbol{z}} \mathbb{E}_{p_{\boldsymbol{z}}}[\tilde{f}] = \nabla_{\boldsymbol{z}} \mathbb{E}_{p_{\boldsymbol{z}}}[f]$.*

3. *The box constraint transforms to a shifted interval: $\tilde{f}(\boldsymbol{x}) \in [-M - E, \ M - E]$.*

*Therefore, in the bounded setting, specifying the mean $E$ is equivalent to specifying the location of the feasible box for the centered function.*

*Proof.* Points 1 and 2 follow immediately from the linearity of expectation and differentiation, identical to the logic in Lemma D.1. For point 3, since $f(\boldsymbol{x}) \in [-M, M]$, subtracting the constant $E$ shifts the inequality to $f(\boldsymbol{x}) - E \in [-M - E, M - E]$. Thus, the centered function $\tilde{f}$ is constrained to a window of width $2M$, but the window's center depends entirely on $E$. $\square$

**Corollary D.3** (Unknown-mean certificate is a supremum over shifted boxes). *Let $\Delta_{\text{univ}}$ denote the worst-case harm in the bounded setting where the mean $E = \mathbb{E}_{p_{\boldsymbol{z}}}[f]$ is unknown (i.e., $f$ is constrained only by the variance, gradient, and pointwise bounds, with $E$ free to vary). Then $\Delta_{\text{univ}}$ is the supremum of the fixed-mean worst-case values:*

$$\Delta_{\text{univ}} = \sup_{E \in [-M, M]} \Delta_{\text{spec}}(E),$$

*where $\Delta_{\text{spec}}(E)$ denotes the worst-case harm for the centered problem with the fixed box shift $\tilde{f} \in [-M - E, M - E]$ defined in Lemma D.2. Consequently, $\Delta_{\text{univ}} \geq \Delta_{\text{spec}}(E_{\text{obs}})$ for any empirically observed mean $E_{\text{obs}}$.*

# E. Proofs in Section 5.1

In this section, we restate and give proof for Proposition 5.1, which gives form for worst case function and closed form for certified radius in unbounded function value setting with variance constraint.

**Proposition 5.1.** *[Certified Radius with Variance Estimate] For fixed nominal point $\boldsymbol{z}$, solution to the optimization problem with variance constraint (5) satisfies*

$$f^*(\boldsymbol{x}) - g(\boldsymbol{z}) = k \left( \frac{p_{\boldsymbol{z}+\boldsymbol{\delta}}(\boldsymbol{x})}{p_{\boldsymbol{z}}(\boldsymbol{x})} - 1 \right)$$

*where $k = \sqrt{\frac{C}{\text{Var}_{p_{\boldsymbol{z}}}(p_{\boldsymbol{z}+\boldsymbol{\delta}}/p_{\boldsymbol{z}})}}$, and $g(\boldsymbol{z}) = \mathbb{E}_{p_{\boldsymbol{z}}} f(\boldsymbol{x})$. In this case, certified radius to ensure $|g(\boldsymbol{z} + \boldsymbol{\delta}) - g(\boldsymbol{z})| < \epsilon$ can be calculated as*

$$R = \sigma \sqrt{\log \left( 1 + \frac{\epsilon^2}{C} \right)}.$$

*Proof.* Since there exists function $f$ such that $g(z + \delta) - g(z) > 0$ (consider for example $f(x) = 1$ for $\frac{p_{z+\delta}(x)}{p_z(x)} > k$ and $f(x) = 0$ otherwise), we can aim to maximize $g(z + \delta) - g(z)$ and drop the absolute value in objective of (5). The objective is:

$$\underset{f}{\text{maximize}} \quad \int f(x)(p_{z+\delta}(x) - p_z(x))dx$$

We can use a Lagrange multiplier $\lambda$ to incorporate the constraint:

$$\mathcal{L}(f, \lambda) = \int f(x)(p_{z+\delta}(x) - p_z(x))dx - \lambda \left( \int (f(x) - g(z))^2 p_z(x)dx - C \right)$$

To find the optimal function $f^*$, we recognize this fits the description of a Calculus of Variations problem which works for the case when $f$ is square integrable with respect to gaussian measure (more details in Section C) We take the functional derivative of $\mathcal{L}$ with respect to $f(x)$ as shown in Section C.6. The stationarity condition requires setting the derivative to zero:

$$\frac{\delta \mathcal{L}}{\delta f(x)} = (p_{z+\delta}(x) - p_x(x)) - 2\lambda(f(x) - g(z))p_x(x) = 0 \tag{15}$$

We first observe that the variance constraint must be active. If $\lambda = 0$, the stationarity condition implies $p_{z+\delta}(x) - p_z(x) = 0$ almost everywhere. This equality holds only if $\delta = 0$. Since we consider a perturbation radius $R > 0$, we must have $\lambda \neq 0$. Given that $\lambda$ corresponds to an inequality constraint, we must have $\lambda \geq 0$, and thus $\lambda > 0$.

Solving for $f^*(x)$, with $\lambda > 0$, we get the form of the worst-case function:

$$2\lambda(f^*(x) - g(z))p_x(x) = p_{z+\delta}(x) - p_z(x)$$

$$f^*(x) - g(z) = k \left( \frac{p_{z+\delta}(x)}{p_z(x)} - 1 \right)$$

where we replaced $1/(2\lambda)$ by $k$. This result is the direct analogue of the Neyman-Pearson lemma: the optimal function is determined by the likelihood ratio of the two distributions.

**Characterizing the Maximum Change** Now we substitute the expression for $f^*(x) - g(z)$ back into the variance constraint at point $z$ to find the constant $k$.

$$C = \text{Var}_{p_z}(f^*) = \mathbb{E}_{p_z}[(f^*(x) - g(z))^2]$$

$$C = \mathbb{E}_{p_z}\left[ \left( k \left( \frac{p_{z+\delta}(x)}{p_z(x)} - 1 \right) \right)^2 \right] = k^2 \mathbb{E}_{p_z}\left[ \left( \frac{p_{z+\delta}(x)}{p_z(x)} - 1 \right)^2 \right]$$

$$C = k^2 \text{Var}_{p_z}\left( \frac{p_{z+\delta}(x)}{p_z(x)} \right)$$

This gives us $k = \sqrt{\frac{C}{\text{Var}_{p_z}(p_{z+\delta}/p_z)}}$.

Next, we calculate the maximum change using $f^*$. Recall the maximum change for fixed $\delta$ is:

$$g(z + \delta) - g(z) = \mathbb{E}_{z+\delta}[f^*(x)] - \mathbb{E}_{p_z}[f^*(x)]$$

We know $\mathbb{E}_{p_z}[f^*(x)] = g(z)$ by construction. So we only need to compute $\mathbb{E}_{p_{z+\delta}}[f^*(x)]$. Plugging in the form of $f^*$:

$$g(z + \delta) - g(z) = \mathbb{E}_{p_{z+\delta}}\left[ g(z) + k \left( \frac{p_{z+\delta}(x)}{p_z(x)} - 1 \right) \right] - g(z)$$

$$= k\mathbb{E}_{p_{z+\delta}}\left[ \frac{p_{z+\delta}(x)}{p_z(x)} - 1 \right]$$

Notice that $\mathbb{E}_{p_{z+\delta}}\left[\frac{p_{z+\delta}}{p_x} - 1\right] = \int \left(\frac{p_{z+\delta}(\boldsymbol{x})}{p_z(\boldsymbol{x})} - 1\right) p_{z+\delta}(\boldsymbol{x})d\boldsymbol{x} = \int \frac{p_{z+\delta}(\boldsymbol{x})^2}{p_z(\boldsymbol{x})}d\boldsymbol{x} - 1$. And $\text{Var}_{p_z}\left(\frac{p_{z+\delta}}{p_z}\right) = \mathbb{E}_{p_z}\left[\left(\frac{p_{z+\delta}}{p_z}\right)^2\right] - \left(\mathbb{E}_{p_z}\left[\frac{p_{z+\delta}}{p_z}\right]\right)^2 = \int \frac{p_{z+\delta}(\boldsymbol{x})^2}{p_z(\boldsymbol{x})}d\boldsymbol{x} - 1$. The terms are identical. Let's call this quantity $V$. That is,

$$V = \int \frac{p_{z+\delta}(\boldsymbol{x})^2}{p_z(\boldsymbol{x})}d\boldsymbol{x} - 1.$$

Therefore we have, $g(\boldsymbol{z} + \boldsymbol{\delta}) - g(\boldsymbol{z}) = k \cdot V = \sqrt{\frac{C}{V}} \cdot V = \sqrt{C \cdot V}$.

Now we just need to compute $V$ for our Gaussian distributions. The integral $\int \frac{p_1(\boldsymbol{x})^2}{p_0(\boldsymbol{x})}d\boldsymbol{x}$ is a standard calculation. The result is:

$$\int \frac{p_1(\boldsymbol{x})^2}{p_0(\boldsymbol{x})}d\boldsymbol{x} = \exp\left(\frac{\|\boldsymbol{\delta}\|^2}{\sigma^2}\right)$$

Therefore, $V = \exp\left(\frac{\|\boldsymbol{\delta}\|^2}{\sigma^2}\right) - 1$. Plugging this back in, we get the maximum possible change for a given $\boldsymbol{\delta}$:

$$g(\boldsymbol{z} + \boldsymbol{\delta}) - g(\boldsymbol{z}) = \sqrt{C\left(\exp\left(\frac{\|\boldsymbol{\delta}\|^2}{\sigma^2}\right) - 1\right)}$$

**Deriving the Certified Radius** We now have all the pieces. We want to guarantee that $|g(\boldsymbol{z}) - g(\boldsymbol{z} + \boldsymbol{\delta})| < \epsilon$. This will hold true as long as $\epsilon$ is greater than the maximum possible change for any $\boldsymbol{\delta}$:

$$\epsilon \geq \max_{\boldsymbol{\delta} \leq R}|g(\boldsymbol{z} + \boldsymbol{\delta}) - g(\boldsymbol{z})| = \sqrt{C\left(\exp\left(\frac{\|\boldsymbol{\delta}\|^2}{\sigma^2}\right) - 1\right)} = \sqrt{C\left(\exp\left(\frac{R}{\sigma^2}\right) - 1\right)}$$

Now, we simply solve for $R$:

$$R \leq \sigma\sqrt{\log\left(1 + \frac{\epsilon^2}{C}\right)}$$

$\square$

## F. Details and Proofs of Section 5.2

In Section 5.2, we derive the worst-case certificate for the setting constrained by the variance $C$ and the gradient vector $\mathbf{G}$ (6).

### F.1. Lagrangian Formulation and Optimal Function Form

Since the problem involves maximizing a linear objective over the intersection of a strictly convex set (the variance ball) and a hyperplane (the gradient constraint), the resulting feasible set remains strictly convex. Consequently, the optimal solution shape is unique (up to a constant shift), and there is only one set of Lagrange multipliers that satisfies the stationarity and constraint conditions.

To solve for the worst-case function $f^*$, we employ the Calculus of Variations. We introduce a scalar Lagrange multiplier $\lambda$ for the variance constraint and a vector multiplier $\boldsymbol{\mu} \in \mathbb{R}^d$ for the gradient constraint. Using the identity of gradient of $g(\boldsymbol{z})$ (equivalently $\mathbb{E}_{p_z}[f]$) derived in Lemma B.2, $\nabla_{\boldsymbol{z}}\mathbb{E}_{p_z}[f] = \mathbb{E}_{p_z}[f(\boldsymbol{x})\frac{\boldsymbol{x} - \boldsymbol{z}}{\sigma^2}]$, we construct the full Lagrangian functional:

$$\mathcal{L}[f, \lambda, \boldsymbol{\mu}] = \int f(\boldsymbol{x})(p_{z+\delta}(\boldsymbol{x}) - p_z(\boldsymbol{x}))d\boldsymbol{x}$$
$$- \lambda\left(\int (f(\boldsymbol{x}) - g(\boldsymbol{z}))^2 p_z(\boldsymbol{x})d\boldsymbol{x} - C\right).$$
$$- \boldsymbol{\mu}^\top\left(\int f(\boldsymbol{x})\frac{\boldsymbol{x} - \boldsymbol{z}}{\sigma^2}p_z(\boldsymbol{x})d\boldsymbol{x} - \mathbf{G}\right)$$

We seek the function $f^*$ that makes this functional stationary. Taking the functional derivative with respect to $f(\boldsymbol{x})$ (as detailed in Appendix Section C.6 regarding the variance term) and setting it to zero yields the Euler-Lagrange condition:

$$\frac{\delta \mathcal{L}}{\delta f(\boldsymbol{x})} = (p_{\boldsymbol{z}+\boldsymbol{\delta}}(\boldsymbol{x}) - p_{\boldsymbol{z}}(\boldsymbol{x})) - 2\lambda(f(\boldsymbol{x}) - g(\boldsymbol{z}))p_{\boldsymbol{z}}(\boldsymbol{x}) - \boldsymbol{\mu}^\top \frac{\boldsymbol{x} - \boldsymbol{z}}{\sigma^2} p_{\boldsymbol{z}}(\boldsymbol{x}) = 0$$

Since the Gaussian density $p_{\boldsymbol{z}}(\boldsymbol{x})$ is strictly positive everywhere, we can divide through by $p_{\boldsymbol{z}}(\boldsymbol{x})$ to isolate the optimal function form. This gives:

$$\left(\frac{p_{\boldsymbol{z}+\boldsymbol{\delta}}(\boldsymbol{x})}{p_{\boldsymbol{z}}(\boldsymbol{x})} - 1\right) - 2\lambda(f(\boldsymbol{x}) - g(\boldsymbol{z})) - \boldsymbol{\mu}^\top \frac{\boldsymbol{x} - \boldsymbol{z}}{\sigma^2} = 0$$

To simplify the notation, we define two auxiliary terms: the *likelihood ratio shift* $A(\boldsymbol{x}, \boldsymbol{\delta})$ and the function $B(\boldsymbol{x})$:

$$A(\boldsymbol{x}, \boldsymbol{\delta}) \triangleq \frac{p_{\boldsymbol{z}+\boldsymbol{\delta}}(\boldsymbol{x})}{p_{\boldsymbol{z}}(\boldsymbol{x})} - 1 = \exp\left(\frac{\boldsymbol{\delta}^\top(\boldsymbol{x} - \boldsymbol{z})}{\sigma^2} - \frac{\|\boldsymbol{\delta}\|^2}{2\sigma^2}\right) - 1 \tag{16}$$

$$B(\boldsymbol{x}) \triangleq \frac{\boldsymbol{x} - \boldsymbol{z}}{\sigma^2} \tag{17}$$

Substituting these definitions into the stationarity condition, we obtain:

$$A(\boldsymbol{x}, \boldsymbol{\delta}) - 2\lambda(f(\boldsymbol{x}) - g(\boldsymbol{z})) - \boldsymbol{\mu}^\top B(\boldsymbol{x}) = 0$$

We observe that the variance constraint must be active, i.e., $\lambda > 0$. If we assume $\lambda = 0$, the stationarity condition implies $A(\boldsymbol{x}, \boldsymbol{\delta}) = \boldsymbol{\mu}^\top B(\boldsymbol{x})$. However, $A(\boldsymbol{x}, \boldsymbol{\delta})$ involves the exponential term $e^{\boldsymbol{\delta}^\top(\boldsymbol{x}-\boldsymbol{z})/\sigma^2}$, whereas $\boldsymbol{\mu}^\top B(\boldsymbol{x})$ is linear in $\boldsymbol{x}$. A non-constant exponential function cannot equal a linear function almost everywhere (they intersect at most on a set of measure zero). Thus, we must have $\lambda \neq 0$. Since $\lambda$ corresponds to an inequality constraint, dual feasibility requires $\lambda \geq 0$, and therefore $\lambda > 0$.

Solving for the centered function $f^*(\boldsymbol{x}) - g(\boldsymbol{z})$:

$$f^*(\boldsymbol{x}) - g(\boldsymbol{z}) = \frac{1}{2\lambda}\left(A(\boldsymbol{x}, \boldsymbol{\delta}) - \boldsymbol{\mu}^\top B(\boldsymbol{x})\right)$$

Again defining the scalar constant $k \triangleq \frac{1}{2\lambda}$, we arrive at the general form of the worst-case function:

$$f^*(\boldsymbol{x}) - g(\boldsymbol{z}) = k\left(A(\boldsymbol{x}, \boldsymbol{\delta}) - \boldsymbol{\mu}^\top B(\boldsymbol{x})\right) \tag{18}$$

This derivation reveals that the optimal function is a linear combination of the likelihood ratio (exponential in $\boldsymbol{x}$) and a function linear in $\boldsymbol{x}$. The unknown constants $k$ and $\boldsymbol{\mu}$ must now be determined by satisfying the primal constraints.

### F.2. Derivation of the Worst-Case Function

Having established the general structure of the optimal function in Section F.1, we must now enforce the primal constraints to determine the Lagrange multipliers. By solving for the scalar $k$ and vector $\boldsymbol{\mu}$ in terms of the problem parameters $(\boldsymbol{\delta}, C, \mathbf{G})$ and substituting them back into the general form, we obtain the explicit worst-case function characterized in the following proposition.

**Proposition 5.2.** *[Worst-Case Function for Fixed Perturbation] Consider the inner maximization of Problem* (6) *with a fixed perturbation vector $\boldsymbol{\delta}$. The base function $f^*$ that maximizes the shift $\mathbb{E}_{p_{\mathbf{z}+\boldsymbol{\delta}}}[f(\boldsymbol{x})] - \mathbb{E}_{p_{\mathbf{z}}}[f(\boldsymbol{x})]$ subject to the variance constraint $\mathrm{Var}_{p_{\mathbf{z}}}(f(\boldsymbol{x})) \leq C$ and gradient constraint $\nabla\mathbb{E}_{p_{\mathbf{z}}}[f(\boldsymbol{x})] = \mathbf{G}$ is a function of the form:*

$$f^*(\boldsymbol{x}) - g(\boldsymbol{z}) = k(\boldsymbol{\delta})\left[\left(e^{\frac{\boldsymbol{\delta}^\top(\boldsymbol{x}-\boldsymbol{z})}{\sigma^2} - \frac{\|\boldsymbol{\delta}\|_2^2}{2\sigma^2}} - 1\right) - \left(\boldsymbol{\delta} - \frac{\sigma^2}{k(\boldsymbol{\delta})}\mathbf{G}\right)^\top \frac{\boldsymbol{x}-\boldsymbol{z}}{\sigma^2}\right] \tag{8}$$

*where $k(\boldsymbol{\delta}) = \sqrt{\dfrac{C - \sigma^2\|\mathbf{G}\|_2^2}{e^{\|\boldsymbol{\delta}\|_2^2/\sigma^2} - 1 - \frac{\|\boldsymbol{\delta}\|_2^2}{\sigma^2}}}$ and $g(\boldsymbol{z}) = \mathbb{E}_{p_{\mathbf{z}}}f(\boldsymbol{x})$.*

To prove this, we first establish three key Gaussian expectations involving the likelihood ratio shift $A(\boldsymbol{x}, \boldsymbol{\delta})$ and the function $B(\boldsymbol{x})$ defined previously.

**Lemma F.1** (Key Gaussian Expectations). *Let $A(\boldsymbol{x}, \boldsymbol{\delta})$ and $B(\boldsymbol{x})$ be defined as in Section F.1. The following identities hold under the nominal distribution $p_{\mathbf{z}}$:*

1. $\mathbb{E}_{p_{\mathbf{z}}}[B(\boldsymbol{x})B(\boldsymbol{x})^\top] = \frac{1}{\sigma^2}I.$

2. $\mathbb{E}_{p_{\mathbf{z}}}[A(\boldsymbol{x}, \boldsymbol{\delta})B(\boldsymbol{x})] = \frac{\boldsymbol{\delta}}{\sigma^2}.$

3. $\mathbb{E}_{p_{\mathbf{z}}}[A(\boldsymbol{x}, \boldsymbol{\delta})^2] = e^{\|\boldsymbol{\delta}\|^2/\sigma^2} - 1.$

*Proof.* Let $\mathbf{v} = \boldsymbol{x} - \boldsymbol{z} \sim \mathcal{N}(0, \sigma^2 I)$. For the first identity, $\mathbb{E}[BB^\top] = \frac{1}{\sigma^4}\mathbb{E}[\mathbf{v}\mathbf{v}^\top] = \frac{1}{\sigma^2}I$. For the second, using Stein's Lemma or the property $\mathbb{E}[\mathbf{v}e^{\mathbf{u}^\top\mathbf{v}}] = \sigma^2\mathbf{u}e^{\frac{1}{2}\sigma^2\|\mathbf{u}\|^2}$, we have:

$$\mathbb{E}[AB] = \frac{e^{-\|\boldsymbol{\delta}\|^2/2\sigma^2}}{\sigma^2}\mathbb{E}\left[\mathbf{v}e^{\frac{\boldsymbol{\delta}^\top\mathbf{v}}{\sigma^2}}\right] - \underbrace{\mathbb{E}[B]}_{0} = \frac{\boldsymbol{\delta}}{\sigma^2}.$$

For the third, the integral $\mathbb{E}_{p_{\mathbf{z}}}[(p_{\mathbf{z}+\boldsymbol{\delta}}/p_{\mathbf{z}} - 1)^2]$ evaluates to $e^{\|\boldsymbol{\delta}\|^2/\sigma^2} - 1$ via standard Gaussian tilting results. $\qquad\square$

*Proof of Proposition 5.2.* We solve for the multipliers using the constraints and the identities from Lemma F.1.

**1. Solving for $\boldsymbol{\mu}$ (Gradient Constraint):** We substitute the general form $f^*(\boldsymbol{x}) - g(\boldsymbol{z}) = k(A - \boldsymbol{\mu}^\top B)$ into the gradient constraint $\mathbb{E}[f^*B] = \mathbf{G}$:

$$\begin{aligned}
\mathbf{G} &= \mathbb{E}_{p_{\mathbf{z}}}[(g(\boldsymbol{z}) + k(A - \boldsymbol{\mu}^\top B))B] \\
&= k\left(\mathbb{E}_{p_{\mathbf{z}}}[AB] - \mathbb{E}_{p_{\mathbf{z}}}[BB^\top]\boldsymbol{\mu}\right).
\end{aligned} \tag{19}$$

Applying Lemma F.1 (Items 1 and 2) to (19):

$$\mathbf{G} = k\left(\frac{\boldsymbol{\delta}}{\sigma^2} - \frac{1}{\sigma^2}I\boldsymbol{\mu}\right).$$

Solving for $\boldsymbol{\mu}$ yields:

$$\boldsymbol{\mu} = \boldsymbol{\delta} - \frac{\sigma^2}{k}\mathbf{G}. \tag{20}$$

**2. Solving for $k$ (Variance Constraint):** We substitute the form of $f^*$ into the variance constraint $\mathbb{E}[(f^* - g)^2] = C$:

$$\begin{aligned}
C &= k^2\mathbb{E}_{p_{\mathbf{z}}}\left[(A - \boldsymbol{\mu}^\top B)^2\right] \\
&= k^2\left(\mathbb{E}_{p_{\mathbf{z}}}[A^2] - 2\boldsymbol{\mu}^\top\mathbb{E}_{p_{\mathbf{z}}}[AB] + \boldsymbol{\mu}^\top\mathbb{E}_{p_{\mathbf{z}}}[BB^\top]\boldsymbol{\mu}\right).
\end{aligned} \tag{21}$$

Using Lemma F.1 in (21) results in:

$$C = k^2(e^{\frac{\|\delta\|^2}{\sigma^2}} - 1) + \frac{k^2}{\sigma^2}\boldsymbol{\mu}^\top(\boldsymbol{\mu} - 2\boldsymbol{\delta}).$$

Substituting $\boldsymbol{\mu} - 2\boldsymbol{\delta} = -(\boldsymbol{\delta} + \frac{\sigma^2}{k}\mathbf{G})$ from (20):

$$\begin{aligned}
\boldsymbol{\mu}^\top(\boldsymbol{\mu} - 2\boldsymbol{\delta}) &= \left(\boldsymbol{\delta} - \frac{\sigma^2}{k}\mathbf{G}\right)^\top \left(-\boldsymbol{\delta} - \frac{\sigma^2}{k}\mathbf{G}\right) \\
&= -\left(\|\boldsymbol{\delta}\|^2 - \frac{\sigma^4}{k^2}\|\mathbf{G}\|_2^2\right).
\end{aligned}$$

Plugging this back into the variance equation:

$$C = k^2\left(e^{\frac{\|\delta\|^2}{\sigma^2}} - 1 - \frac{\|\boldsymbol{\delta}\|^2}{\sigma^2}\right) + \sigma^2\|\mathbf{G}\|_2^2.$$

Rearranging for $k$ yields

$$k(\boldsymbol{\delta}) = \sqrt{\frac{C - \sigma^2\|\mathbf{G}\|_2^2}{e^{\|\delta\|_2^2/\sigma^2} - 1 - \frac{\|\delta\|_2^2}{\sigma^2}}}. \tag{22}$$

Substituting this expression for $k(\boldsymbol{\delta})$ and the solution for $\boldsymbol{\mu}$ from (20) back into the general form (18) yields the specific functional form claimed in (8), completing the proof. $\qquad\square$

### F.3. Worst-Case Harm and Direction

With the optimal function $f^*$ determined for a fixed perturbation $\boldsymbol{\delta}$, we can now compute the worst-case harm and optimize the perturbation direction.

**Proposition 5.3.** *Let $\Delta_{\max}(\boldsymbol{\delta})$ denote the optimal value achieved of objective function in (6). For any radius budget $R > 0$, the maximum of $\Delta_{\max}(\boldsymbol{\delta})$ over all perturbations with norm $R$ is achieved when $\boldsymbol{\delta}$ is aligned with the gradient $\mathbf{G}$. That is:*

$$\arg\max_{\boldsymbol{\delta}:\|\boldsymbol{\delta}\|_2=R} \Delta_{\max}(\boldsymbol{\delta}) = R\frac{\mathbf{G}}{\|\mathbf{G}\|_2}$$

*Proof.* Recall the objective function is $\Delta(\boldsymbol{\delta}) = \mathbb{E}_{p_{z+\delta}}[f] - \mathbb{E}_{p_z}[f]$. Using the likelihood ratio shift $A(\boldsymbol{x}, \boldsymbol{\delta}) = \frac{p_{z+\delta}}{p_z} - 1$, we can rewrite this as an expectation under the nominal density $p_z$:

$$\Delta(\boldsymbol{\delta}) = \mathbb{E}_{p_z}[f(\boldsymbol{x})A(\boldsymbol{x}, \boldsymbol{\delta})].$$

Substituting the general optimal form $f^*(\boldsymbol{x}) - g(\boldsymbol{z}) = k(A - \boldsymbol{\mu}^\top B)$ from Equation (18):

$$\begin{aligned}
\Delta(\boldsymbol{\delta}) &= \mathbb{E}_{p_z}\left[\left(g(\boldsymbol{z}) + k(A - \boldsymbol{\mu}^\top B)\right) A\right] \\
&= g(\boldsymbol{z})\underbrace{\mathbb{E}[A]}_{0} + k\mathbb{E}[A^2] - k\boldsymbol{\mu}^\top\mathbb{E}[BA].
\end{aligned}$$

Using the identities from Lemma F.1 ($\mathbb{E}[A^2] = e^{\|\delta\|^2/\sigma^2} - 1$ and $\mathbb{E}[BA] = \boldsymbol{\delta}/\sigma^2$):

$$\Delta(\boldsymbol{\delta}) = k\left(e^{\frac{\|\delta\|^2}{\sigma^2}} - 1\right) - k\boldsymbol{\mu}^\top\frac{\boldsymbol{\delta}}{\sigma^2}.$$

We substitute $\boldsymbol{\mu} = \boldsymbol{\delta} - \frac{\sigma^2}{k}\mathbf{G}$ from Equation (20):

$$\begin{aligned}
\Delta(\boldsymbol{\delta}) &= k\left(e^{\frac{\|\delta\|^2}{\sigma^2}} - 1\right) - \frac{k}{\sigma^2}\left(\boldsymbol{\delta} - \frac{\sigma^2}{k}\mathbf{G}\right)^\top\boldsymbol{\delta} \\
&= k\left(e^{\frac{\|\delta\|^2}{\sigma^2}} - 1\right) - \frac{k\|\boldsymbol{\delta}\|^2}{\sigma^2} + \mathbf{G}^\top\boldsymbol{\delta} \\
&= k\left(e^{\frac{\|\delta\|^2}{\sigma^2}} - 1 - \frac{\|\boldsymbol{\delta}\|^2}{\sigma^2}\right) + \mathbf{G}^\top\boldsymbol{\delta}.
\end{aligned}$$

Finally, substituting the explicit expression for $k(\boldsymbol{\delta})$ from Equation (22):

$$k(\boldsymbol{\delta}) = \sqrt{\frac{C - \sigma^2 \|\mathbf{G}\|_2^2}{e^{\|\boldsymbol{\delta}\|^2/\sigma^2} - 1 - \|\boldsymbol{\delta}\|^2/\sigma^2}},$$

we obtain the result:

$$\Delta(\boldsymbol{\delta}) = \sqrt{C - \sigma^2 \|\mathbf{G}\|_2^2} \sqrt{e^{\frac{\|\boldsymbol{\delta}\|^2}{\sigma^2}} - 1 - \frac{\|\boldsymbol{\delta}\|^2}{\sigma^2}} + \mathbf{G}^\top \boldsymbol{\delta}. \tag{23}$$

The final step is to maximize this expression for all perturbations $\boldsymbol{\delta}$ with a fixed norm $\|\boldsymbol{\delta}\|_2 = R$. As $k(\boldsymbol{\delta})$ depends on $\boldsymbol{\delta}$ only through its norm $\|\boldsymbol{\delta}\|$, the entire first term is a constant with respect to the direction of $\boldsymbol{\delta}$:

$$\Delta_{\max}(\boldsymbol{\delta}) = k(R)\left(e^{R^2/\sigma^2} - 1 - \frac{R^2}{\sigma^2}\right) + \mathbf{G}^T \boldsymbol{\delta}$$

Therefore, maximizing the objective is equivalent to maximizing the term $\mathbf{G}^T \boldsymbol{\delta}$. By the Cauchy-Schwarz inequality, this dot product is maximized when $\boldsymbol{\delta}$ is aligned with $\mathbf{G}$. The maximizing perturbation must be:

$$\boldsymbol{\delta} = R \frac{\mathbf{G}}{\|\mathbf{G}\|_2}$$

$\square$

**Corollary 5.4.** *For the worst-case perturbation $\boldsymbol{\delta} = R\frac{\mathbf{G}}{\|\mathbf{G}\|_2}$, the worst case shift is:*

$$\Delta_{\max}(R) = \sqrt{C - \sigma^2 \|\mathbf{G}\|_2^2} \cdot \sqrt{e^{R^2/\sigma^2} - 1 - R^2/\sigma^2}$$
$$+ R \|\mathbf{G}\|_2$$

*Proof.* Both results are direct consequences of substituting the worst-case perturbation $\boldsymbol{\delta} = R\frac{\mathbf{G}}{\|\mathbf{G}\|_2}$ into the results from the preceding propositions.

**Proof of (a):** The form of $f^*$ is found by substituting this worst-case $\boldsymbol{\delta}$ into the general form from Proposition 5.2. The terms involving $\boldsymbol{\delta}$ simplify as $\|\boldsymbol{\delta}\|_2$ becomes $R$, and $\boldsymbol{\delta}^T(\boldsymbol{x} - \boldsymbol{z})$ becomes $\frac{R}{\|\mathbf{G}\|_2}\mathbf{G}^T(\boldsymbol{x} - \boldsymbol{z})$. This substitution directly yields the stated expression.

**Proof of (b):** The value $\Delta_{\max}(R)$ is found by substituting the worst-case $\boldsymbol{\delta}$ into (23). Setting $\|\boldsymbol{\delta}\|_2 = R$ and $\boldsymbol{\delta} = R\frac{\mathbf{G}}{\|\mathbf{G}\|_2}$, the dot product simplifies to $\mathbf{G}^T\boldsymbol{\delta} = R\|\mathbf{G}\|_2$. The expression becomes:

$$\Delta_{\max}(R) = \sqrt{C - \sigma^2 \|\mathbf{G}\|_2^2} \cdot \sqrt{e^{R^2/\sigma^2} - 1 - R^2/\sigma^2} + R \|\mathbf{G}\|_2$$

$\square$

# G. Bounded Function Value with Mean and Variance Constraint

Let $p_{\boldsymbol{z}} = \mathcal{N}(\boldsymbol{z}, \sigma^2 I_d)$. We consider the following optimization problem where the function $f$ is constrained by a fixed mean $E$, a variance budget $C$, and pointwise bounds $[-M, M]$:

$$\max_{f, \boldsymbol{\delta}} \quad \Delta(f, \boldsymbol{\delta}) = \mathbb{E}_{p_{\boldsymbol{z}+\boldsymbol{\delta}}}[f(\boldsymbol{x})] - \mathbb{E}_{p_{\boldsymbol{z}}}[f(\boldsymbol{x})] \tag{24}$$
$$\text{s.t.} \quad \|\boldsymbol{\delta}\|_2 \le R$$
$$(E) \quad \mathbb{E}_{p_{\boldsymbol{z}}}[f(\boldsymbol{x})] = E$$
$$(C) \quad \mathbb{E}_{p_{\boldsymbol{z}}}[(f(\boldsymbol{x}) - E)^2] \le C$$
$$(M) \quad -M \le f(\boldsymbol{x}) \le M \quad \forall \boldsymbol{x} \in \mathbb{R}^d$$

**High-Variance vs. Low-Variance Regimes**   Before deriving the worst-case function, we distinguish between two fundamental regimes determined by the relationship between the variance budget $C$ and the geometric bounds of the function space.

A function bounded by $[-M, M]$ with mean $E$ achieves its maximum possible variance when it takes only the extreme values $\{-M, M\}$ (a "bang-bang" solution). The variance of such a function is exactly $M^2 - E^2$. This bound delineates two cases:

1. **The High-Variance Regime** $(C \geq M^2 - E^2)$**:** The variance constraint is redundant. The adversary can construct a step function (Neyman-Pearson solution) using only the values $\{-M, M\}$ that satisfies the variance constraint trivially. This reduces to the standard bounded certification problem (e.g., Cohen et al. (2019), Levine et al. (2020)).

2. **The Low-Variance Regime** $(C < M^2 - E^2)$**:** The variance constraint is strictly tighter than the natural bound imposed by $M$. The adversary is precluded from using a simple step function and must construct a solution that transitions smoothly or is clipped to satisfy the stricter variance budget.

In the context of robust regression, we are specifically interested in the **Low-Variance Regime**, as it characterizes functions that possess limited volatility. Consequently, we proceed under the assumption that $C < M^2 - E^2$.

### G.1. Derivation of the 1-Dimensional Reduction

We claim that the $d$-dimensional optimization problem can be reduced to a 1-dimensional optimization problem over a scalar function $\phi(t)$ by exploiting the symmetry of the Gaussian distribution and the isotropic nature of the constraints.

**Symmetry and Projection**   Since the prior distribution $p_{\boldsymbol{z}}$ is isotropic and the constraints $(E)$, $(C)$, and $(M)$ are rotationally invariant, the problem is symmetric with respect to the direction of perturbation. Without loss of generality, we fix the perturbation direction. Let $\boldsymbol{\delta}$ be an arbitrary perturbation with magnitude $\alpha = \|\boldsymbol{\delta}\|_2$. We define the unit vector $\boldsymbol{u} = \boldsymbol{\delta}/\alpha$.

We decompose the input space relative to $\boldsymbol{u}$. Let $t = \boldsymbol{u}^\top(\boldsymbol{x} - \boldsymbol{z})$ be the projection of the input $\boldsymbol{x}$ onto the direction of perturbation. We define a 1-dimensional function $\phi : \mathbb{R} \to \mathbb{R}$ as the conditional expectation of $f(\boldsymbol{x})$ given the projection $t$:

$$\phi(t) = \mathbb{E}_{p_{\boldsymbol{z}}}[f(\boldsymbol{x}) \mid \boldsymbol{u}^\top(\boldsymbol{x} - \boldsymbol{z}) = t] \tag{25}$$

**Equivalence of the Objective Function**   The likelihood ratio between the shifted distribution $p_{\boldsymbol{z}+\boldsymbol{\delta}}$ and the base distribution $p_{\boldsymbol{z}}$ depends only on the projection $t$:

$$w(t) \triangleq \frac{p_{\boldsymbol{z}+\boldsymbol{\delta}}(\boldsymbol{x})}{p_{\boldsymbol{z}}(\boldsymbol{x})} - 1 = \exp\left(\frac{\alpha t}{\sigma^2} - \frac{\alpha^2}{2\sigma^2}\right) - 1 \tag{26}$$

Using the tower property of conditional expectation, the objective function becomes:

$$\begin{aligned}
\mathbb{E}_{p_{\boldsymbol{z}+\boldsymbol{\delta}}}[f(\boldsymbol{x})] - \mathbb{E}_{p_{\boldsymbol{z}}}[f(\boldsymbol{x})] &= \mathbb{E}_{p_{\boldsymbol{z}}}\left[f(\boldsymbol{x})\left(\frac{p_{\boldsymbol{z}+\boldsymbol{\delta}}(\boldsymbol{x})}{p_{\boldsymbol{z}}(\boldsymbol{x})} - 1\right)\right] \\
&= \mathbb{E}_{p_{\boldsymbol{z}}}[f(\boldsymbol{x})w(t)] \\
&= \mathbb{E}_{T \sim \mathcal{N}(0,\sigma^2)}[\mathbb{E}[f(\boldsymbol{x}) \mid T = t]w(t)] \\
&= \mathbb{E}_{T \sim \mathcal{N}(0,\sigma^2)}[\phi(T)w(T)]
\end{aligned} \tag{27}$$

**Preservation of Constraints**   We verify that the projected function $\phi(t)$ satisfies the constraints if $f(\boldsymbol{x})$ does.

- **Mean (E):** $\mathbb{E}_{p_{\boldsymbol{z}}}[f(\boldsymbol{x})] = \mathbb{E}_T[\mathbb{E}[f(\boldsymbol{x})|T]] = \mathbb{E}_T[\phi(T)] = E$.

- **Boundedness (M):** Since $-M \leq f(\boldsymbol{x}) \leq M$ holds almost surely, the conditional expectation satisfies $-M \leq \phi(t) \leq M$ almost surely.

- **Variance (C):** Applying Jensen's Inequality to the convex function $h(y) = (y - E)^2$:

$$(\phi(t) - E)^2 = (\mathbb{E}[f(\boldsymbol{x}) - E \mid t])^2 \le \mathbb{E}[(f(\boldsymbol{x}) - E)^2 \mid t]$$

Taking the expectation over $T$ yields $\mathbb{E}_T[(\phi(T) - E)^2] \le C$.

Thus, maximizing the original objective is equivalent to finding the worst-case 1-D function $\phi(t)$ and shift $\alpha \in [0, R]$.

## G.2. The Lagrangian Formulation and Derivation of Optimal Form

We solve the inner optimization problem for the function $\phi$ given a fixed shift $\alpha$. Once the optimal $\phi_\alpha^*$ is found, we maximize over $\alpha \in [0, R]$. The primal optimization problem is:

$$\max_\phi \quad \int \phi(t)w(t)p_0(t)\, dt$$

$$\text{s.t.} \quad \int (\phi(t) - E)^2 p_0(t)\, dt \le C \quad \text{(Variance)}$$

$$\int \phi(t)p_0(t)\, dt = E \quad \text{(Mean)}$$

$$-M \le \phi(t) \le M \quad \forall t \in \mathbb{R} \quad \text{(Boundedness)}$$

We construct the Lagrangian $\mathcal{L}$ with Lagrange multipliers $\lambda$ (variance), $\mu$ (mean), $\eta_1(t)$ (upper bound), and $\eta_2(t)$ (lower bound).

$$\mathcal{L}(\phi, \lambda, \mu, \eta_1, \eta_2) = \int \phi(t)w(t)p_0(t)\, dt$$

$$- \lambda \left( \int (\phi(t) - E)^2 p_0(t)\, dt - C \right)$$

$$- \mu \left( \int \phi(t)p_0(t)\, dt - E \right)$$

$$- \int \eta_1(t)(\phi(t) - M)p_0(t)\, dt$$

$$- \int \eta_2(t)(-\phi(t) - M)p_0(t)\, dt$$

Grouping terms, the Lagrangian becomes:

$$\mathcal{L} = \int \left[ \phi(t)w(t) - \lambda(\phi(t) - E)^2 - \mu\phi(t) - \eta_1(t)(\phi(t) - M) + \eta_2(t)(\phi(t) + M) \right] p_0(t)\, dt + \text{const}$$

OPTIMALITY CONDITIONS

**1. Stationarity Condition**
Taking the functional derivative of $\mathcal{L}$ with respect to $\phi(t)$ and setting it to zero gives the necessary condition for the optimal solution $\phi^*$:

$$\left. \frac{\delta\mathcal{L}}{\delta\phi(t)} \right|_{\phi^*} = (w(t) - 2\lambda(\phi^*(t) - E) - \mu - \eta_1(t) + \eta_2(t)) \, p_0(t) = 0$$

Rearranging for $\phi^*(t)$:

$$2\lambda(\phi^*(t) - E) = w(t) - \mu - \eta_1(t) + \eta_2(t) \tag{28}$$

**2. KKT Conditions**
For $\phi^*(t)$ to be optimal, it must satisfy:

- **Primal Feasibility:** $-M \leq \phi^*(t) \leq M$.

- **Dual Feasibility:** $\lambda \geq 0$, $\eta_1(t) \geq 0$, and $\eta_2(t) \geq 0$.

- **Complementary Slackness:**

$$\lambda(\mathbb{E}[(\phi^* - E)^2] - C) = 0, \quad \eta_1(t)(\phi^*(t) - M) = 0, \quad \eta_2(t)(-\phi^*(t) - M) = 0$$

PROOF THAT THE VARIANCE CONSTRAINT IS ACTIVE ($\lambda > 0$)

We prove that $\lambda > 0$ by contradiction, given the assumption that $C < M^2 - E^2$. Assume the variance constraint is inactive ($\lambda = 0$). The stationarity condition (Eq. 28) becomes $\eta_1(t) - \eta_2(t) = w(t) - \mu$. Based on the sign of $w(t) - \mu$, one of the multipliers must be positive, forcing $\phi^*(t)$ to the boundary:

- If $w(t) > \mu \implies \eta_1(t) > 0 \implies \phi^*(t) = M$.

- If $w(t) < \mu \implies \eta_2(t) > 0 \implies \phi^*(t) = -M$.

Thus, if $\lambda = 0$, the optimal function $\phi^*(t)$ takes only the extreme values $\{-M, M\}$ almost everywhere. The variance of such a solution is maximal: $\mathrm{Var}(\phi^*) = M^2 - E^2$. This contradicts the Low-Variance Regime assumption that $\mathrm{Var}(\phi) \leq C < M^2 - E^2$. Therefore, the assumption $\lambda = 0$ is false, and we must have $\lambda > 0$.

DERIVING THE OPTIMAL FORM

With $\lambda > 0$, we define the *unconstrained candidate function* $h(t)$:

$$h(t; \lambda, \mu) \triangleq E + \frac{w(t) - \mu}{2\lambda}$$

Analyzing the KKT conditions for the active/inactive bound cases yields the clipped solution:

$$\phi^*(t) = \mathrm{clip}_{[-M,M]}\left(E + \frac{w(t) - \mu}{2\lambda}\right) \tag{29}$$

### G.3. Numerical Procedure

**Solving for the Lagrange Multipliers** The multipliers $\lambda$ and $\mu$ are determined by enforcing the variance and mean constraints. Let $\mathbf{v} = [\lambda, \mu]^\top$ be the vector of unknowns. We define the residual vector $\mathbf{F}(\lambda, \mu)$:

$$F_1(\lambda, \mu) \triangleq \mathbb{E}_T\left[(\phi^*(T; \lambda, \mu) - E)^2\right] - C = 0 \tag{30}$$

$$F_2(\lambda, \mu) \triangleq \mathbb{E}_T\left[\phi^*(T; \lambda, \mu)\right] - E = 0 \tag{31}$$

**Monotonicity and Bisection** Let $W(R) \triangleq \max_{\|\boldsymbol{\delta}\|_2 \leq R} \Delta(\boldsymbol{\delta})$. Due to rotational invariance, this is equivalent to maximizing over the shift magnitude $\alpha \in [0, R]$. Since the set of allowable shifts grows with $R$, $W(R)$ is non-decreasing. This validates the use of bisection search for the certified radius.

**Numerical Root Finding** The system $\mathbf{F}(\lambda, \mu) = \mathbf{0}$ corresponds to the stationarity conditions of the dual problem. Since the primal problem is convex with linear constraints, the dual function is concave. This ensures that the solution $[\lambda^*, \mu^*]$ is unique and corresponds to the global optimum. We solve this system using a standard numerical root-finding algorithm (e.g., hybrid Powell method or Levenberg-Marquardt).

---

**Algorithm 2** Calculating the Certified Radius (Bounded Mean+Variance)

---

1: **function** ComputeRadius($C, E, \epsilon, \sigma, M, r_{\text{high}}$, tol)
2:    **Input:** Variance $C$, Mean $E$, threshold $\epsilon$, noise $\sigma$, bounds $[-M, M]$.
3:    **Output:** The certified radius $R$.
     ▷ Subroutine to compute worst-case harm for a fixed shift $\alpha$
4:    **function** WorstHarm($\alpha$)
5:       **1. Setup:** Define $w(t) = \exp(\alpha t/\sigma^2 - \alpha^2/2\sigma^2) - 1$.
6:       **2. Solve Dual:** Find roots $\lambda^*, \mu^*$ of Eq. (30), (31).
7:       *Method:* Standard root solver (e.g., `scipy.optimize.root`).
8:       *Implicitly:* $\phi^*(t) = \text{clip}_{[-M,M]}\left(E + \frac{w(t)-\mu^*}{2\lambda^*}\right)$.
9:       **3. Evaluate:** $\Delta \leftarrow \mathbb{E}_T[\phi^*(T)w(T)]$.
10:      **return** $\Delta$
     ▷ Main Loop: Bisection search for $R$
11:   $r_{\text{low}} \leftarrow 0, \quad r_{\text{high}} \leftarrow r_{\text{high}}$
12:   **while** $(r_{\text{high}} - r_{\text{low}}) > $ tol **do**
13:      $r_{\text{mid}} \leftarrow r_{\text{low}} + (r_{\text{high}} - r_{\text{low}})/2$
14:      $val \leftarrow \text{WorstHarm}(r_{\text{mid}})$
15:      **if** $val < \epsilon$ **then**
16:        $r_{\text{low}} \leftarrow r_{\text{mid}}$
17:      **else**
18:        $r_{\text{high}} \leftarrow r_{\text{mid}}$
19:   **return** $R \leftarrow r_{\text{low}}$

---

## H. Bounded Function with Mean, Variance and Gradient Constraint

In this section, we show in the bounded function setting, with mean, variance and gradient constraint, the nested optimization of finding the worst case perturbation in $d$-dimension $\|\delta\| \leq R$ while finding the worst case function $f$ can be reduced to finding a 1-dimensional perturbation $\alpha \in \mathbb{R}$, and a function $\phi$ which takes one dimensional input. We first show that the worst case perturbation $\delta$ has to align with gradient $\mathbf{G}$'s direction. This result allows us to reduce the problem on $d$-dimensional variable to be on 1-dimension variable.

**Theorem H.1** (Existence of an aligned optimal perturbation). *Let $p_{\mathbf{z}} = \mathcal{N}(\mathbf{z}, \sigma^2 I_d)$ and fix $R > 0$, $C > 0$, $M > 0$, and $\mathbf{G} \in \mathbb{R}^d$. Consider*

$$\max_{f,\boldsymbol{\delta}} \quad \Delta(\boldsymbol{\delta}) \triangleq \mathbb{E}_{p_{\mathbf{z}+\delta}}[f(\mathbf{x})] - \mathbb{E}_{p_{\mathbf{z}}}[f(\mathbf{x})]$$

$$s.t. \quad \|\boldsymbol{\delta}\|_2 = R,$$
$$(E) \quad \mathbb{E}_{p_{\mathbf{z}}}[(f(\mathbf{x})] = E,$$
$$(C) \quad \mathbb{E}_{p_{\mathbf{z}}}[(f(\mathbf{x}) - E)^2] \leq C,$$
$$(G) \quad \mathbb{E}_{p_{\mathbf{z}}}[f(\mathbf{x})\boldsymbol{B}(\mathbf{x})] = \mathbf{G}, \qquad \boldsymbol{B}(\mathbf{x}) \triangleq \frac{\mathbf{x} - \mathbf{z}}{\sigma^2},$$
$$(M) \quad -M \leq f(\mathbf{x}) \leq M$$

*Assume the feasible set is nonempty. Let $\mathbf{u}$ be the unit vector in $\mathbf{G}$'s direction, i.e. $\mathbf{u} \triangleq \frac{\mathbf{G}}{\|\mathbf{G}\|_2}$. Then there exists an optimal pair $(f^\star, \boldsymbol{\delta}^\star)$ such that $\boldsymbol{\delta}^\star$ is aligned with $\mathbf{G}$, i.e. $\boldsymbol{\delta}^\star \in \{+R\mathbf{u}, -R\mathbf{u}\}$. Moreover, in the aligned case the $d$-dimensional problem reduces losslessly to a 1-dimensional optimization problem.*

*Proof.* We follow four steps in this proof. We first center the function $f$ to handle the mean constraint. In step 1, we rewrite the objective and constraints to show that because of rotational invariance the problem depends only on the geometry of $\mathbf{G}$ and $\boldsymbol{\delta}$. In Step 2, we project the problem onto the 2D subspace spanned by $\mathbf{G}$ and $\boldsymbol{\delta}$. In Step 3, we set up gaussian coordinate system $(U, V)$ aligned with $\boldsymbol{\delta}$ and express the function we are optimizing over in terms of $(U, V)$. In step 4, we decompose the function we are optimizing over into two components $l(U)$ which is useful for maximizing objective, and

$r(U, V)$ which is the residual. We argue misalignment of $\boldsymbol{\delta}$ with $\mathbf{G}$ (their angle $\theta \neq 0$) forces the residual to be nonzero to satisfy the gradient constraint, which reduces the variance budget $C$. Thus $\theta = 0$ achieves optimal.

**Preprocessing (Centering).**  Define the centered function $\tilde{f}(\boldsymbol{x}) \triangleq f(\boldsymbol{x}) - E$. Since $\mathbb{E}_{p_{\boldsymbol{z}}}[\boldsymbol{B}(\boldsymbol{x})] = \boldsymbol{0}$, the gradient constraint (G) is unchanged. The objective is invariant to additive constants, so $\Delta(\boldsymbol{\delta}) = \mathbb{E}_{p_{\boldsymbol{z}+\boldsymbol{\delta}}}[\tilde{f}(\boldsymbol{x})] - \mathbb{E}_{p_{\boldsymbol{z}}}[\tilde{f}(\boldsymbol{x})]$.

We translate the constraints to $\tilde{f}$. The constraint $\mathbb{E}_{p_{\boldsymbol{z}}}[f(\boldsymbol{x})] = E$ implies $\mathbb{E}_{p_{\boldsymbol{z}}}[\tilde{f}(\boldsymbol{x}) + E] = E$. Thus $\mathbb{E}_{p_{\boldsymbol{z}}}[\tilde{f}(\boldsymbol{x})] = 0$. About variance constraint, $\mathbb{E}_{p_{\boldsymbol{z}}}[(f(\boldsymbol{x}) - E)^2] \leq C$. Substituting $f(\boldsymbol{x}) = \tilde{f}(\boldsymbol{x}) + E$, we get $\mathbb{E}_{p_{\boldsymbol{z}}}[\tilde{f}(\boldsymbol{x})^2] \leq C$. The boundedness constraint shifts by $E$: $-M - E \leq \tilde{f}(\boldsymbol{x}) \leq M - E$.

**Step 1 (Likelihood-ratio form and rotation invariance).**  Let $\boldsymbol{e} \triangleq \boldsymbol{x} - \boldsymbol{z}$. Under the null distribution $p_{\boldsymbol{z}}$, we have $\boldsymbol{e} \sim \mathcal{N}(0, \sigma^2 I)$. We rewrite the objective using change of measure from the shifted distribution $p_{\boldsymbol{z}+\boldsymbol{\delta}}$ back to $p_{\boldsymbol{z}}$. Recall that $\Delta(\boldsymbol{\delta}) = \mathbb{E}_{p_{\boldsymbol{z}+\boldsymbol{\delta}}}[\tilde{f}(\boldsymbol{x})] - \mathbb{E}_{p_{\boldsymbol{z}}}[\tilde{f}(\boldsymbol{x})]$. Writing this as an integral over densities:

$$
\begin{aligned}
\Delta(\boldsymbol{\delta}) &= \int \tilde{f}(\boldsymbol{x}) \left( \frac{p_{\boldsymbol{z}+\boldsymbol{\delta}}(\boldsymbol{x})}{p_{\boldsymbol{z}}(\boldsymbol{x})} - 1 \right) p_{\boldsymbol{z}}(\boldsymbol{x}) \, d\boldsymbol{x} \\
&= \mathbb{E}_{\boldsymbol{x} \sim p_{\boldsymbol{z}}} \left[ \tilde{f}(\boldsymbol{x}) \left( \frac{p_{\boldsymbol{z}+\boldsymbol{\delta}}(\boldsymbol{x})}{p_{\boldsymbol{z}}(\boldsymbol{x})} - 1 \right) \right].
\end{aligned}
\tag{32}
$$

The density ratio for Gaussians $\mathcal{N}(\boldsymbol{z} + \boldsymbol{\delta}, \sigma^2 I)$ and $\mathcal{N}(\boldsymbol{z}, \sigma^2 I)$ is:

$$
\begin{aligned}
\frac{p_{\boldsymbol{z}+\boldsymbol{\delta}}(\boldsymbol{x})}{p_{\boldsymbol{z}}(\boldsymbol{x})} &= \frac{\exp\left(-\frac{1}{2\sigma^2}\|\boldsymbol{x} - (\boldsymbol{z}+\boldsymbol{\delta})\|_2^2\right)}{\exp\left(-\frac{1}{2\sigma^2}\|\boldsymbol{x} - \boldsymbol{z}\|_2^2\right)} \\
&= \exp\left(-\frac{1}{2\sigma^2}\left(\|\boldsymbol{e} - \boldsymbol{\delta}\|_2^2 - \|\boldsymbol{e}\|_2^2\right)\right) \\
&= \exp\left(-\frac{1}{2\sigma^2}\left(\|\boldsymbol{e}\|_2^2 - 2\boldsymbol{\delta}^\top\boldsymbol{e} + \|\boldsymbol{\delta}\|_2^2 - \|\boldsymbol{e}\|_2^2\right)\right) \\
&= \exp\left(\frac{\boldsymbol{\delta}^\top\boldsymbol{e}}{\sigma^2} - \frac{\|\boldsymbol{\delta}\|_2^2}{2\sigma^2}\right).
\end{aligned}
$$

Substituting this back into (32):

$$
\begin{aligned}
\Delta(\boldsymbol{\delta}) &= \mathbb{E}_{\boldsymbol{e} \sim \mathcal{N}(0, \sigma^2 I)}\left[ f(\boldsymbol{z} + \boldsymbol{e}) \left( \exp\left(\frac{\boldsymbol{\delta}^\top\boldsymbol{e}}{\sigma^2} - \frac{\|\boldsymbol{\delta}\|_2^2}{2\sigma^2}\right) - 1 \right) \right] \\
&=: \mathbb{E}\left[ f(\boldsymbol{z} + \boldsymbol{e}) \, A_{\boldsymbol{\delta}(\boldsymbol{e})} \right],
\end{aligned}
\tag{33}
$$

where $A_{\boldsymbol{\delta}(\boldsymbol{e})} \triangleq \exp\left(\frac{\boldsymbol{\delta}^\top\boldsymbol{e}}{\sigma^2} - \frac{\|\boldsymbol{\delta}\|^2}{2\sigma^2}\right) - 1$.

*Rotational Invariance.* We now rigorously establish that the problem depends only on the relative geometry of $\mathbf{G}$ and $\boldsymbol{\delta}$. Let $Q \in \mathbb{R}^{d \times d}$ be any orthogonal matrix ($Q^\top Q = I$). Consider the change of variables $\boldsymbol{e}' = Q\boldsymbol{e}$. Since the standard normal distribution is rotationally symmetric, $\boldsymbol{e}' \stackrel{d}{=} \boldsymbol{e}$. For any feasible function $\tilde{f}$, define the rotated function $\tilde{f}_Q(\boldsymbol{v}) \triangleq \tilde{f}(\boldsymbol{z} + Q^\top\boldsymbol{v})$. Substituting $\boldsymbol{e} = Q^\top\boldsymbol{e}'$ into the objective and constraints:

- **Objective:** The term $\boldsymbol{\delta}^\top\boldsymbol{e}$ becomes $\boldsymbol{\delta}^\top(Q^\top\boldsymbol{e}') = (Q\boldsymbol{\delta})^\top\boldsymbol{e}'$. Thus, the objective value $\Delta(\boldsymbol{\delta})$ using $\tilde{f}$ is identical to $\Delta(Q\boldsymbol{\delta})$ using $\tilde{f}_Q$.

- **Gradient Constraint (G):**

$$
\mathbb{E}[\tilde{f}(\boldsymbol{z} + \boldsymbol{e})\boldsymbol{e}] = \mathbb{E}[\tilde{f}_Q(\boldsymbol{e}')(Q^\top\boldsymbol{e}')] = Q^\top\mathbb{E}[\tilde{f}_Q(\boldsymbol{e}')\boldsymbol{e}'].
$$

  Thus, if the original satisfies $\mathbb{E}[\tilde{f}\boldsymbol{e}] = \sigma^2\mathbf{G}$, the rotated function satisfies $\mathbb{E}[\tilde{f}_Q\boldsymbol{e}'] = \sigma^2(Q\mathbf{G})$.

- **Scalar Constraints (C, M):**

- *Boundedness Constraint (M):* The set of values taken by the rotated function $\tilde{f}_Q$ is identical to the set of values taken by $\tilde{f}$. Formally, for any $v \in \mathbb{R}^d$, $\tilde{f}_Q(v) = \tilde{f}(z + Q^\top v)$. Since the map $v \mapsto z + Q^\top v$ is a bijection on $\mathbb{R}^d$,

$$\inf_v \tilde{f}_Q(v) = \inf_x \tilde{f}(x) \quad \text{and} \quad \sup_v \tilde{f}_Q(v) = \sup_x \tilde{f}(x).$$

Thus, if $-M - E \leq \tilde{f}(x) \leq M - E$ for all $x$, then $-M - E \leq \tilde{f}_Q(v) \leq M - E$ for all $v$.

- *Variance Constraint (C):* We utilize the rotational invariance of the Gaussian measure. We can write the expectation of the original function in terms of the rotated variable:

$$\mathbb{E}_e\big[\tilde{f}(z + e)^2\big] = \mathbb{E}_{e'}\big[\tilde{f}(z + Q^\top e')^2\big] = \mathbb{E}_{e'}\big[\tilde{f}_Q(e')^2\big].$$

Thus, $\mathbb{E}[\tilde{f}^2] \leq C \iff \mathbb{E}[\tilde{f}_Q^2] \leq C$.

*Conclusion of Step 1 (Geometric Dependence).* Let $V(\mathbf{G}, \boldsymbol{\delta})$ denote the maximum value of the objective $\Delta(\boldsymbol{\delta})$ over all feasible functions $f$, for fixed vectors $\mathbf{G}$ and $\boldsymbol{\delta}$ (and fixed scalars $C, M$). The rotational invariance established above implies:

$$V(\mathbf{G}, \boldsymbol{\delta}) = V(Q\mathbf{G}, Q\boldsymbol{\delta}) \quad \text{for any orthogonal } Q.$$

This equality means the value of the problem depends solely on the intrinsic geometry: the norms $\|\mathbf{G}\|_2$, $\|\boldsymbol{\delta}\|_2$ and the inner product $\mathbf{G}^\top \boldsymbol{\delta}$. Consequently, we do not need to solve the optimization for every possible vector orientation. We can analyze the problem by fixing the vectors $\mathbf{G}$ and $\boldsymbol{\delta}$ and decomposing the space relative to them. This justifies the decomposition in the next step, where we restrict our analysis to the subspace spanned by these two vectors.

**Step 2 (Reduction to the 2D plane spanned by G and $\boldsymbol{\delta}$).** Fix $\boldsymbol{\delta}$ and set $u \triangleq \mathbf{G}/\|\mathbf{G}\|_2$ (if $\mathbf{G} = 0$, alignment is trivial). Define the subspace $\mathsf{S} \triangleq \mathrm{span}\{u, \boldsymbol{\delta}\}$. Let $\Pi_\mathsf{S}$ be the orthogonal projection onto $\mathsf{S}$. We decompose the Gaussian vector $e$ into independent components:

$$e = e_\| + e_\perp, \quad \text{where } e_\| \triangleq \Pi_\mathsf{S} e \in \mathsf{S} \text{ and } e_\perp \perp \mathsf{S}.$$

Given any feasible candidate $\tilde{f}(z + e)$, we define its "smoothed" projection $\bar{f}$ on $\mathsf{S}$ via the conditional expectation:

$$\bar{f}(e_\|) \triangleq \mathbb{E}_{e_\perp}\big[\tilde{f}(z + e_\| + e_\perp) \mid e_\|\big].$$

We now verify that $\bar{f}$ preserves or improves the objective and all constraints:

- **Objective (preserved):** The likelihood ratio term $A_{\boldsymbol{\delta}}(e)$ depends on $e$ only through the inner product $\boldsymbol{\delta}^\top e$. Since $\boldsymbol{\delta} \in \mathsf{S}$, we have $\boldsymbol{\delta}^\top e_\perp = 0$, so $A_{\boldsymbol{\delta}}(e) = A_{\boldsymbol{\delta}}(e_\|)$. Applying the tower property of conditional expectation:

$$\mathbb{E}[\tilde{f}(z + e)A_{\boldsymbol{\delta}}(e)] = \mathbb{E}\big[\mathbb{E}[\tilde{f}(z + e)A_{\boldsymbol{\delta}}(e_\|) \mid e_\|]\big] = \mathbb{E}\big[A_{\boldsymbol{\delta}}(e_\|)\mathbb{E}[\tilde{f}(z + e) \mid e_\|]\big] = \mathbb{E}[\bar{f}(e_\|)A_{\boldsymbol{\delta}}(e_\|)].$$

Thus, $\bar{f}$ achieves the same objective value as $\tilde{f}$.

- **Gradient Constraint (preserved):** The original constraint is the full vector equation $\mathbb{E}[\tilde{f}(z + e)e] = \sigma^2 \mathbf{G}$.

We calculate the gradient of the reduced function $\bar{f}$ by decomposing the expectation into two orthogonal terms:

$$\mathbb{E}[\bar{f}(e_\|)e] = \mathbb{E}[\bar{f}(e_\|)e_\|] + \mathbb{E}[\bar{f}(e_\|)e_\perp].$$

We evaluate these two terms separately:

- *First term (Parallel component):* Using the definition of $\bar{f}$ and the property $\mathbb{E}[\mathbb{E}[X|Y]Y] = \mathbb{E}[XY]$, we have:

$$\mathbb{E}\big[\bar{f}(e_\|)\,e_\|\big] = \mathbb{E}\Big[\mathbb{E}\big[\tilde{f}(z + e_\| + e_\perp) \mid e_\|\big]\,e_\|\Big] = \mathbb{E}\big[\tilde{f}(z + e_\| + e_\perp)\,e_\|\big].$$

This is exactly the projection of the original constraint onto $\mathsf{S}$. Since the original $\tilde{f}$ was feasible, $\mathbb{E}[\tilde{f}(z + e)e_\|] = \Pi_\mathsf{S}(\sigma^2 \mathbf{G}) = \sigma^2 \mathbf{G}$. Thus, the first term equals $\sigma^2 \mathbf{G}$.

– *Second term (Perpendicular component):* In the Gaussian distribution, the orthogonal component $\boldsymbol{e}_\perp$ is independent of the subspace component $\boldsymbol{e}_\parallel$. Consequently, $\boldsymbol{e}_\perp$ is independent of the function value $\bar{f}(\boldsymbol{e}_\parallel)$. Using independence and the zero-mean property of $\boldsymbol{e}_\perp$:

$$\mathbb{E}\big[\bar{f}(\boldsymbol{e}_\parallel)\,\boldsymbol{e}_\perp\big] = \mathbb{E}\big[\bar{f}(\boldsymbol{e}_\parallel)\big] \cdot \mathbb{E}[\boldsymbol{e}_\perp] = \mathbb{E}\big[\bar{f}(\boldsymbol{e}_\parallel)\big] \cdot \mathbf{0} = \mathbf{0}.$$

Combining these terms, we recover the original constraint:

$$\mathbb{E}[\bar{f}(\boldsymbol{e}_\parallel)\boldsymbol{e}] = \sigma^2 \mathbf{G} + \mathbf{0} = \sigma^2 \mathbf{G}.$$

- **Variance Constraint (improved):** By the conditional Jensen's inequality applied to the convex function $x \mapsto x^2$:

$$\bar{f}(\boldsymbol{e}_\parallel)^2 = \big(\mathbb{E}[\tilde{f}(\boldsymbol{z}+\boldsymbol{e}) \mid \boldsymbol{e}_\parallel]\big)^2 \leq \mathbb{E}[\tilde{f}(\boldsymbol{z}+\boldsymbol{e})^2 \mid \boldsymbol{e}_\parallel].$$

Taking expectations over $\boldsymbol{e}_\parallel$ yields $\mathbb{E}[\bar{f}^2] \leq \mathbb{E}[\tilde{f}^2] \leq C$.

- **Boundedness Constraint (preserved):** Since $-M - E \leq \tilde{f}(\boldsymbol{z}+\boldsymbol{e}) \leq M - E$, the monotonicity of expectation implies the conditional mean $\bar{f}(\boldsymbol{e}_\parallel)$ satisfies the same bounds almost surely.

*Conclusion of Step 2.* For any feasible function $\tilde{f}$ in the original high-dimensional space, there exists a function $\bar{f}$ depending only on the projection $\boldsymbol{e}_\parallel \in \mathsf{S}$ that is feasible and yields the same objective value. Hence, we may restrict the optimization domain to functions defined on the 2D plane $\mathsf{S}$ without loss of generality.

**Step 3 (Basis rotation and constraint projection).** We now rigorously map the problem from the high-dimensional vector space to the 2D plane $\mathsf{S}$. Construct an orthonormal basis $(\boldsymbol{u}_\theta, \boldsymbol{u}_{\theta\perp})$ for $\mathsf{S}$ where: $\boldsymbol{u}_\theta \triangleq \boldsymbol{\delta}/\|\boldsymbol{\delta}\|_2$ is the direction aligned with the perturbation; and $\boldsymbol{u}_{\theta\perp}$ is the unit vector in $\mathsf{S}$ orthogonal to $\boldsymbol{u}_\theta$ such that the gradient direction $\boldsymbol{u} \triangleq \mathbf{G}/\|\mathbf{G}\|_2$ has a non-negative projection onto it. By construction, the unit vector $\boldsymbol{u}$ decomposes as:

$$\boldsymbol{u} = \cos\theta\,\boldsymbol{u}_\theta + \sin\theta\,\boldsymbol{u}_{\theta\perp},$$

where $\theta \in [0, \pi]$ is the angle between $\mathbf{G}$ and $\boldsymbol{\delta}$.

Define the scalar coordinates $U \triangleq \boldsymbol{u}_\theta^\top \boldsymbol{e}$ and $V \triangleq \boldsymbol{u}_{\theta\perp}^\top \boldsymbol{e}$. Since $\boldsymbol{u}_\theta, \boldsymbol{u}_{\theta\perp} \in \mathsf{S}$, these are exactly the coordinates of the projection $\boldsymbol{e}_\parallel$ in this basis:

$$\boldsymbol{e}_\parallel = U\boldsymbol{u}_\theta + V\boldsymbol{u}_{\theta\perp}.$$

Since $\boldsymbol{e} \sim \mathcal{N}(0, \sigma^2 I)$ and the basis vectors are orthonormal, $U$ and $V$ are independent $\mathcal{N}(0, \sigma^2)$ random variables.

We define the scalar representative function $h : \mathbb{R}^2 \to \mathbb{R}$ as:

$$h(u, v) \triangleq \bar{f}(u\boldsymbol{u}_\theta + v\boldsymbol{u}_{\theta\perp}).$$

Substituting the random variables, we have the identity $h(U, V) = \bar{f}(\boldsymbol{e}_\parallel)$. We now translate the problem constraints using $h(U, V)$:

- **Objective:** Recalling that $A_{\boldsymbol{\delta}}(\boldsymbol{e})$ depends only on $\boldsymbol{\delta}^\top \boldsymbol{e} = RU$:

$$\mathbb{E}\big[\bar{f}(\boldsymbol{e}_\parallel)A_{\boldsymbol{\delta}}(\boldsymbol{e}_\parallel)\big] = \mathbb{E}\big[h(U,V)A(U)\big], \quad \text{where } A(U) \triangleq \exp\left(\frac{R}{\sigma^2}U - \frac{R^2}{2\sigma^2}\right) - 1. \tag{34}$$

- **Gradient Constraints:** From Step 2, we know $\bar{f}$ satisfies the projected vector constraint $\mathbb{E}[\bar{f}(\boldsymbol{e}_\parallel)\boldsymbol{e}_\parallel] = \sigma^2 \mathbf{G}$. Substituting $\boldsymbol{e}_\parallel = U\boldsymbol{u}_\theta + V\boldsymbol{u}_{\theta\perp}$ into the left-hand side:

$$\mathbb{E}\big[h(U,V)(U\boldsymbol{u}_\theta + V\boldsymbol{u}_{\theta\perp})\big] = \sigma^2\|\mathbf{G}\|_2(\cos\theta\,\boldsymbol{u}_\theta + \sin\theta\,\boldsymbol{u}_{\theta\perp}).$$

By linearity, the LHS is $\mathbb{E}[h(U,V)U]\boldsymbol{u}_\theta + \mathbb{E}[h(U,V)V]\boldsymbol{u}_{\theta\perp}$. To isolate the scalar constraints, we take the inner product of this vector equation with the orthonormal basis vectors $\boldsymbol{u}_\theta$ and $\boldsymbol{u}_{\theta\perp}$:

$$\text{Coefficient of } \boldsymbol{u}_\theta: \quad \mathbb{E}[h(U,V)U] = \sigma^2\|\mathbf{G}\|_2\cos\theta, \tag{35}$$

$$\text{Coefficient. of } \boldsymbol{u}_{\theta\perp}: \quad \mathbb{E}[h(U,V)V] = \sigma^2\|\mathbf{G}\|_2\sin\theta. \tag{36}$$

- **Scalar Constraints:** The variance and boundedness constraints translate directly:

$$\mathbb{E}[h(U,V)^2] \leq C, \qquad -M - E \leq h(U,V) \leq M - E$$

**Step 4 (Variance decomposition: misalignment wastes variance budget).** Let $h(U, V)$ be any feasible scalar function from Step 3. We decompose $h$ into its conditional expectation given $U$ and a residual:

$$l(U) \triangleq \mathbb{E}[h(U, V) \mid U], \qquad r(U, V) \triangleq h(U, V) - l(U).$$

By construction, $\mathbb{E}[r(U, V) \mid U] = 0$ and $h(U, V) = l(U) + r(U, V)$. We analyze how this decomposition interacts with the objective and constraints:

- **Objective depends only on $g$:** Since $A(U)$ is a function of $U$ alone, the tower property of conditional expectation implies:

$$\begin{aligned} \mathbb{E}[h(U, V)A(U)] &= \mathbb{E}\big[(l(U) + r(U, V))A(U)\big] \\ &= \mathbb{E}[l(U)A(U)] + \mathbb{E}\big[A(U)\mathbb{E}[r(U, V) \mid U]\big] \\ &= \mathbb{E}[l(U)A(U)]. \end{aligned} \qquad (37)$$

Thus, the residual $r(U, V)$ contributes nothing to the maximization objective.

- **Gradient constraint in $\boldsymbol{u}_\theta$ direction depends only on $g$:** Substituting $h(U, V) = l(U) + r(U, V)$ into the constraint (35):

$$\mathbb{E}[h(U, V)U] = \mathbb{E}[l(U)U] + \mathbb{E}\big[U\mathbb{E}[r \mid U]\big] = \mathbb{E}[l(U)U].$$

Therefore, the first constraint becomes $\mathbb{E}[l(U)U] = \sigma^2 \|\mathbf{G}\|_2 \cos\theta$.

- **Gradient constraint in $\boldsymbol{u}_{\theta\perp}$ direction depends only on $r(U, V)$:** Since $U$ and $V$ are independent, $l(U)$ is independent of $V$. Thus, $\mathbb{E}[l(U)V] = \mathbb{E}[l(U)]\mathbb{E}[V] = 0$. Substituting into (36):

$$\mathbb{E}[h(U, V)V] = \mathbb{E}[l(U)V] + \mathbb{E}[r(U, V)V] = \mathbb{E}[r(U, V)V].$$

Therefore, the second constraint forces the residual to satisfy $\mathbb{E}[r(U, V)V] = \sigma^2 \|\mathbf{G}\|_2 \sin\theta$.

- **Variance Cost of Misalignment:** We calculate the variance of $h$. The cross-term between $l(U)$ and $r(U, V)$ vanishes because $r$ has zero mean conditional on $U$:

$$\mathbb{E}[h(U, V)^2] = \mathbb{E}[(l(U) + r(U, V))^2] = \mathbb{E}[l(U)^2] + \mathbb{E}[r(U, V)^2] + 2\mathbb{E}[l(U)\mathbb{E}[r(U, V) \mid U]] = \mathbb{E}[l(U)^2] + \mathbb{E}[r(U, V)^2].$$

Since $h$ is feasible, $\mathbb{E}[l(U)^2] + \mathbb{E}[r(U, V)^2] \leq C$. We lower-bound the variance of the residual $\mathbb{E}[r(U, V)^2]$ using the Cauchy-Schwarz inequality applied to the random variables $r(U, V)$ and $V$:

$$\big(\mathbb{E}[r(U, V)V]\big)^2 \leq \mathbb{E}[r(U, V)^2]\mathbb{E}[V^2].$$

We know $\mathbb{E}[r(U, V)V] = \sigma^2 \|\mathbf{G}\|_2 \sin\theta$ (from above) and $\mathbb{E}[V^2] = \sigma^2$ (variance of the Gaussian coordinate). Thus:

$$\mathbb{E}[r(U, V)^2] \geq \frac{(\sigma^2 \|\mathbf{G}\|_2 \sin\theta)^2}{\sigma^2} = \frac{(\sigma^2 \|\mathbf{G}\|_2)^2 \sin^2\theta}{\sigma^2} = \sigma^2 \|\mathbf{G}\|_2^2 \sin^2\theta.$$

- **Boundedness Constraint:** Recall the bounded constraints:

$$-M - E \leq h(U, V) \leq M - E$$

By the monotonicity of conditional expectation, these bounds are preserved for $l(U) = \mathbb{E}[h \mid U]$:

$$\mathbb{E}[-M - g(\boldsymbol{z}) \mid U] \leq \mathbb{E}[h(U, V) \mid U] \leq \mathbb{E}[M - g(\boldsymbol{z}) \mid U].$$

Since $M$ and $g(\boldsymbol{z})$ are constants:

$$-M - g(\boldsymbol{z}) \leq l(U) \leq M - g(\boldsymbol{z})$$

Combining these results, any feasible $h(U, V)$ induces a 1D function $l(U)$ that preserves the objective value but satisfies a strictly tighter variance budget if $\theta \neq 0$:

$$\mathbb{E}[l(U)U] = \sigma^2 \|\mathbf{G}\|_2 \cos\theta, \qquad (38)$$

$$\mathbb{E}[l(U)^2] \leq C - \sigma^2 \|\mathbf{G}\|_2^2 \sin^2\theta, \qquad (39)$$

$$-M - g(\boldsymbol{z}) \leq l(U) \leq M - g(\boldsymbol{z}). \qquad (40)$$

**Conclusion of Step 4 (Optimality of Alignment).** Comparing the misaligned case ($\theta \neq 0$) to the aligned case ($\theta = 0$) reveals that misalignment is strictly suboptimal. It reduces the available variance budget by $\sigma^2 \|\mathbf{G}\|_2^2 \sin^2 \theta$, shrinking the feasible set of an active constraint, while simultaneously forcing the correlation $\mathbb{E}[l(U)U]$ to a lower target. Since the objective weight $A(U)$ is strictly increasing, the optimizer benefits from maximal correlation and the full variance budget. Therefore, the optimal value is achieved at $\theta = 0$, reducing the $d$-dimensional problem to a lossless 1D optimization.

□

## H.1. Proofs for Section 5.3: Bounded Function Value

Theorem H.1 shows in the bounded function value setting with variance, gradient, and expected value constraints, the worst perturbation $\boldsymbol{\delta}$ aligns with gradient $\mathbf{G}$, we can reduce the optimization problem in $d$-dimensional $\boldsymbol{\delta}$ to an optimization problem on 1-dimensional problem input.

We build on this result. In below proof given to Proposition H.2, we write out the optimization problem of finding the worst-case perturbation in 1-dimensional variable $\alpha \in \mathbb{R}$ and a function $\phi$ of a 1-dimensional input.

**Proposition H.2.** *The $d$-dimensional optimization problem*

$$\max_{f, \boldsymbol{\delta}} \quad \Delta(\boldsymbol{\delta}) \triangleq \mathbb{E}_{p_{z+\delta}}[f(\boldsymbol{x})] - \mathbb{E}_{p_z}[f(\boldsymbol{x})]$$

$$s.t. \quad \|\boldsymbol{\delta}\|_2 = R,$$

$$(E) \quad \mathbb{E}_{p_z}[f(\boldsymbol{x})] = E,$$

$$(C) \quad \mathbb{E}_{p_z}[(f(\boldsymbol{x}) - E)^2] \leq C,$$

$$(G) \quad \mathbb{E}_{p_z}\left[f(\boldsymbol{x})\frac{\boldsymbol{x} - \boldsymbol{z}}{\sigma^2}\right] = \mathbf{G},$$

$$(M) \quad -M \leq f(\boldsymbol{x}) \leq M \quad a.s.$$

*is equivalent to the following 1-dimensional optimization problem over a **centered** scalar function $\phi : \mathbb{R} \to \mathbb{R}$ and a scalar shift $\alpha$:*

$$\max_{\alpha \in \{-R, R\}, \phi} \quad \mathbb{E}_{T+\alpha}[\phi(T)] - \mathbb{E}_T[\phi(T)]$$

$$s.t. \quad \mathbb{E}_T[\phi(T)^2] \leq C$$

$$\mathbb{E}_T[\phi(T)] = 0$$

$$\mathbb{E}_T[\phi(T) \cdot T] = \sigma^2 \|\mathbf{G}\|_2$$

$$-M - E \leq \phi(t) \leq M - E \quad \forall t \in \mathbb{R}$$

*where $T \sim \mathcal{N}(0, \sigma^2)$ is the projection of the input along the gradient direction.*

*Proof.* We utilize the alignment result from Theorem H.1 and project the constraints to the gradient direction.

**Step 1: Alignment and Projection.** By Theorem H.1, the optimal perturbation $\boldsymbol{\delta}$ must align with the gradient $\mathbf{G}$. Let $\boldsymbol{u} \triangleq \mathbf{G}/\|\mathbf{G}\|_2$. We can thus restrict the perturbation to $\boldsymbol{\delta} = \alpha \boldsymbol{u}$ with $\alpha \in \{-R, R\}$. Define the 1-D coordinate $T \triangleq \boldsymbol{u}^\top(\boldsymbol{x} - \boldsymbol{z})$. Since $\boldsymbol{x} \sim \mathcal{N}(\boldsymbol{z}, \sigma^2 I)$, the projection follows $T \sim \mathcal{N}(0, \sigma^2)$. Let $\tilde{f}(\boldsymbol{x}) \triangleq f(\boldsymbol{x}) - E$ be the centered function. As shown in centering step in proof of Theorem H.1, the optimization problem on $f$ can be reduced to an equivalent problem with $\tilde{f}$.

**Step 2: Reduction of Constraints.** For any feasible $\tilde{f}$, define the scalar function $\phi(t) \triangleq \mathbb{E}[\tilde{f}(\boldsymbol{x}) \mid T = t]$. We verify that $\phi$ satisfies the 1-D constraints:

- **Mean and Bounds:** $\mathbb{E}[\phi(T)] = \mathbb{E}[\tilde{f}(\boldsymbol{x})] = 0$. Similarly, since $-M - E \leq \tilde{f}(\boldsymbol{x}) \leq M - E$, the conditional expectation $\phi(t)$ satisfies the same bounds.

- **Variance:** By the conditional Jensen's inequality,

$$\mathbb{E}[\phi(T)^2] = \mathbb{E}[(\mathbb{E}[\tilde{f}(\boldsymbol{x}) \mid T])^2] \leq \mathbb{E}[\mathbb{E}[\tilde{f}(\boldsymbol{x})^2 \mid T]] = \mathbb{E}[\tilde{f}(\boldsymbol{x})^2] \leq C.$$

- **Gradient:** The gradient constraint $\mathbb{E}[\tilde{f}(\boldsymbol{x})(\boldsymbol{x} - \boldsymbol{z})] = \sigma^2 \mathbf{G}$ projects onto the direction $\boldsymbol{u}$ as: $\boldsymbol{u}^\top \mathbb{E}[\tilde{f}(\boldsymbol{x})(\boldsymbol{x} - \boldsymbol{z})] = \boldsymbol{u}^\top(\sigma^2 \mathbf{G})$ which means $\mathbb{E}[\tilde{f}(\boldsymbol{x})T] = \sigma^2 \|\mathbf{G}\|_2$. Using the tower property, $\mathbb{E}[\tilde{f}(\boldsymbol{x})T] = \mathbb{E}[\mathbb{E}[\tilde{f} \mid T]T] = \mathbb{E}[\phi(T)T]$. Thus, $\mathbb{E}[\phi(T)T] = \sigma^2 \|\mathbf{G}\|_2$.

**Step 3: Reduction of the Objective.** We rewrite the objective difference as a single expectation under the nominal distribution $p_{\boldsymbol{z}}$ via the likelihood ratio:

$$\Delta(\boldsymbol{\delta}) = \mathbb{E}_{p_{\boldsymbol{z}}} \left[ \tilde{f}(\boldsymbol{x}) \left( \frac{p_{\boldsymbol{z}+\boldsymbol{\delta}}(\boldsymbol{x})}{p_{\boldsymbol{z}}(\boldsymbol{x})} - 1 \right) \right].$$

With the alignment $\boldsymbol{\delta} = \alpha \boldsymbol{u}$, the density ratio depends only on the projection $T$:

$$\frac{p_{\boldsymbol{z}+\boldsymbol{\delta}}(\boldsymbol{x})}{p_{\boldsymbol{z}}(\boldsymbol{x})} = \frac{\exp\left( -\frac{\|\boldsymbol{x}-(\boldsymbol{z}+\alpha \boldsymbol{u})\|_2^2}{2\sigma^2} \right)}{\exp\left( -\frac{\|\boldsymbol{x}-\boldsymbol{z}\|_2^2}{2\sigma^2} \right)} = \exp\left( \frac{2\alpha T - \alpha^2}{2\sigma^2} \right).$$

Let $A(T) \triangleq \exp(\frac{2\alpha T - \alpha^2}{2\sigma^2}) - 1$. Using the tower property, the objective becomes:

$$\Delta(\boldsymbol{\delta}) = \mathbb{E}[\tilde{f}(\boldsymbol{x})A(T)] = \mathbb{E}_T[\phi(T)A(T)].$$

Finally, recognizing that $A(T)+1$ is exactly the likelihood ratio $p_\alpha(T)/p_0(T)$ between $\mathcal{N}(\alpha, \sigma^2)$ and $\mathcal{N}(0, \sigma^2)$, we recover the shift form:

$$\mathbb{E}_T[\phi(T)A(T)] = \mathbb{E}_T\left[ \phi(T)\left( \frac{p_\alpha(T)}{p_0(T)} - 1 \right) \right] = \mathbb{E}_{T+\alpha}[\phi(T)] - \mathbb{E}_T[\phi(T)].$$

This confirms the equivalence of the optimization problems. $\qquad \square$

## H.2. General Properties of the Optimal Solution

Before deriving the explicit functional form of the solution, we establish a critical property regarding the variance constraint. Specifically, we show that for the class of functions relevant to neural networks (continuous or smooth functions), the variance constraint is strictly active ($\lambda > 0$).

**Proposition H.3** (Necessity of Bang-Bang Structure). *Let $f : \mathbb{R}^d \to \mathbb{R}$ be a measurable function satisfying the boundedness constraint $|f(\mathbf{x})| \leq M$ for all $\mathbf{x} \in \mathbb{R}^d$. Let $\phi(t)$ be the 1-D reduced function obtained by marginalizing $f$ along the aligned direction $\mathbf{u} = \boldsymbol{\delta}/\|\boldsymbol{\delta}\|_2$. Specifically, let $\mathbf{e} = \mathbf{x} - \mathbf{z}$ be the noise vector, and let $T = \mathbf{u}^\top \mathbf{e}$ be the projection of the noise along $\mathbf{u}$. The reduced function is defined as:*

$$\phi(t) = \mathbb{E}[f(\mathbf{x}) \mid T = t].$$

*If the reduced function $\phi(t)$ is a "Bang-Bang" function (i.e., $|\phi(t)| = M$ almost everywhere), then the original high-dimensional function $f(\mathbf{x})$ must also be "Bang-Bang" (i.e., $|f(\mathbf{x})| = M$ almost everywhere).*

*Consequently, taking the contrapositive: if $f(\mathbf{x})$ takes values strictly in the interior $(-M, M)$ on a set of non-zero measure (e.g., if $f$ is continuous and transitions between bounds), its 1-D reduction $\phi(t)$ cannot be Bang-Bang.*

*Remark* H.4. In this proposition, we define a "Bang-Bang" function as any measurable function satisfying the condition $|f(\mathbf{x})| = M$ almost everywhere. This definition is broader than the class of step functions typically considered in randomized smoothing literature, as it does not require the regions where $f = M$ and $f = -M$ to be simple geometric shapes like intervals or half-spaces.

*Proof.* We decompose the noise vector $\mathbf{e} \sim \mathcal{N}(\mathbf{0}, \sigma^2 I_d)$ into the component along $\mathbf{u}$ and the orthogonal component. Let $T = \mathbf{u}^\top \mathbf{e}$ and let $\mathbf{e}_\perp$ denote the projection of $\mathbf{e}$ onto the subspace orthogonal to $\mathbf{u}$. The input $\mathbf{x}$ can be reconstructed as:

$$\mathbf{x} = \mathbf{z} + \mathbf{e} = \mathbf{z} + T\mathbf{u} + \mathbf{e}_\perp.$$

Conditioned on $T = t$, the term $T\mathbf{u}$ is fixed, while $\mathbf{e}_\perp$ remains a centered Gaussian random vector in the $(d-1)$-dimensional orthogonal subspace, distributed as $\mathcal{N}(\mathbf{0}, \sigma^2 I_{d-1})$. Let $p_\perp(\mathbf{v})$ denote the density of $\mathbf{e}_\perp$.

The definition of $\phi(t)$ becomes:

$$\phi(t) = \mathbb{E}[f(\mathbf{z} + t\mathbf{u} + \mathbf{e}_\perp) \mid T = t] = \int f(\mathbf{z} + t\mathbf{u} + \mathbf{v}) \, p_\perp(\mathbf{v}) \, d\mathbf{v}.$$

Suppose that for a specific value $t$, the reduced function achieves the upper bound: $\phi(t) = M$.

$$\int f(\mathbf{z} + t\mathbf{u} + \mathbf{v}) \, p_\perp(\mathbf{v}) \, d\mathbf{v} = M.$$

We are given the global bound $f(\mathbf{x}) \leq M$, which implies $f(\mathbf{z} + t\mathbf{u} + \mathbf{v}) \leq M$ for all $\mathbf{v}$. Rearranging the integral:

$$\int \underbrace{(M - f(\mathbf{z} + t\mathbf{u} + \mathbf{v}))}_{\geq 0} \underbrace{p_\perp(\mathbf{v})}_{> 0} \, d\mathbf{v} = 0.$$

Since the integrand is non-negative and the density $p_\perp$ is strictly positive, the term $(M - f(\mathbf{z} + t\mathbf{u} + \mathbf{v}))$ must be zero almost everywhere with respect to the measure on $\mathbf{v}$. This implies that $f(\mathbf{x}) = M$ for almost all $\mathbf{x}$ in the hyperplane defined by $T = t$ (i.e., the set $\{\mathbf{z} + t\mathbf{u} + \mathbf{v} \mid \mathbf{v} \in \mathbb{R}_\perp^{d-1}\}$).

By symmetric logic, if $\phi(t) = -M$, then $f(\mathbf{x}) = -M$ almost everywhere on the hyperplane $T = t$. Therefore, if $\phi(t) \in \{-M, M\}$ for almost all $t \in \mathbb{R}$, then for almost all hyperplanes $T = t$, the function $f(\mathbf{x})$ is constant at the boundaries. Integrating over $t$, we conclude that $f(\mathbf{x}) \in \{-M, M\}$ for almost all $\mathbf{x} \in \mathbb{R}^d$. $\qquad\square$

**Proposition H.5** (Equivalence of Inactive Variance and Bang-Bang Form). *Consider the 1-D reduced optimization problem with objective $\Delta(\phi)$, bounds $[-M - E, M - E]$, and variance budget $C$. Let $\phi^*$ be the optimal solution and $\lambda \geq 0$ be the Lagrange multiplier associated with the variance constraint. The following statements are equivalent:*

1. *The variance constraint is inactive (i.e., $\lambda = 0$).*

2. *The optimal function $\phi^*(t)$ is a Bang-Bang function (taking values in $\{-M - E, M - E\}$ almost everywhere).*

*Proof.* Let $\mathcal{L}$ be the Lagrangian of the problem. The pointwise maximization with respect to $\phi(t)$ takes the form:

$$\phi^*(t) = \arg \max_{z \in [-M-E, M-E]} \left( z \cdot L(t) - \lambda z^2 \right)$$

where $L(t)$ is the effective linear weight function derived from the objective and the linear constraints.

**Direction (1) $\implies$ (2):** If the constraint is inactive, then $\lambda = 0$. The local objective becomes linear in $z$: $\max z \cdot L(t)$. Since $L(t)$ is composed of non-trivial functions (exponentials and polynomials), the set of points where $L(t) = 0$ has measure zero. For any $t$ where $L(t) \neq 0$, the maximum of the linear function $z \cdot L(t)$ over the closed interval is unique and lies strictly at the boundary determined by the sign of $L(t)$. Thus, $\phi^*(t)$ is Bang-Bang almost everywhere.

**Direction (2) $\implies$ (1):** We prove the contrapositive: If $\lambda > 0$, then $\phi^*$ is not Bang-Bang. If $\lambda > 0$, the local objective $z \cdot L(t) - \lambda z^2$ is a strictly concave parabola with its unconstrained peak at $z^* = L(t)/2\lambda$. Since the functions defining $L(t)$ (likelihood ratio and linear terms) are continuous, $L(t)$ is continuous. By the Intermediate Value Theorem, the continuous function $L(t)/2\lambda$ must take values strictly inside the bounds $(-M - E, M - E)$ on a set of non-zero measure. In these regions, the optimal solution is $\phi^*(t) = L(t)/2\lambda$ (clipped only when it exceeds bounds), which varies smoothly and is not Bang-Bang. Thus, $\lambda > 0$ implies the function has interior values. By contraposition, if the function is Bang-Bang almost everywhere, we must have $\lambda = 0$. $\qquad\square$

**Corollary H.6** (Necessity of Active Variance Constraint). *Let $f : \mathbb{R}^d \to \mathbb{R}$ be the underlying high-dimensional function. If $f$ is not a "Bang-Bang" function (i.e., if there exists a set of non-zero measure where $|f(\mathbf{x})| < M$), then the variance constraint in the corresponding 1-D optimization problem must be **active** ($\lambda > 0$).*

*Proof.* This result follows directly by combining the contrapositives of Proposition H.3 and Proposition H.5. First, by the contrapositive of Proposition H.3, since $f(\mathbf{x})$ is not Bang-Bang almost everywhere, its 1-D reduction $\phi^*(t)$ is also not Bang-Bang. Second, by the contrapositive of Proposition H.5 (Direction 2 $\implies$ 1), since $\phi^*(t)$ is not Bang-Bang, the Lagrange multiplier for the variance constraint cannot be zero ($\lambda \neq 0$). Since $\lambda \geq 0$ by dual feasibility, we conclude $\lambda > 0$. $\qquad\square$

*Remark* H.7. Since standard neural networks define continuous functions that transition smoothly between bounds, they satisfy the condition of not being Bang-Bang almost everywhere. Thus, for neural network certification, the active-variance regime ($\lambda > 0$) is the theoretically correct model.

### H.3. The Lagrangian Formulation and Optimal Form

**Corollary 5.6.** *[Optimal Function Form] For a fixed shift magnitude $\alpha$ (where $\alpha = R$ in the worst case), the optimal univariate function $\phi^*(t)$ is a clipped affine transformation of the likelihood ratio:*

$$\phi^*(t) = \text{clip}_{[-M-E, M-E]}\left(\frac{w(t) - \mu t - \nu}{2\lambda}\right) \tag{9}$$

*where $w(t) = \exp(\frac{\alpha t}{\sigma^2} - \frac{\alpha^2}{2\sigma^2}) - 1$ denotes the centered likelihood ratio, and $\lambda, \mu, \nu$ are the Lagrange multipliers for the variance, gradient, and mean constraints, respectively.*

*Proof.* We solve the optimization problem for the function $\phi$ given a fixed shift $\alpha$. Once the optimal $\phi_\alpha^*$ is found, we maximize over $\alpha \in \{-R, R\}$.

We adopt the notation for the 1-D Gaussian densities: $p_0(t) \triangleq \mathcal{N}(0, \sigma^2)$ and $p_\alpha(t) \triangleq \mathcal{N}(\alpha, \sigma^2)$.

The primal optimization problem is:

$$\max_\phi \quad \int \phi(t)(p_\alpha(t) - p_0(t))\, dt$$

$$\text{s.t.} \quad \int \phi(t)^2 p_0(t)\, dt \leq C \quad \text{(Variance)}$$

$$\int \phi(t) t p_0(t)\, dt = \sigma^2 \|\mathbf{G}\|_2 \quad \text{(Gradient)}$$

$$\int \phi(t) p_0(t)\, dt = 0 \quad \text{(Mean)}$$

$$-M - E \leq \phi(t) \leq M - E \quad \forall t \in \mathbb{R} \quad \text{(Boundedness)}$$

For brevity, let $K \triangleq \sigma^2 \|\mathbf{G}\|_2$ and define the centered bounds $M_{upper} \triangleq M - E$ and $M_{lower} \triangleq -M - E$.

We construct the Lagrangian $\mathcal{L}$ with Lagrange multipliers $\lambda$ (variance), $\mu$ (gradient), $\nu$ (mean), $\eta_1(t)$ (upper bound), and $\eta_2(t)$ (lower bound).

$$\mathcal{L}(\phi, \lambda, \mu, \nu, \eta_1, \eta_2) = \int \phi(t)(p_\alpha(t) - p_0(t))\, dt$$

$$- \lambda\left(\int \phi(t)^2 p_0(t)\, dt - C\right)$$

$$- \mu\left(\int \phi(t) t p_0(t)\, dt - K\right)$$

$$- \nu\left(\int \phi(t) p_0(t)\, dt - 0\right)$$

$$- \int \eta_1(t)(\phi(t) - M_{upper}) p_0(t)\, dt$$

$$- \int \eta_2(t)(-\phi(t) + M_{lower}) p_0(t)\, dt$$

Note that we treat the pointwise multipliers $\eta_1, \eta_2$ as scaled by the density $p_0(t)$ (which has full support) to simplify the integral grouping. Grouping terms, the Lagrangian becomes:

$$\mathcal{L} = \int \left[\phi(t)w(t) - \lambda\phi(t)^2 - \mu\phi(t)t - \nu\phi(t) - \eta_1(t)(\phi(t) - M_{upper}) + \eta_2(t)(\phi(t) - M_{lower})\right] p_0(t)\, dt + \text{const}$$

where the shifted likelihood ratio is $w(t) \triangleq \frac{p_\alpha(t)}{p_0(t)} - 1 = \exp\left(\frac{\alpha t}{\sigma^2} - \frac{\alpha^2}{2\sigma^2}\right) - 1$.

OPTIMALITY CONDITIONS

**1. Stationarity Condition**
Taking the functional derivative of $\mathcal{L}$ with respect to $\phi(t)$ and setting it to zero gives the necessary condition for the optimal solution $\phi^*$:

$$\left.\frac{\delta\mathcal{L}}{\delta\phi(t)}\right|_{\phi^*} = (w(t) - 2\lambda\phi^*(t) - \mu t - \nu - \eta_1(t) + \eta_2(t))\, p_0(t) = 0$$

Since $p_0(t) > 0$ everywhere, the term in brackets must vanish. Rearranging for $\phi^*(t)$:

$$2\lambda\phi^*(t) = w(t) - \mu t - \nu - \eta_1(t) + \eta_2(t) \tag{41}$$

**2. KKT Conditions**
For $\phi^*(t)$ to be optimal, it must satisfy the Karush-Kuhn-Tucker conditions for all $t$:

- **Primal Feasibility:** $M_{lower} \le \phi^*(t) \le M_{upper}$.

- **Dual Feasibility:** $\lambda \ge 0$, $\eta_1(t) \ge 0$, and $\eta_2(t) \ge 0$.

- **Complementary Slackness:**

$$\lambda(\mathbb{E}[\phi^2] - C) = 0, \quad \eta_1(t)(\phi^*(t) - M_{upper}) = 0, \quad \eta_2(t)(-\phi^*(t) + M_{lower}) = 0$$

DERIVING THE OPTIMAL FORM

We assume the variance constraint is active ($\lambda > 0$). This is justified by Corollary H.6, which establishes that for any function $f$ that takes values strictly inside the bounds on a set of non-zero measure (such as a continuous neural network), the optimal solution must satisfy the variance constraint with equality. We define the *unconstrained candidate function* $h(t)$ based on the multipliers:

$$h(t; \lambda, \mu, \nu) \triangleq \frac{w(t) - \mu t - \nu}{2\lambda}$$

We determine $\phi^*(t)$ by considering three cases based on the value of $h(t)$:

**Case 1: Inactive Bounds ($M_{lower} < h(t) < M_{upper}$)**
If the candidate solution lies strictly within the bounds, complementarity implies $\eta_1(t) = 0$ and $\eta_2(t) = 0$. Substituting these into Eq. (41):

$$2\lambda\phi^*(t) = w(t) - \mu t - \nu \implies \phi^*(t) = h(t)$$

**Case 2: Upper Bound Active ($h(t) \ge M_{upper}$)**
The solution is clamped to the upper bound: $\phi^*(t) = M_{upper}$. From complementary slackness, $\eta_2(t) = 0$. Solving Eq. (41) for $\eta_1(t)$:

$$\eta_1(t) = (w(t) - \mu t - \nu) - 2\lambda M_{upper} = 2\lambda(h(t) - M_{upper})$$

Since $h(t) \ge M_{upper}$ and $\lambda > 0$, we have $\eta_1(t) \ge 0$, satisfying dual feasibility.

**Case 3: Lower Bound Active ($h(t) \le M_{lower}$)**
The solution is clamped to the lower bound: $\phi^*(t) = M_{lower}$. From complementary slackness, $\eta_1(t) = 0$. Solving Eq. (41) for $\eta_2(t)$:

$$\eta_2(t) = 2\lambda M_{lower} - (w(t) - \mu t - \nu) = 2\lambda(M_{lower} - h(t))$$

(Note: From slackness condition $\eta_2(-\phi + M_{lower}) = 0$, if $\phi = M_{lower}$, the constraint term is zero. We check the sign of $\eta_2$). Wait, substituting back into stationarity: $w - \mu t - \nu + \eta_2 = 2\lambda\phi = 2\lambda M_{lower}$. So $\eta_2 = 2\lambda M_{lower} - (w - \mu t - \nu) = 2\lambda(M_{lower} - h(t))$. Since $h(t) \le M_{lower}$ implies $M_{lower} - h(t) \ge 0$, and $\lambda > 0$, we have $\eta_2(t) \ge 0$, satisfying dual feasibility.

**Conclusion: The Clipped Form**
Combining these cases, the optimal function $\phi^*(t)$ is the candidate function $h(t)$ clipped to the interval $[M_{lower}, M_{upper}]$:

$$\phi^*(t) = \text{clip}_{[-M-E,\, M-E]}\left(\frac{w(t) - \mu t - \nu}{2\lambda}\right)$$

$\square$

### H.4. Numerical Procedure

### Solving for the Lagrange Multipliers

With the analytic form of $\phi^*(t)$ established in Corollary 5.6, we must determine the values of the Lagrange multipliers $\lambda, \mu,$ and $\nu$ that satisfy the original constraints. We treat the multipliers as variables in a dual optimization problem.

**Monotonicity of the Bounded Problem** Before detailing the solver, we verify that the worst-case harm is monotonic in the radius $R$, which validates the use of bisection search.

Let $W(R)$ denote the worst-case function value change given a perturbation budget $R$:

$$W(R) \triangleq \max_{\|\boldsymbol{\delta}\|_2 \leq R} \Delta(\boldsymbol{\delta}).$$

Using our 1-D reduction, this is equivalent to:

$$W(R) = \max_{\alpha \in \{-R, R\}} \Delta(\alpha).$$

Observe that the set of feasible perturbations grows with $R$. If $R_1 < R_2$, then any perturbation valid for $R_1$ is also valid for $R_2$ (conceptually, though our reduction fixes $\|\boldsymbol{\delta}\| = R$, the relaxation to inequality $\|\boldsymbol{\delta}\| \leq R$ yields the same optimum due to convexity/concavity arguments in the radius). Maximizing over a larger set yields a value greater than or equal to maximizing over a subset. Thus, $W(R_2) \geq W(R_1)$. This guarantee ensures that a unique solution (or a single transition point) exists for $W(R) = \epsilon$, making bisection search valid.

**Lagrange Dual Method** The dual function $g(\lambda, \mu, \nu)$ is derived by maximizing the Lagrangian with respect to $\phi$ for fixed multipliers:

$$g(\lambda, \mu, \nu) = \max_{\phi: -M-E \leq \phi \leq M-E} \mathcal{L}(\phi, \lambda, \mu, \nu)$$

Recall the Lagrangian from Appendix H.3 (excluding pointwise multipliers which are handled implicitly by the maximization):

$$\mathcal{L} = \int \phi(t)w(t)p_0(t)\,dt - \lambda \left( \int \phi(t)^2 p_0(t)\,dt - C \right)$$
$$- \mu \left( \int \phi(t)t p_0(t)\,dt - \sigma^2\|\mathbf{G}\|_2 \right) - \nu \left( \int \phi(t)p_0(t)\,dt - 0 \right)$$

Collecting constant terms and grouping the integrals, we express the dual function as:

$$g(\lambda, \mu, \nu) = \underbrace{\int \left( \phi^*(t)[w(t) - \mu t - \nu] - \lambda\phi^*(t)^2 \right) p_0(t)\,dt}_{\text{Optimized Integral Part}} + \underbrace{\lambda C + \mu\sigma^2\|\mathbf{G}\|_2}_{\text{Constant Parts}}$$

where $\phi^*(t)$ is the optimal clipped form dependent on $(\lambda, \mu, \nu)$. We minimize this dual function to find the optimal multipliers:

$$\min_{\lambda > 0, \mu \in \mathbb{R}, \nu \in \mathbb{R}} g(\lambda, \mu, \nu)$$

**Gradients of the Dual** By the Envelope Theorem, the gradient of the dual function with respect to the multipliers is exactly the negative of the constraint residuals. Since the dual function is convex, we can use these gradients for optimization (e.g., via Gradient Descent):

$$\frac{\partial g}{\partial \lambda} = C - \int \phi^*(t)^2 p_0(t)\,dt$$
$$\frac{\partial g}{\partial \mu} = \sigma^2\|\mathbf{G}\|_2 - \int \phi^*(t)t p_0(t)\,dt$$
$$\frac{\partial g}{\partial \nu} = 0 - \int \phi^*(t)p_0(t)\,dt$$

---

**Algorithm 3** Calculating the Certified Radius (Bounded + Mean, Variance and Gradient Constraints)

---

1: **function** ComputeRadius($C, E, \|\mathbf{G}\|, \epsilon, \sigma, M, r_{\text{high}}, \text{tol}$)
2:    **Input:** Variance $C$, Mean $E$, Gradient $\|\mathbf{G}\|$, threshold $\epsilon$, bounds $[-M, M]$.
3:    **Output:** The certified radius $R$.
4:    Define centered bounds: $B_{up} \leftarrow M - E$, $B_{lo} \leftarrow -M - E$.
5:    **function** WorstHarmBounded($r$)
6:       ▷ Step 1: Solve for multipliers assuming shift $\alpha = r$
7:       $w(t) \leftarrow \exp(rt/\sigma^2 - r^2/2\sigma^2) - 1$
8:       $(\lambda^*, \mu^*, \nu^*) \leftarrow$ Run Algorithm 4 with input $w(t)$
9:       ▷ Step 2: Compute worst-case change $\Delta(r)$
10:      $\phi^*(t) \leftarrow \text{clip}_{[B_{lo}, B_{up}]}\left(\frac{w(t) - \mu^* t - \nu^*}{2\lambda^*}\right)$
11:      $\Delta \leftarrow \mathbb{E}_T[\phi^*(T) \cdot w(T)]$
12:      **return** $\Delta$
    ▷ Bisection search for $R$
13:    $r_{\text{low}} \leftarrow 0, \quad r_{\text{high}} \leftarrow r_{\text{high}}$
14:    **while** $(r_{\text{high}} - r_{\text{low}}) > \text{tol}$ **do**
15:      $r_{\text{mid}} \leftarrow r_{\text{low}} + (r_{\text{high}} - r_{\text{low}})/2$
16:      $val \leftarrow$ WorstHarmBounded($r_{\text{mid}}$)
17:      **if** $val < \epsilon$ **then**
18:        $r_{\text{low}} \leftarrow r_{\text{mid}}$
19:      **else**
20:        $r_{\text{high}} \leftarrow r_{\text{mid}}$
21:    **return** $R \leftarrow r_{\text{low}}$

---

**Algorithm 4** Dual Optimization for Multipliers (3 Variables)

---

**Require:** Shift $\alpha$, Centered Bounds $B_{lo}, B_{up}$, Constants $C, \sigma, \|\mathbf{G}\|_2$, Learning rate $\gamma$.
1: **Initialize** $\lambda > 0$, $\mu = 0$, $\nu = 0$.
2: Define $w(t) \triangleq \exp(\frac{\alpha t}{\sigma^2} - \frac{\alpha^2}{2\sigma^2}) - 1$.
3: **while** not converged **do**
4:    **1. Construct Current Unconstrained Form:**
5:      $h(t) \leftarrow \frac{w(t) - \mu t - \nu}{2\lambda}$
6:    **2. Identify Active Regions (Root Finding):**
7:      Find roots of $h(t) = B_{up}$ and $h(t) = B_{lo}$.
8:      Sort roots to form intervals $I_1, I_2, \ldots, I_K$ partitioning $\mathbb{R}$.
9:      Identify type of each interval: *Upper* ($\phi^* = B_{up}$), *Lower* ($\phi^* = B_{lo}$), or *Free* ($\phi^* = h(t)$).
10:    **3. Compute Constraint Integrals (Moments):**
11:      $V_{val} \leftarrow \sum_k \int_{I_k} (\phi^*(t))^2 p_0(t)\, dt$ ▷ Variance term
12:      $G_{val} \leftarrow \sum_k \int_{I_k} \phi^*(t) t p_0(t)\, dt$ ▷ Gradient term
13:      $M_{val} \leftarrow \sum_k \int_{I_k} \phi^*(t) p_0(t)\, dt$ ▷ Mean term
14:    **4. Compute Gradients (Constraint Residuals):**
15:      $\nabla_\lambda \leftarrow C - V_{val}$
16:      $\nabla_\mu \leftarrow \sigma^2 \|\mathbf{G}\|_2 - G_{val}$
17:      $\nabla_\nu \leftarrow 0 - M_{val}$
18:    **5. Update Multipliers (Gradient Descent):**
19:      $\lambda \leftarrow \max(\epsilon, \lambda - \gamma \nabla_\lambda)$    *// Project to $\lambda > 0$*
20:      $\mu \leftarrow \mu - \gamma \nabla_\mu$
21:      $\nu \leftarrow \nu - \gamma \nabla_\nu$
22:    **6. Check Convergence:**
23:      **if** $|\nabla_\lambda| < \text{tol}$ **and** $|\nabla_\mu| < \text{tol}$ **and** $|\nabla_\nu| < \text{tol}$ **then** break
   **return** $(\lambda, \mu, \nu)$

---

# I. Details of Estimation

In this section, we provide the theoretical details for the estimators used in Section 6. We first establish the validity of our joint certification procedure using the union bound, then state the general asymptotic properties of U-statistics, and finally apply these results to derive the specific confidence intervals for the variance and the gradient norm.

## I.1. Joint Confidence Validity via Union Bound

To guarantee that the certified radius is valid with probability at least $1 - \alpha$, we must bound the estimation error for both the variance $C$ and the gradient norm $\|\mathbf{G}\|_2$. We use a union bound to split the failure budget, allocating $\alpha/2$ to each term. We construct two-sided confidence intervals for both parameters at the significance level $\alpha/2$ (i.e., using the critical value $z_{1-\alpha/4}$). This approach ensures validity as follows:

1. **Variance ($C$):** We calculate a two-sided interval $[C_{\text{low}}, C_{\text{high}}]$ and utilize the conservative upper bound $C_{\text{high}}$. Due to the two-sided construction, the probability that the true variance exceeds this bound is at most $\alpha/4$ (half of the $\alpha/2$ budget).

2. **Gradient Norm ($\|\mathbf{G}\|_2$):** We calculate a two-sided interval $[G_{\text{low}}, G_{\text{high}}]$. Since the certified radius is non-monotonic with respect to the gradient norm (Corollary 5.4), we compute the radius by finding the worst-case gradient norm $G^* \in [G_{\text{low}}, G_{\text{high}}]$ that minimizes the radius. The true gradient norm falls within this interval with probability $1 - \alpha/2$.

The union bound guarantees that the joint certificate holds with probability at least $1 - (\alpha/4 + \alpha/2) = 1 - 0.75\alpha \geq 1 - \alpha$.

## I.2. General Theory: Asymptotic Normality of U-Statistics

Both our variance and gradient norm estimators rely on U-statistics of order $r = 2$. We utilize the standard asymptotic theory for non-degenerate U-statistics (Hoeffding, 1992; van der Vaart, 2000).

**Theorem I.1** (Asymptotic Normality). *Let $X_1, \ldots, X_n$ be i.i.d. random variables (or vectors). Let $h(x_1, x_2)$ be a symmetric kernel of order $r = 2$, and let $\theta = \mathbb{E}[h(X_1, X_2)]$ be the parameter to be estimated. The U-statistic estimator is defined as:*

$$U_n = \binom{n}{2}^{-1} \sum_{1 \leq i < j \leq n} h(X_i, X_j).$$

*Provided that $\mathbb{E}[h(X_1, X_2)^2] < \infty$ and the asymptotic variance component $\zeta_1$ is positive, $U_n$ satisfies:*

$$\sqrt{n}(U_n - \theta) \xrightarrow{d} \mathcal{N}(0, 4\zeta_1),$$

*where $\zeta_1 = \text{Cov}(h(X_1, X_2), h(X_1, X_2')) = \text{Var}(\mathbb{E}[h(X_1, X_2) \mid X_1])$.*

To construct confidence intervals, we estimate the asymptotic variance $4\zeta_1$ using consistent sample estimates of the underlying moments.

## I.3. Variance Estimation

We estimate the local variance $C = \text{Var}(f(\boldsymbol{x}))$ where $\boldsymbol{x} \sim \mathcal{N}(\boldsymbol{z}, \sigma^2 I)$. Let $Y = f(\boldsymbol{x})$ denote the random function output. We are given $n$ i.i.d. samples $Y_1, \ldots, Y_n$. Using the identity $\text{Var}(Y) = \frac{1}{2}\mathbb{E}[(Y_1 - Y_2)^2]$, we employ the kernel $h(y_i, y_j) = \frac{1}{2}(y_i - y_j)^2$.

**Estimator equivalence.** As noted in the main text, the U-statistic with this kernel simplifies exactly to the standard unbiased sample variance:

$$\hat{\sigma}^2 = \binom{n}{2}^{-1} \sum_{i<j} \frac{1}{2}(Y_i - Y_j)^2 = \frac{1}{n-1} \sum_{i=1}^{n} (Y_i - \bar{Y})^2 = S^2.$$

**Asymptotic Variance.** We calculate $\zeta_1$ by projecting the kernel onto one variable. To distinguish from the smoothing parameter $\sigma$ and Lagrange multipliers used elsewhere, we explicitly denote the true moments of the output distribution $Y$ as

$\mu_Y = \mathbb{E}[Y]$ and $\sigma_Y^2 = \mathrm{Var}(Y)$. The projection is:

$$h_1(t) = \mathbb{E}[h(t, Y_j)] = \frac{1}{2}\mathbb{E}[(t - Y_j)^2] = \frac{1}{2}((t - \mu_Y)^2 + \sigma_Y^2).$$

The variance of this projection is:

$$\zeta_1 = \mathrm{Var}(h_1(Y_i)) = \mathrm{Var}\left(\frac{1}{2}(Y_i - \mu_Y)^2\right) = \frac{1}{4}(m_4 - \sigma_Y^4),$$

where $m_4 = \mathbb{E}[(Y - \mu_Y)^4]$ is the fourth central moment of $Y$.

**Confidence Interval.** We require a high-probability *upper bound* on the variance. We estimate the asymptotic variance using the sample fourth central moment $\hat{m}_4$ and sample variance $S^2$. Consistent with our union bound strategy, we use the critical value $z_{1-\alpha/4}$ (corresponding to a two-sided interval with significance level $\alpha/2$). The upper bound is:

$$C_{upper} = S^2 + z_{1-\alpha/4}\sqrt{\frac{\hat{m}_4 - (S^2)^4}{n}}.$$

## I.4. Gradient Norm Estimation

We estimate the squared gradient norm $\theta = \|\nabla g(\boldsymbol{z})\|^2$. Recall that $\nabla g(\boldsymbol{z}) = \mathbb{E}[\mathbf{W}]$, where $\mathbf{W} = \frac{1}{\sigma^2}(\boldsymbol{x} - \boldsymbol{z})f(\boldsymbol{x})$. We are given $n$ i.i.d. samples $\mathbf{W}_1, \ldots, \mathbf{W}_n$. We use the kernel $h(\mathbf{w}_i, \mathbf{w}_j) = \mathbf{w}_i^\top \mathbf{w}_j$.

**Asymptotic Variance.** The parameter to be estimated is $\theta = \|\mathbb{E}[\mathbf{W}]\|^2$. We denote the true mean vector (the gradient) as $\mathbf{m} = \mathbb{E}[\mathbf{W}]$ and the covariance matrix as $\Sigma_{\mathbf{W}} = \mathrm{Cov}(\mathbf{W})$. The projection of the kernel is:

$$h_1(\mathbf{w}) = \mathbb{E}[\mathbf{w}^\top \mathbf{W}_j] = \mathbf{w}^\top \mathbb{E}[\mathbf{W}_j] = \mathbf{w}^\top \mathbf{m}.$$

The variance of this projection is $\zeta_1 = \mathrm{Var}(\mathbf{W}_i^\top \mathbf{m}) = \mathbf{m}^\top \Sigma_{\mathbf{W}} \mathbf{m}$. Therefore, the U-statistic $\hat{\theta}$ follows:

$$\sqrt{n}(\hat{\theta} - \theta) \xrightarrow{d} \mathcal{N}(0, 4\mathbf{m}^\top \Sigma_{\mathbf{W}} \mathbf{m}).$$

**Computational Stability in High Dimensions.** Directly estimating the $d \times d$ covariance matrix $\Sigma_{\mathbf{W}}$ to compute $\mathbf{m}^\top \Sigma_{\mathbf{W}} \mathbf{m}$ can be computationally expensive and unstable when $d$ is large. However, we observe that $\mathbf{m}^\top \Sigma_{\mathbf{W}} \mathbf{m}$ is simply the variance of the scalar projection $\mathbf{W}^\top \mathbf{m}$. We estimate this scalar variance directly using the sample variance of the projected points $\{\mathbf{W}_i^\top \hat{\mathbf{m}}\}_{i=1}^n$, where $\hat{\mathbf{m}} = \frac{1}{n}\sum \mathbf{W}_i$. This avoids explicit covariance matrix computation. Let $\hat{v}^2$ denote this estimated scalar variance. We estimate the asymptotic variance scalar $\mathbf{m}^\top \Sigma_{\mathbf{W}} \mathbf{m}$ using the sample variance of the projected points $\mathbf{W}_i^\top \hat{\mathbf{m}}$. While this estimator may have finite-sample bias due to the dependency between $\hat{\mathbf{m}}$ and $\mathbf{W}_i$, it is a consistent estimator of the true asymptotic variance. By Slutsky's theorem, consistency is sufficient to ensure the validity of the derived confidence intervals. While the estimation of the projection direction $\mathbf{m}$ itself is subject to high-dimensional noise (scaling with $d/n$), the use of a conservative upper confidence bound on the scalar projection variance ensures that the resulting gradient norm estimate remains reliable for certification purposes.

**Confidence Interval.** We construct a two-sided confidence interval for $\theta$ with significance level $\alpha/2$ using the critical value $z_{1-\alpha/4}$. Let $\hat{\sigma}_\theta = \sqrt{4\hat{v}^2/n}$. The bounds for the squared norm $\theta$ are:

$$\theta_{low} = \hat{\theta} - z_{1-\alpha/4}\hat{\sigma}_\theta, \quad \theta_{high} = \hat{\theta} + z_{1-\alpha/4}\hat{\sigma}_\theta.$$

To obtain the interval for the gradient norm $G = \sqrt{\theta}$, we take the square root of the bounds (truncating the lower bound at zero):

$$G_{low} = \sqrt{\max(0, \theta_{low})}, \quad G_{high} = \sqrt{\max(0, \theta_{high})}.$$

These bounds define the search interval $[G_{low}, G_{high}]$ used in our certification procedure.

# J. More Experiment Details

## J.1. Synthetic Functions and Ground Truth Computation

We validate our certification methods on synthetic functions where the true worst-case radius can be computed analytically or via high-precision numerical methods. This allows us to verify soundness (certified radius $\leq$ true radius) and evaluate tightness (how close certified radius is to true radius). All functions operate on 2-dimensional input space ($d = 2$), with input coordinates $x = (x_1, x_2) \in \mathbb{R}^2$.

We evaluate three functions, each designed to test different aspects of the certification method.

The **quadratic** function is defined as $f(x) = s \cdot \|x - c\|_2^2$ with center $c = (0.0, 0.0)$ and scale $s = 1.0$. This function is smooth and grows quadratically, making it ideal for testing tightness on well-behaved functions. This function has no output clipping, allowing us to test the method's behavior when outputs can grow arbitrarily large.

The **slice** function is defined as $f(x) = \max(0, x_1 - t)$ with threshold $t = 0.0$. This function has a discontinuous jump at $x_1 = t$ and grows linearly for $x_1 > t$, testing the method's robustness to non-smooth functions without explicit output bounds. The function is unbounded above but bounded below at zero.

The **sandwich** function is defined as $f(x) = \max(0, \min(1, x_1 - t))$ with threshold $t = 0.0$. This function is piecewise constant with a transition region, creating a challenging case for certification. While the function itself is bounded (outputs in $[0, 1]$), we still use the unbounded $(C, G)$ certifier (without providing an explicit bound $M$) to test the method's behavior when no output bound is supplied.

We compute the true worst-case radius $R_{\text{true}}$ using optimization-based methods. Since these functions lack analytical solutions for the worst-case radius, we use a combination of Monte Carlo expectation estimation and differential evolution optimization (Storn & Price, 1997) (implemented via `scipy.optimize.differential_evolution`). Differential evolution is a population-based global optimization algorithm that is well-suited for finding worst-case perturbations in non-convex optimization problems. For each test point $z$, we first compute the baseline expectation $g(z) = E[f(z + e)]$, $e \sim \mathcal{N}(0, \sigma^2 I)$ using Monte Carlo estimation with $50,000$ samples. We then use differential evolution to find the worst-case perturbation direction $\delta^*$ at a given radius $r$ that maximizes the change in expectation $|g(z + \delta) - g(z)|$ subject to $\|\delta\|_2 \leq r$. The true radius is found via binary search: we search for the largest radius $r$ such that the worst-case expectation change at radius $r$ (computed using $100,000$ samples for final evaluation) does not exceed the output tolerance $\epsilon$. The binary search uses 30 iterations with tolerance $10^{-4}$, and the differential evolution uses 50 iterations with population size 10. This approach provides high-precision estimates of the true worst-case radius while remaining computationally tractable.

All synthetic function experiments use consistent parameters: test points drawn uniformly from $[-1.0, 1.0] \times [-1.0, 1.0]$ using random seed 42, 10 randomly sampled test points per function, cross-validation over $\sigma \in \{0.1, 0.2, 0.5\}$ for our certifier and over both $\sigma \in \{0.1, 0.2, 0.5\}$ and $\alpha \in \{0.35, 0.49\}$ for $\alpha$-smoothing to find optimal parameters, output tolerance levels $\epsilon \in \{0.2, 0.5\}$, $N = 5,000$ samples, and $P = 0.9$ success probability for both our certifier and $\alpha$-smoothing.

## J.2. Convergence Analysis of Radius Estimators

We validate that our statistical estimators converge correctly and that certified radii converge to their theoretical values as the sample size $N$ increases. We perform this analysis on a single test sample from the MNIST rotation task, using sample sizes $N \in \{100, 500, 1000, 5000, 10000\}$ with multiple independent trials per $N$ value. Theoretical values for $C$ and $\|\mathbf{G}\|$ are estimated using large-sample Monte Carlo with $N = 50,000$ samples.

### J.2.1. PART 1: ESTIMATOR CONVERGENCE

For each $N$, we compute point estimates $\hat{C}$ and $\hat{\theta}$ using U-statistic estimators, along with upper confidence bounds (UCBs) $C_{\text{UCB}}$ and $\theta_{\text{UCB}}$ using analytical confidence intervals based on asymptotic normality (z-critical or t-distribution). Figure 3 shows that both estimators converge to their true values with $O(1/\sqrt{N})$ rate, with mean bias less than $5\%$ at $N = 10,000$ and confidence intervals shrinking proportionally.

### J.2.2. PART 2: RADIUS CONVERGENCE

We compute the empirical certified radius $R_{\text{empirical}}$ using estimated $C_{\text{UCB}}$ and $\theta_{\text{UCB}}$, and the theoretical radius $R_{\text{theoretical}}$ using the large-sample estimates. Figure 4 shows that the empirical radius converges to the theoretical radius as $N$ increases:

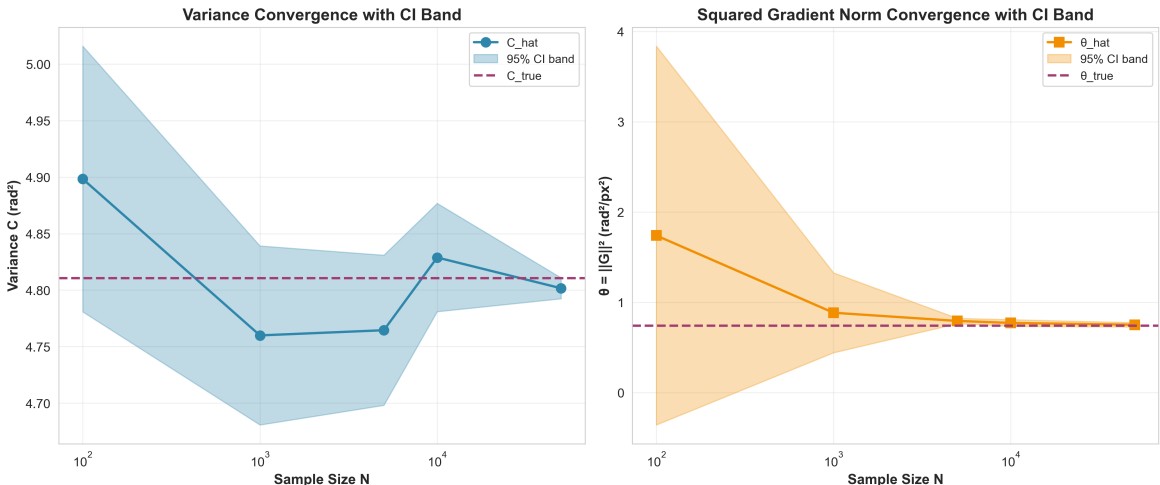

*Figure 3.* Detailed convergence analysis showing variance and squared gradient norm convergence with confidence interval bands. The plots demonstrate $O(1/\sqrt{N})$ convergence rate as expected from central limit theorem arguments, with confidence intervals shrinking proportionally as sample size $N$ increases.

at $N = 5,000$, within $10\%$ of the theoretical value, and at $N = 10,000$, within $5\%$. This validates that estimation error decreases with $N$ and that the certification method is well-calibrated.

### J.2.3. EXPERIMENTAL SETUP

We use a single MNIST test image (index 0) with parameters $\sigma = 0.5$, $\epsilon = 10°$ (0.175 radians), $M = \pi$ radians, and confidence level $95\%$ (failure probability $\alpha = 0.05$).

## J.3. MNIST Rotation Experimental Details

### J.3.1. MODEL ARCHITECTURE

We use an E(2)-equivariant convolutional neural network (E2CNN) architecture for rotation angle prediction. The model uses rotation-equivariant convolutions that preserve the E(2) symmetry group structure with $N = 8$ rotational symmetries. The architecture consists of three convolutional layers with increasing channel dimensions (16, 32, 64), followed by a final layer mapping to invariant features (128 channels). The output layer is a regression head that predicts rotation angles in the range $[-\pi, \pi]$ radians via a $(\cos\theta, \sin\theta)$ representation. The model contains approximately 50,000 parameters. Full architecture details and implementation are available in our code repository.

### J.3.2. TRAINING PROCEDURE

The model was trained on the rotated MNIST dataset using the Adam optimizer with an initial learning rate of $0.001$, which was reduced by a factor of $0.5$ every 20 epochs using a StepLR scheduler. Training was performed for 30 epochs with a batch size of 64, using mean squared error (MSE) loss on the $(\cos\theta, \sin\theta)$ representation of angles. The validation metric was circular mean absolute error (MAE) in degrees, and no additional data augmentation was applied since the dataset already contains full rotation range $[0°, 360°]$. During training, angles are converted to the range $[-180°, 180°]$ for consistency with the model's output range. The data was split into $80\%$ training and $20\%$ validation sets.

Training was performed on GPU and converged within 30 epochs. The final model achieves a test set mean absolute error (MAE) of $4.53°$ ($\pm 9.48°$ standard deviation). Training and validation losses decreased steadily, with validation MAE stabilizing around epoch 20-25 (coinciding with the learning rate reduction at epoch 20). The model demonstrates good generalization, with test set performance matching validation performance.

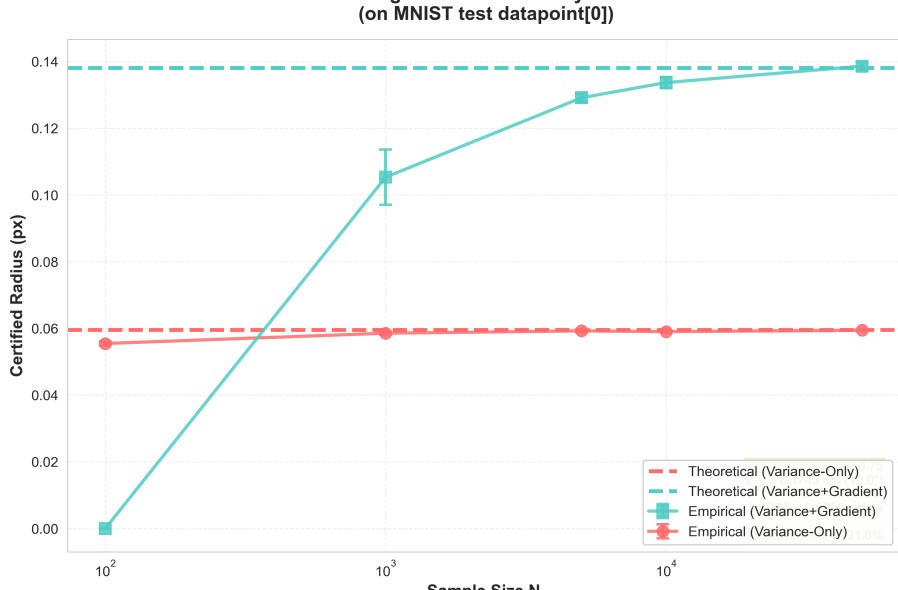

*Figure 4.* Certified radius convergence on a single MNIST test point. Shows empirical certified radii (computed with estimated statistics) converging to theoretical radii (computed using large-sample Monte Carlo estimates with $N = 50,000$) as sample size $N$ increases.

### J.3.3. TEST SET SELECTION

We evaluate on 100 stratified test samples from the rotated MNIST test set, with 10 samples per digit class (0–9). Samples were selected using stratified random sampling with seed 42 for reproducibility. The test indices are stored in the test manifest file and match the indices used in all comparison experiments to ensure fair evaluation across methods.

### J.3.4. CERTIFICATION PARAMETERS

For all certification experiments, we use a sample size of $N = 10,000$ for statistical estimation, which was validated through convergence analysis, showing stable estimates with small confidence intervals. We employ a total confidence level of $1 - \alpha$ (total failure probability $\alpha$), where different experiments may use different values of $\alpha$ depending on the desired success probability. The mean estimate $E = \mathbb{E}_{e \sim \mathcal{N}(0, \sigma^2 I)}[f(z + e)]$ is computed as a point estimate (sample mean) without a confidence interval. Numerical integration is performed using 40 Gauss-Hermite quadrature points, and the dual optimization problem is solved with a tolerance of $10^{-6}$ and a maximum of 1000 iterations. The output tolerance is set to $\epsilon = 10°$ (0.1745 radians).

**Confidence level details:** We use a total confidence level of $1 - \alpha$ (total failure probability $\alpha$). The allocation of $\alpha$ depends on the method: For methods using variance and gradient constraints $(E, C, G) + M$, we allocate $\alpha/2$ failure probability to each constraint (variance $C$ and gradient norm $\|\mathbf{G}\|$) via the union bound, ensuring the total failure probability does not exceed $\alpha$. For methods using only the variance constraint $(E, C) + M$, the entire failure probability $\alpha$ is allocated to the variance constraint $C$ (no split). For each constraint, we construct a two-sided confidence interval, which further reduces the per-tail failure probability. This ensures that with probability at least $1 - \alpha$, the relevant constraints are simultaneously satisfied within their respective confidence intervals. Note the per-sigma comparison table (Table 4) uses $\alpha = 0.05$ (95% confidence) for $(E, C, G) + M$ and $\alpha = 0.025$ (97.5% confidence) for $(E, C) + M$, while the main text comparison (Section 7.2) uses $\alpha = 0.10$ (90% confidence) for both methods. The mean estimate $E$ is computed as the sample mean point estimate without a confidence interval. This is justified because with $N = 10,000$ samples, the standard error of the mean is $\sqrt{C/N}$, which is small relative to the variance and gradient bounds that dominate the certified radius.

**Quadrature points:** We use 40 Gauss-Hermite quadrature points to numerically approximate Gaussian expectations of the form $\mathbb{E}[f(T)]$ where $T \sim \mathcal{N}(0, \sigma^2)$. Gauss-Hermite quadrature is a standard numerical integration technique that provides exact integration for polynomials up to degree $2n - 1$ using $n$ quadrature points. We use this method because the dual optimization problem requires computing expectations of functions $\phi^*(t)$ that may have discontinuities (due to clipping at

$\pm M$), making analytical integration intractable. The choice of 40 points provides a good balance between accuracy and computational efficiency, as verified through sensitivity analysis.

**Dual optimization tolerance:** The dual optimization problem is solved using iterative methods (gradient-based optimization or root finding) to find the Lagrange multipliers $(\lambda, \mu, \nu)$ that satisfy the Karush-Kuhn-Tucker (KKT) conditions. The tolerance $10^{-6}$ controls when the optimization algorithm terminates, ensuring that constraint violations (e.g., $|\mathbb{E}[\phi^2] - C|$) are below this threshold. This tolerance is chosen to be sufficiently tight to guarantee numerical accuracy while remaining computationally tractable.

### J.3.5. ALPHA-SMOOTHING PARAMETERS

For the $\alpha$-smoothing baseline comparison, we evaluate both trimming parameters $\alpha \in \{0.35, 0.49\}$ across all noise levels $\sigma \in \{0.06, 0.12, 0.25, 0.50, 0.75\}$ and report results using the $\alpha$ value that yields the best performance at each $\sigma$. In practice, we found that $\alpha = 0.49$ performs best for $\sigma = 0.06$ (the optimal $\sigma$ for $\alpha$-smoothing), while $\alpha = 0.35$ may be preferred at other $\sigma$ values. We use a probability threshold of $P = 0.9$ and allocate $n_{\text{tr}} = 10,000$ samples for $p_A$ estimation (matching the $N = 10,000$ used in our methods). For the Binomial mapping (described below), we use $n_{\text{sample}} = 500$ when $\alpha = 0.49$ or $n_{\text{sample}} = 200$ when $\alpha = 0.35$. We enable circular distance for angle predictions, and all experiments use random seed 42 for reproducibility.

**Note on sample sizes:** The parameter $n_{\text{tr}} = 10,000$ is the main Monte Carlo sample size used for estimating $p_A$ (the probability that the output remains within tolerance), which directly corresponds to the $N = 10,000$ samples used in our $(E, C) + M$ and $(E, C, G) + M$ methods. The parameter $n_{\text{sample}}$ is used for a different purpose: computing the quantile $q$ via Binomial distribution mapping, and does not affect the main statistical estimation. Therefore, the comparison is fair: all methods use 10,000 samples for the primary statistical estimation.

**Binomial mapping:** The $\alpha$-smoothing method uses a Binomial distribution mapping to convert the trimming parameter $\alpha$ and sample size $n_{\text{sample}}$ into a quantile threshold $q$ that appears in the certified radius formula. Specifically, given $\alpha$, $n_{\text{sample}}$, and target probability $P$, the method solves for $q$ such that $\text{BinomCDF}(\lfloor \alpha \cdot n_{\text{sample}} \rfloor, n_{\text{sample}}, 1 - q) = P$, where BinomCDF is the cumulative distribution function of a Binomial random variable. This mapping accounts for the fact that $\alpha$-trimming removes the $\alpha$ fraction of extreme values, and the quantile $q$ represents the probability threshold that ensures the trimmed mean remains within the acceptable region with probability at least $P$. For details, see Rekavandi et al. (2024).

**Center method and comparison with our methods:** The implementation of $\alpha$-smoothing method measures the output shift relative to the base function's clean prediction $f(z)$ rather than the smoothed regressor's output $g_\alpha(z)$. This choice is required by the theoretical guarantee of Rekavandi et al. (2024). The acceptable region is defined as $[f(z) - \epsilon, f(z) + \epsilon]$ (with circular wrapping for angles). In contrast, our $(E, C) + M$ and $(E, C, G) + M$ methods certify output shift relative to the smoothed regressor's expectation $g(z) = \mathbb{E}_{\boldsymbol{e}}[f(\boldsymbol{z} + \boldsymbol{e})]$.

### J.3.6. NOISE LEVEL SELECTION

We evaluate all methods across multiple noise levels $\sigma \in \{0.06, 0.12, 0.25, 0.50, 0.75\}$ to understand performance sensitivity. Based on mean certified radius, the optimal $\sigma$ values are: $\sigma = 0.06$ for $(E, C) + M$ (mean radius = 0.090 pixels), $\sigma = 0.75$ for $(E, C, G) + M$ (mean radius = 0.207 pixels), and $\sigma = 0.06$ with $\alpha = 0.49$ for $\alpha$-smoothing (mean radius = 0.120 pixels). The different optimal $\sigma$ values reflect the different mechanisms: methods without gradient information benefit from smaller noise (less restrictive variance bounds), while methods with gradient information can leverage larger noise levels effectively.

### J.3.7. PER-SIGMA PERFORMANCE

Table 4 shows the mean certified radius for each method across all noise levels. Our $(E, C, G) + M$ method demonstrates robust performance across all noise levels, maintaining non-zero certified radii at all $\sigma$ values. In contrast, $\alpha$-smoothing exhibits catastrophic failure at large $\sigma$ values ($\sigma \geq 0.5$), certifying zero radius for all samples. This highlights a critical limitation of methods that rely solely on probability mass estimation, which becomes unreliable when the probability mass becomes too small at large noise levels.

*Table 4.* Mean certified radius (pixels) for each method across different noise levels $\sigma$. $(E, C) + M$ uses 97.5% success probability ($\alpha = 0.025$), $(E, C, G) + M$ uses 95% success probability ($\alpha = 0.05$), and $\alpha$-smoothing uses $P = 0.9$ (90% success probability).

| Method | $\sigma = 0.06$ | $\sigma = 0.12$ | $\sigma = 0.25$ | $\sigma = 0.50$ | $\sigma = 0.75$ |
|---|---|---|---|---|---|
| $(E, C) + M$ (97.5%) | 0.090 | 0.058 | 0.045 | 0.059 | 0.064 |
| $(E, C, G) + M$ (95%) | 0.078 | 0.068 | 0.081 | 0.156 | 0.207 |
| $\alpha$-smoothing ($P = 0.9$) | 0.120 | 0.049 | 0.002 | 0.000 | 0.000 |

### J.3.8. COMPUTATIONAL DETAILS

All certification experiments were performed on **NVIDIA A100 GPUs** using batched model inference for efficient Monte Carlo sampling. For each test sample, the certification process involves: (1) statistical estimation using $N = 10,000$ Monte Carlo samples to compute variance $C$ and gradient norm $\|\mathbf{G}\|$ estimates, and (2) dual optimization to solve for the certified radius. The average wall-clock time per certificate computation is roughly 5–10 seconds on an A100 GPU, with the majority of time spent on statistical estimation (Monte Carlo sampling and U-statistic computation). The dual optimization step is relatively fast (typically under 1 second) due to the efficient numerical integration using Gauss-Hermite quadrature. For the full experimental evaluation across 100 test samples and 5 noise levels ($\sigma \in \{0.06, 0.12, 0.25, 0.50, 0.75\}$), the total compute time corresponds to roughly 1–2 GPU-hours. Parallelization across multiple test samples on the cluster further reduced the experimental turnaround time.

### J.4. Age-Estimation Experimental Details

#### J.4.1. TASK AND DATASET

The age-estimation experiment evaluates robustness for an aperiodic scalar regression task. Each input is a face image and the target is the subject's age in years. Unlike MNIST rotation, the output is not circular, so prediction error is measured by ordinary absolute error in years. We use the UTKFace dataset,[1] whose filenames encode age labels, and evaluate on 100 fixed test images. The same selected images are used for every method and every parameter setting.

#### J.4.2. MODEL AND PREPROCESSING

The base regressor is a pretrained MiVOLO-v2 face-age model (Kuprashevich & Tolstykh, 2023) used only for inference;[2]. We do not retrain or fine-tune it. The certification input is a $64 \times 64$ RGB image with pixel values scaled to $[0, 1]$. Gaussian smoothing noise is added at this resolution, so certified radii are $\ell_2$ perturbation norms in the same normalized pixel space.

Before passing a noisy image to MiVOLO-v2, we resize it to the model's $384 \times 384$ input size using bilinear interpolation: each new pixel is computed as a weighted average of the four nearest pixels in the $64 \times 64$ image. We then apply the pretrained model's channel normalization constants and run inference. This resizing and normalization step is deterministic preprocessing applied after the randomized smoothing perturbation. We use the known valid age range $[0, 116]$ as the output bound for our bounded certificates, with $M = 116$.

#### J.4.3. TEST SET SELECTION

We evaluate on 100 fixed UTKFace test images selected from the held-out split and store the selected test indices for reuse across all sweeps. This is important because the UTKFace runs were split across separate jobs for our methods and for $\alpha$-smoothing; using the same selected indices ensures that all paired comparisons are pointwise comparisons on identical images.

#### J.4.4. CERTIFICATION PARAMETERS

For our $(E, C, G) + M$ certificate, we use $N = 5,000$ Monte Carlo samples per point to estimate the mean, variance, and gradient-norm quantities. The output tolerance is $\epsilon = 6$ years. We evaluate success probability 0.9 for all methods. For $(E, C, G) + M$, the failure probability is split across the estimated constraints used by the certificate. For $(E, C) + M$, we reuse the same saved Monte Carlo estimates but recompute the variance and mean confidence bounds with the two-way

---

[1] UTKFace dataset page: `https://github.com/aicip/UTKFace`.
[2] MiVOLO-v2 model checkpoint: `https://huggingface.co/iitolstykh/mivolo_v2`

union-bound split for $(E, C)$ at success probability 0.9, avoiding another expensive model-inference sweep. The dual certificate optimization uses Gauss-Hermite quadrature as in the MNIST rotation experiment.

For each $\sigma$, we also estimate the smoothed predictor $g(x) = \mathbb{E}_\eta[f(x + \eta)]$ by Monte Carlo averaging over the same noisy model evaluations used for certification. This gives a real-valued non-adversarial accuracy measure for the smoothed model. The base regressor has clean mean absolute error 6.73 years on the 100 evaluated points; the appendix tradeoff plot reports how the smoothed MAE changes with $\sigma$.

### J.4.5. SINGLE-POINT CONVERGENCE ANALYSIS

Mirroring the MNIST convergence analysis, we validate the statistical estimators on one representative UTKFace test image. The selected image has true age 26 years and clean model prediction 26.55 years. We use $\sigma = 0.06$ and output tolerance $\epsilon = 5$ years, and compare finite-sample estimates at $N \in \{100, 500, 1000, 2000, 5000\}$ against large-sample pseudo-true values computed with $N = 10,000$. Each finite-sample setting is repeated over 10 independent trials.

The large-sample pseudo-true estimates are $g(x) = 28.24$, $C = 0.971$, $\theta = \|G\|_2^2 = 768.73$, and $\|G\|_2 = 27.73$. Figure 5 shows convergence of the variance estimator and squared-gradient estimator toward these reference values: point estimates are averaged over trials and shaded bands show the corresponding 95% confidence bands. The variance estimate is already within about 2% of the reference value across the tested sample sizes. The gradient estimator is noisier at small $N$, as expected in the high-dimensional image input space, but moves toward the pseudo-true value as $N$ increases.

Figure 5 reports confidence-interval coverage and certified-radius convergence. The nominal confidence level is 95% in this diagnostic run. Empirical coverage is at least 90% for $C$ and 100% for $\theta$ and $\|G\|_2$ across the tested sample sizes. The certified radius converges quickly to the large-sample reference radius 0.1088: the mean radius is within 1.8% at $N = 500$, within 0.8% at $N = 2000$, and within 0.5% at $N = 5000$.

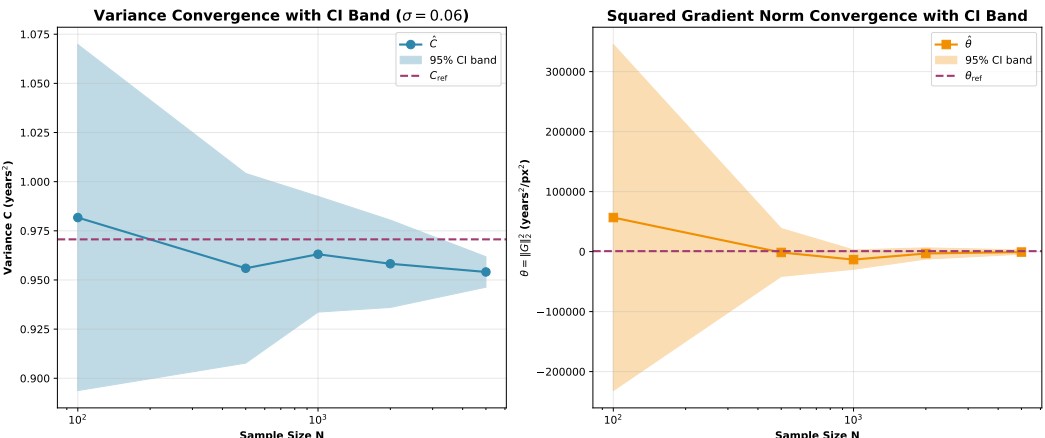

*Figure 5.* Single-point convergence analysis for the UTKFace age-estimation task. We compare finite-sample estimates to large-sample pseudo-true values on one held-out image. Panels show convergence of variance $C$, squared gradient norm $\theta = \|G\|_2^2$, empirical confidence-interval coverage, and certified radius.

### J.4.6. ALPHA-SMOOTHING PARAMETERS

For the $\alpha$-smoothing baseline, we evaluate trimming parameters $\alpha \in \{0.35, 0.49\}$ across the same noise levels $\sigma \in \{0.06, 0.12, 0.25, 0.5, 0.75\}$. We use probability threshold $P = 0.9$ and $n_{\mathrm{tr}} = 5,000$ samples to estimate $p_A$, matching the main Monte Carlo sample size used by our methods. As in the MNIST rotation experiment, the additional Binomial mapping sample size controls the quantile threshold used by $\alpha$-smoothing and is separate from the main $p_A$ estimation sample size; we use $n_{\mathrm{sample}} = 200$ for $\alpha = 0.35$ and $n_{\mathrm{sample}} = 500$ for $\alpha = 0.49$.

### J.4.7. NOISE LEVEL SELECTION AND PER-SIGMA PERFORMANCE

Each method selects the fixed grid configuration with the largest mean certified radius over the 100 test images. Table 5 reports the per-sigma mean certified radius. For $\alpha$-smoothing, the table reports the better trimming parameter at each $\sigma$;

$\alpha = 0.49$ is best at all five noise levels in this sweep. All three methods select $\sigma = 0.75$ by mean certified radius.

*Table 5.* Age-estimation mean certified radius (pixels) across noise levels. Radii are $\ell_2$ perturbation norms in normalized pixel space. For $\alpha$-smoothing, we report the better trimming parameter between $\alpha = 0.35$ and $\alpha = 0.49$ at each $\sigma$.

| Method | $\sigma = 0.06$ | $\sigma = 0.12$ | $\sigma = 0.25$ | $\sigma = 0.50$ | $\sigma = 0.75$ |
|---|---|---|---|---|---|
| $(E, C) + M$ | 0.095 | 0.168 | 0.396 | 1.054 | 1.651 |
| $(E, C, G) + M$ | 0.126 | 0.239 | 0.617 | 1.520 | 2.152 |
| $\alpha$-smoothing | 0.097 | 0.100 | 0.147 | 0.526 | 0.817 |

*Table 6.* Age-estimation best fixed-configuration comparison on 100 fixed test points. Each method selects the grid point with largest mean certified radius. Paired mean gain is the mean of $(E, C, G) + M$ minus $\alpha$-smoothing over matched points; gain/loss ratio is the sum of positive gains divided by the sum of $\alpha$-smoothing's positive gains.

| Method | Best $\sigma$ | $\alpha$ | Mean Cert. Radius |
|---|---|---|---|
| $(E, C) + M$ | 0.75 | – | 1.651 |
| $(E, C, G) + M$ | 0.75 | – | 2.152 |
| $\alpha$-smoothing | 0.75 | 0.49 | 0.817 |
| Paired mean gain | – | – | 1.335 |
| Gain/loss ratio | – | – | 24.74 |

### J.4.8. RADIUS–ACCURACY TRADEOFF AND COMPUTATION

The best-radius configuration for our methods uses the largest smoothing noise in the grid. As expected, larger $\sigma$ improves certified radius but can degrade the smoothed predictor's accuracy. Figure 6 reports this tradeoff using the Monte Carlo estimate of the smoothed age predictor's MAE.

The experiment was run on NVIDIA A100 GPUs using batched model inference. We used mixed precision and tensor-native preprocessing for efficient Monte Carlo evaluation. The grid was sharded across test points and parameter settings on the cluster; the $(E, C) + M$ curve was then postprocessed from saved estimates without rerunning model inference.

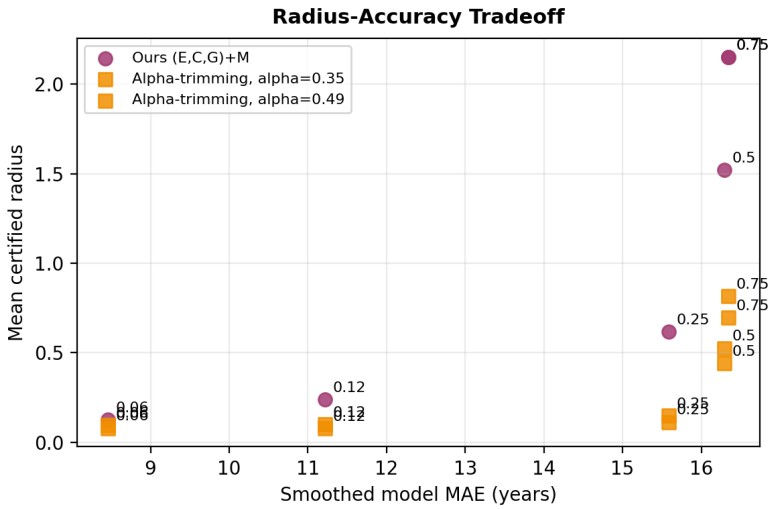

*Figure 6.* Age-estimation radius–accuracy tradeoff across smoothing levels. Larger $\sigma$ improves mean certified radius but worsens the Monte Carlo estimate of the smoothed regressor's MAE.

## J.5. Certified Accuracy Experimental Details

The metric definitions for absolute certified accuracy, conditional certified accuracy, and certified mean distance are provided in Section 7.4. Here we describe the experimental implementation details.

For rotation angle predictions, which are in the range $[-\pi, \pi]$ radians ($[-180°, 180°]$ degrees), we use circular distance to account for the periodic nature of angles. The circular distance between two angles $\hat{y}$ and $y$ (in degrees) is defined as $d(\hat{y}, y) = \min(|\hat{y} - y|, 360° - |\hat{y} - y|)$, which ensures that angles near the boundaries of $[-180°, 180°]$ are correctly compared. For example, if the true angle is $-175°$ and the prediction is $175°$, the circular distance is $10°$ (not $350°$), correctly reflecting that these angles are close on the circle.

### J.5.1. SIGMA SELECTION AND EVALUATION PROTOCOL

For each method and each radius threshold $R$, we use different sigma selection strategies depending on the metric. For certified accuracy, we select the $\sigma$ value that maximizes certified accuracy at that threshold. For mean distance, we select the $\sigma$ value that minimizes mean distance (i.e., gives the best prediction quality) for certified samples at that threshold.

We evaluate certified accuracy on 100 stratified test samples (10 per digit class) from the rotated MNIST test set. For each sample, we compute the certified radius $r_i$ using the certification method, compute the clean model prediction $\hat{y}_i$ (smoothed model output), extract the true label $y_i$ from the rotated MNIST dataset, and compute the prediction error $d(\hat{y}_i, y_i)$ using circular distance. We then check if the sample is both correct ($d(\hat{y}_i, y_i) \leq 10°$) and certified ($r_i \geq R$), and count the fraction of samples satisfying both conditions. This evaluation protocol uses the true label $y_i$ (from the rotated dataset) rather than the model output, ensuring that we measure accuracy relative to ground truth rather than self-consistency.

## J.6. Optimization-Based Radius Computation Methodology

The optimization-based radius provides an empirical estimate of the true worst-case certified radius by using white-box optimization to find adversarial perturbations. While this is not a provable guarantee, it provides a useful lower bound for evaluating certificate tightness.

### J.6.1. PROBLEM FORMULATION

For a smoothed function $g_\sigma(x) = \mathbb{E}_{\eta \sim \mathcal{N}(0, \sigma^2 I)}[f(x + \eta)]$, the optimization-based radius is the largest radius $R$ such that:

$$\max_{\|\delta\|_2 \leq R} |g_\sigma(x + \delta) - g_\sigma(x)| \leq \epsilon, \tag{42}$$

where $\epsilon$ is the output tolerance. This is computed using projected gradient descent (PGD) to find worst-case perturbations, combined with Monte Carlo estimation to evaluate the smoothed function.

### J.6.2. ALGORITHM OVERVIEW

The computation proceeds in three steps:

**Step 1: Baseline Estimation.** We compute the clean baseline $g_\sigma(x)$ using Monte Carlo estimation with $N_{\text{mc}} = 50,000$ samples:

$$g_\sigma(x) \approx \frac{1}{N_{\text{mc}}} \sum_{i=1}^{N_{\text{mc}}} f(x + \eta_i), \tag{43}$$

where $\eta_i \sim \mathcal{N}(0, \sigma^2 I)$ are noise samples. For angle predictions, which are in the range $[-\pi, \pi]$ radians ($[-180°, 180°]$ degrees), we use circular averaging to handle the periodic nature of angles. Specifically, we compute the mean of cosine and sine values separately: $\bar{\cos} = \frac{1}{N_{\text{mc}}} \sum_{i=1}^{N_{\text{mc}}} \cos(\theta_i)$ and $\bar{\sin} = \frac{1}{N_{\text{mc}}} \sum_{i=1}^{N_{\text{mc}}} \sin(\theta_i)$, where $\theta_i$ are the predicted angles in radians. We then use the two-argument arctangent function $\arctan2(\bar{\sin}, \bar{\cos})$ to recover the mean angle. The arctan2 function computes the angle in the correct quadrant by taking both sine and cosine as inputs, ensuring that angles near the boundaries of $[-\pi, \pi]$ are correctly averaged (e.g., averaging $-175°$ and $175°$ gives $180°$ rather than $0°$).

**Step 2: Worst-Case Perturbation Search.** For a candidate radius $R$, we use PGD to find the worst-case perturbation $\delta^*$:

$$\delta^{(t+1)} = \Pi_{\|\delta\|_2 \leq R} \left( \delta^{(t)} + \alpha \nabla_\delta |g_\sigma(x + \delta^{(t)}) - g_\sigma(x)| \right), \tag{44}$$

where $\Pi_{\|\delta\|_2 \leq R}$ projects onto the $\ell_2$ ball of radius $R$, and $\alpha$ is the step size. At each iteration, we estimate $g_\sigma(x + \delta^{(t)})$ using Monte Carlo with the same noise samples (Common Random Numbers, CRN) to reduce variance.

**Step 3: Binary Search.** We perform binary search on $R$ to find the largest radius where the worst-case output change $\leq \epsilon$. We initialize $R_{\text{low}} = 0$ and $R_{\text{high}} = R_{\text{max}}$ (typically $5.0$ pixels for MNIST experiments). While $R_{\text{high}} - R_{\text{low}} > \text{tol}$ (where

tol $= 10^{-3}$ pixels), we compute $R_{\text{mid}} = (R_{\text{low}} + R_{\text{high}})/2$, evaluate the worst-case change at $R_{\text{mid}}$ using PGD, and update the bounds: if the change $\leq \epsilon$, we set $R_{\text{low}} = R_{\text{mid}}$; otherwise, we set $R_{\text{high}} = R_{\text{mid}}$. The algorithm returns $R_{\text{low}}$ as the optimization-based radius. The binary search typically converges in 10-15 iterations.

### J.6.3. IMPLEMENTATION DETAILS

We use $N_{\text{mc}} = 50,000$ Monte Carlo samples, validated to provide stable estimates of the smoothed function. To reduce variance, we employ Common Random Numbers (CRN): the same noise bank is reused across all evaluations for a given sample.

For the PGD optimization, we use $n_{\text{restarts}} = 5$ restarts to avoid local minima, with $n_{\text{steps}} = 50$ steps per restart and a step size of $\alpha = 0.01$ (adaptive based on radius). The perturbation is projected onto the $\ell_2$ ball and clipped to the $[0, 1]$ pixel range to ensure valid image values.

The binary search is initialized with range $[0, R_{\text{max}}]$ where $R_{\text{max}} = 5.0$ pixels for MNIST experiments, using a tolerance of tol $= 10^{-3}$ pixels. The search typically converges in 10-15 iterations, well below the maximum of 20 iterations.

### J.6.4. LIMITATIONS AND ASSUMPTIONS

The optimization-based radius is an *approximation* with several limitations. First, since it relies on optimization, PGD may not find the global optimum, so the optimization-based radius may underestimate the true worst-case radius. Second, the smoothed function $g_\sigma(x)$ is estimated with finite Monte Carlo samples, introducing estimation error. Third, the method requires white-box access, meaning we can query the model many times (typically $O(N_{\text{mc}} \times n_{\text{restarts}} \times n_{\text{steps}})$ evaluations per sample) to compute gradients and evaluate the smoothed function at different perturbations. Note that this does not require access to the model's internal parameters or weights; we only need the ability to evaluate the model function $f(x)$ at arbitrary inputs, which is available during evaluation but not in deployment scenarios. Finally, the computational cost of $O(N_{\text{mc}} \times n_{\text{restarts}} \times n_{\text{steps}})$ model evaluations per sample makes it expensive for large-scale evaluation. Despite these limitations, the optimization-based radius provides a useful empirical benchmark for evaluating certificate tightness, as it represents the best achievable estimate using optimization-based methods.

### J.6.5. VALIDATION

To validate the optimization-based radius computation, we verify three properties. First, we check soundness: certified radii should be $\leq$ optimization-based radii (conservative), which is confirmed empirically. Second, we verify convergence: results are stable across different random seeds. Third, we assess sensitivity: results are robust to PGD hyperparameters (step size, restarts). Empirical validation shows that certified radii are consistently $\leq$ optimization-based radii, confirming that our certificates are sound (conservative) as expected.

### J.6.6. EXPERIMENTAL SETUP FOR TIGHTNESS ANALYSIS

We evaluate tightness on the MNIST rotation task using 100 stratified test samples from rotated MNIST (10 per digit class). These same 100 samples are used across all certification methods: variance/gradient estimation, certified radius computation (both $(E, C) + M$ and $(E, C, G) + M$ methods), and alpha-smoothing certification.

We evaluate at a single noise level $\sigma = 0.5$ due to the high computational cost of optimization-based radius computation, with output tolerance $\epsilon = 10°$ (0.1745 radians). The binary search uses a maximum search radius of $R_{\text{max}} = 5.0$ pixels in raw pixel space, with a tolerance of $10^{-3}$ pixels. Of the 100 samples, 18 (18.0%) hit the $R_{\text{max}}$ bound and are excluded from ratio analysis.

For the 82 samples that do not hit the bound, we compute the tightness ratio $r_i = R_{\text{opt},i}/R_{\text{cert},i}$ for each sample $i$. The mean ratio reported in Table 3 is the arithmetic mean of individual ratios: $\bar{r} = \frac{1}{82} \sum_{i=1}^{82} r_i$. Note that this differs from the ratio of means $(\bar{R}_{\text{opt}}/\bar{R}_{\text{cert}})$ due to the nonlinearity of division; we report the mean of ratios as it better represents the typical per-sample tightness. All summary statistics in Table 3 (mean certified, mean optimization-based, mean ratio, median ratio) are computed over the same 82 samples for consistency.

For the optimization-based radius computation, we use $N_{\text{mc}} = 50,000$ Monte Carlo samples for the baseline estimation and $N_{\text{attack}} = 2,000$ samples per PGD step for evaluating the smoothed function during optimization. We use 5 PGD restarts with 100 steps per restart to avoid local minima, resulting in approximately $50,000 + (5 \times 100 \times 2,000) = 1,050,000$

model evaluations per sample. All computations are performed on GPU.

### J.6.7. SOUNDNESS ANALYSIS: SAMPLES NEAR THE BOUNDARY

A ratio $r \geq 1.0$ indicates a sound certificate (certified radius does not exceed the empirically-found worst-case radius), while $r < 1.0$ suggests potential finite-sample estimation error.

We find 2 samples (2.4% of uncapped samples) with ratio $< 1.0$: Sample 77 with $r = 0.39$ and Sample 13 with $r = 0.65$. For these samples, the certified radius exceeds the optimization-based radius by factors of $2.6\times$ and $1.5\times$ respectively, indicating that finite-sample variance and gradient estimates led to overconfident certificates. This is consistent with the probabilistic nature of our confidence intervals: with $95\%$ confidence level, we expect approximately $5\%$ of samples to fall outside the bounds, and the observed $2.4\%$ rate is within this tolerance.

Additionally, 4 samples (4.9% of uncapped) have ratios in the range $[1.1, 1.3]$: Samples 1, 51, 55, and 61. These samples are technically sound (ratio $\geq 1.0$) but represent cases where the certificates are relatively tight, with certified radii within $14\% - 30\%$ of the optimization-based radii. This is visible in the tightness histogram (Figure 2) as samples clustered near the $r = 1.0$ threshold.

In total, 6 out of 82 uncapped samples (7.3%) have tightness ratios below 1.3, representing the "edge cases" where statistical estimation is most challenging. The remaining 76 samples (92.7%) have ratios $\geq 1.3$, with a mean ratio of $3.40\times$ and median of $3.46\times$, indicating substantial conservatism in the certified radii. Including the 18 capped samples (which are definitively sound with optimization-based radius $\geq 5.0$ pixels exceeding all certified radii), the overall empirical soundness rate is 98% (98 out of 100 samples).

