# OpenReview forum: "Higher-Order Certified Robustness for Regression"
_ICML.cc/2026/Conference — ICML 2026 regular_

### Official Review · Reviewer_qKsM · 2026-03-02

**Soundness:** 3
**Presentation:** 2
**Significance:** 3
**Originality:** 3
**Overall Recommendation:** 4
**Confidence:** 4

**Summary:**

The authors explore certification mechanisms against regression, building upon developments in $\alpha$-smoothing.

**Compliance With Llm Reviewing Policy:**

Affirmed.

**Final Justification:**

I'm confident at this stage in my review of a 4 - I found it to be a well written paper, with decent mathematical detail, and I appreciated the authors extension into a broader class of evaluation problems over the rebuttal period. But at the same time, I'm still left cautious about the scope of evaluations, and the unaddressed review concern that this is a certification mechanism that is intrinsically dependent upon an approximation - which is somewhat anathema to the nature of certifications, which require tight bounding of approximations in order to ensure that their guarantees are, in fact, guarantees.

In short, I could see this being published at ICML, and there is scope for impact here, however there are elements that give me doubt, and rob me of my ability to be an enthusiastic cheerleader for this paper.

**Key Questions For Authors:**

Could the authors discuss more about the implications of the parameter estimates on overall robustness? How well would this approach apply to move complex regression tasks?

**Limitations:**

See above: discussion of limitations are limited, and there is grounds for considering certifications beyond a strictly beneficial concept.

**Strengths And Weaknesses:**

On first pass, I could find nothing wrong with the work - the technical details make sense, the presentation of data was of high quality, and there were no notable writing issues. However, while usually I would only read a paper under review at most twice, I came back to this one again and again, because I couldn't put a finger on why I didn't like the authors work.

It took a few passes to realise that while the technical elements of the writing were strong, it was the overall construction that was leading me to this sense of unease. This is a work that stands atop quite narrow foundations - Rekavandi (2024, 2025) - yet does not properly articulate its point of difference to these past works. And I think this is a consequence of a broader issue in the work - there's so much content here, that the authors have essentially created a full length paper in the appendices, to which they've provided densely packed signposting within the main body.

This density would, it appears, also drive the limited set of evaluations, which, while highly detailed for what they are, are also incredibly limited. While obviously the space of regression problems is more limited, I would expect to see broader evaluation - spanning aperiodic functions, or multidimensional regression problems. Of greater concern is the fact that after reading this, I'm not left with any great knowledge of how this technique would work in such contexts.

Finally, I'm also not a massive fan of certification mechanisms that are intrinsically dependent upon estimates - in this case of the variance and the gradient nor. While I appreciate that the authors considered this on Page 8 and in the appendices, the treatment is relatively thin on the the ground, given the potential applications of this.

I also found the limitations to be a little lacking, and even the implications - while I appreciate that 99% of certification papers would present this the same way, there are technically some benefits to a lack of robustness (for example, adversarial privacy). Certifications are not strictly benign.

---

> ### Author Rebuttal · Authors · 2026-03-30
>
> We sincerely thank the reviewer for their dedicated time and multiple readings of our work. We are glad you found the technical details sound and the data presentation high-quality. Your close reading has accurately identified areas where the narrative structure and contextualization can be significantly improved.
>
> **Point of Difference and Broader Foundations:** "This is a work that stands atop quite narrow foundations - Rekavandi (2024, 2025) - yet does not properly articulate its point of difference... there's so much content here... This density would, it appears, also drive the limited set of evaluations..."
>
> **Response:** We appreciate the opportunity to clarify the broader foundations of our work and our exact point of difference from the baselines. To address the concern regarding narrow foundations, we highlight two key theoretical distinctions:
>
> **1. Fundamental Difference from Rekavandi et al. (2024, 2025):** The theoretical bounds in Rekavandi et al. are intrinsically probabilistic, yielding only a fractional probability of robustness even with infinite samples. In contrast, our theoretical bounds are strictly deterministic at the population level. Our practical certificates are probabilistic solely due to finite-sample estimation; if provided infinite samples to compute exact moments, our certified radius becomes an absolute, deterministic guarantee.
>
> **2. Broader Foundations in Robust Classification:** Our work does not solely stand atop recent regression baselines. We draw on a broader foundation from the robust classification literature—specifically works that leverage smoothed gradient information to yield tighter certificates (e.g., Levine et al., 2020). A core contribution of our paper is pioneering the mathematical translation of these gradient-based certification concepts into the continuous regression domain. We develop a novel variational framework in order to extend gradient-based certification to the regression setting.
>
> **Evaluation on Aperiodic and Multidimensional Tasks:**
>
> **Response:** We are not entirely clear on the specific limitations the reviewer is asking about regarding these contexts, and  we welcome further clarification if we have misunderstood. Regarding aperiodic functions, our theoretical framework and certified bounds are completely general and apply natively to any standard continuous regression task without modification. For multidimensional regression, our method naturally extends via the established coordinate-wise approach (e.g., Rekavandi et al., 2024), guaranteeing joint robustness by taking the minimum certified radius across all output dimensions.
>
> **Limitations and Dual-Use Nature:** "there are technically some benefits to a lack of robustness (for example, adversarial privacy). Certifications are not strictly benign."
>
> **Response:** We completely agree with this insightful point: adversarial machine learning has clear dual-use implications, and robustness certifications are not strictly benign (e.g., neutralizing adversarial privacy protections against surveillance). We will expand our Impact Statement to explicitly acknowledge these broader risks.
> However, we will also clarify the intended scope of our specific framework. Because our method is designed for continuous regression, our primary motivating applications are safety-critical physical systems—such as certifying steering angles in autonomous driving or real-valued robotic control parameters—rather than privacy-targeting applications. We will ensure this distinction between our intended physical-safety use cases and the broader dual-use risks is clearly articulated in the final version.

---

> > ### Author Rebuttal · Reviewer_qKsM · 2026-04-02
> >
> > Thank you for the response - I appreciate the clarification on this works foundations.
> >
> > To the point regarding aperiodic and multidimensional evaluation - I was specifically thinking about line 333 here, and how the task you were testing on a scenario that exhibited continuous periodic angles. To be more specific, I would prefer to see evaluation on tasks that did not have ideal properties - ie. that happens in an unbounded function, or where there was not periodicity in the values (ie. bounded within [-1,1], but where going past 1 does not shoot to -1).
> >
> > Similarly for multidimensional evaluation - if your argument is that you've made a step forward from Rekavandi, then stating that you are confident that your technique would work in these scenarios because it worked for them might slightly undercut the argument that you've made a significant step forward. I would argue that testing these scenarios would be highly beneficial.

---

> > > ### Author Response · Authors · 2026-04-06
> > >
> > > ### Advances from Rekvandi et. al.
> > > Both Rekavandi et al. and our paper develop smoothing-based robustness certificates for single-output continuous regression. Rekavandi et al. additionally propose a procedure for utilizing multiple single-output certificates to generate a certificate for the multi-output setting. Our present paper focuses exclusively on the first step—deriving stronger certificates for the single-output case—and does not claim to address multi-output regression directly.
> > >
> > > Within this shared scope, our method constitutes a substantial advance over Rekavandi et al. on two fronts:
> > > 1. **Deterministic guarantees:** Rekavandi et al. produce inherently probabilistic bounds—guaranteeing certification only with some probability strictly less than one, regardless of sample size. Our certificates are fully deterministic at the population level.
> > > 2. **Empirical improvement:** These stronger guarantees translate into concrete empirical gains, with our method achieving up to a 233% increase in certified radius on the Rotated MNIST benchmark.
> > >  Regarding multi-output regression: this is explicitly outside the scope of the present paper. Rekavandi et al.'s procedure for extending single-output certificates to the multi-output setting treats the single-output certificate as a modular subroutine, and our method is a drop-in replacement for that subroutine — so an improvement in the single-output setting is guaranteed to carry over in principle. However, the magnitude of the improvement observed in the single-output setting (233% on average on Rotated MNIST) is an empirical result, and we agree that the extent to which comparable gains materialize in the multi-output setting is an empirical open question. We view this as a natural direction for future work.
> > > ### Bounded vs. Unbounded outputs
> > > We would like to clarify that bounded outputs do not represent an "ideal" case for our method. Our framework addresses both bounded and unbounded outputs on their own terms: in the bounded case, we incorporate the output constraint into the certificate to obtain a tighter bound, exploiting additional problem structure rather than simplifying the problem.
> > > This stands in contrast to α-smoothing (Rekavandi et al.), where the bounded case genuinely is the easier one: their method requires trimming techniques specifically to handle the unbounded setting, meaning periodicity and boundedness do represent favorable properties for their approach. For our method, no such asymmetry exists.
> > > ### Additional experiments
> > >
> > > We are currently running experiments on an aperiodic regression task, and will get back to you once we have the results.
> > >
> > > **Update:** **We have now added new results on the aperiodic regression task;** **please see our reply to reviewer fZiM.**

---

### Official Review · Reviewer_RzPq · 2026-03-10

**Soundness:** 3
**Presentation:** 3
**Significance:** 2
**Originality:** 2
**Overall Recommendation:** 4
**Confidence:** 3

**Summary:**

The paper provides a new method for estimating the certified radius for a regression task. The method uses gradient information to imrpove the certified radius information.

**Compliance With Llm Reviewing Policy:**

Affirmed.

**Key Questions For Authors:**

1) The method assumes l2-norm perturbations, what happens if my perturbations are not in l2 space? A lot of robustness application prefer the l-inf norm, and the authors might be much more informed on this aspect than myself, so I kindly ask them for the reason to restrict the domain to l2-based perturbations.

2) I guess a more critical question follows from the weaknesses discussed earlier about regression. I do understand that a lot of the machine learning literature is dedicated to regression studies, and regression only. The problem is I am failing to realize the importance of focusing on regression only for this problem of certified robustness? What is important about regression that we are willing to discard the larger, widely used machine learning baselines?

**Limitations:**

1) The method is restricted to regression, a very nice special case in machine learning.
2) The scale of experiments is not large, with MNIST being the biggest baaseline.
3) The paper does not compare against well-defined methods in certified robustness.

**Strengths And Weaknesses:**

Strengths:

1) Sound theoretical analysis: the method rigorously derives the certified radius and worst-case perturbations.
2) Experimental improvement: the method shows improvements over common baselines that do not use gradient information.


Weaknesses:

1) The paper's main robustness domain is regression, but regression is the simplest form of machine learning, and its expressivity is limited. This means that the impact of the given paper is limited. Most of the machine learning algorithms employ different variants of the deep learning structures, which are much more complicated than regression. This also raises a question about the applicability of the method. Is there any variant for deep structures? For example, can we have (approximate guarantees) for deep structures? The obvious is that the method is generalizable, but I am definitely willing to change my view and score if enough evidence is provided.

2) The scale of the experiments is not big. More experiments are needed to verify the applicability of the methods to larger. MNIST, while providing an intuitive indication of the performance of the method, is not enough, and we would like to see results on larger benchmarks, going from CIFAR-10 to Image-Net. I do understand that such benchmarks may not work for the case of regression, which again reiterates the point about the applicability of regression.

---

> ### Author Rebuttal · Authors · 2026-03-30
>
> We thank the reviewer for finding our theoretical analysis rigorous and our experimental improvements clear. We respectfully address the specific concerns below, clarifying a fundamental misunderstanding regarding the scope of our work.
>
> **Applicability to Deep Structures:**
>
> Response: We respectfully clarify that there appears to be a conflation between the basic algorithm of "linear regression" and the broad machine learning task of "continuous regression" (predicting real-valued outputs). Our paper addresses the latter and is explicitly designed for deep neural networks. Continuous regression is ubiquitous in safety-critical domains (e.g., autonomous steering, robotic control) but remains severely under-explored in certified robustness compared to classification. Answering your question regarding deep structures directly: our framework is evaluated on them. Section 7.2 utilizes an E(2)-equivariant convolutional neural network (E2CNN) with roughly 50,000 parameters operating on high-dimensional images. Our higher-order framework provides deterministic robustness guarantees for these complex, deep regression models.
>
> **Scale of Experiments:**
>
> Response: As you correctly intuited, CIFAR-10 and ImageNet are classification benchmarks and lack continuous target variables. We chose the Rotated MNIST angle prediction task because it provides a high-dimensional input space (784 pixels) mapped to a continuous, periodic target using a deep E2CNN. This 784-dimensional space triggers the curse of dimensionality inherent to adversarial robustness, allowing us to isolate and demonstrate how gradient constraints can overcome the limitations of isotropic variance bounds in settings where previous methods become ineffective.
>
> **Key Question (1)** (Restriction to l2​-norm):
>
> Response: We completely agree that extending certificates to other perturbation spaces, particularly the l-inf norm, is a highly desirable goal. However, achieving this within a randomized smoothing framework presents fundamental mathematical hurdles.
>
> Regarding l-inf, there are important theoretical barriers already known in the randomized smoothing literature for classification. In particular, Blum et al. (2020) show that smoothing-based l-inf certification can become vacuous in high-dimensional settings. While that result is not stated specifically for our regression setting, it illustrates a broader difficulty: smoothing is much better suited to l2 certification than to l-inf, especially in high dimensions.
>
> Extending our specific closed-form certificates to l1​ or l-inf would require replacing Gaussian noise with Laplace or Uniform distributions. Doing so breaks the analytical tractability of the likelihood-ratio derivations and gradient constraints required to find our worst-case functions. Adapting our approach  to other noise distributions to tackle these l1 and l-inf barriers is a highly promising, albeit non-trivial, avenue for future work.
>
> **Limitations** (Comparison against Baselines):
>
> Response: We respectfully point to our extensive evaluation against $\alpha$-smoothing (Rekavandi et al., 2024; 2025). While the field of classification has dozens of established baselines, the literature for direct, continuous regression certification is highly limited. $\alpha$-smoothing represents the current state-of-the-art for certified regression. Our experiments specifically demonstrate that our higher-order method achieves up to 1.76x the mean certified radius of this SOTA baseline while certifying a vast number of samples where the baseline completely fails.

---

> > ### Author Rebuttal · Reviewer_RzPq · 2026-03-31
> >
> > I thank the authors for the detailed rebuttal. I appreciate their careful and thorough examination of my concerns, although some might have been misguided. I will increase my score.

---

> > > ### Author Response · Authors · 2026-04-02
> > >
> > > Thank you for taking the time to revisit the paper and for updating your assessment—we appreciate it.
> > >
> > > To clarify the overall contribution, our method establishes state-of-the-art certified robustness for regression models, addressing a relatively underexplored setting. In particular:
> > >
> > > - **Deterministic guarantees:** Prior work (e.g., Rekavandi et al.) provides intrinsically probabilistic bounds, yielding only a fractional probability of robustness even with infinite samples. In contrast, our bounds are fully deterministic at the population level.
> > > - **Empirical improvement:** Our experiments show a substantial increase in certified radius (up to 233% on Rotated MNIST).
> > >
> > > We would be grateful if you could let us know if any remaining concerns prevent a stronger recommendation. We’re happy to clarify further.

---

### Official Review · Reviewer_Tk8Q · 2026-03-12

**Soundness:** 3
**Presentation:** 3
**Significance:** 2
**Originality:** 2
**Overall Recommendation:** 4
**Confidence:** 3

**Summary:**

This paper is broadly focused on the $l_2$-additive adversarial attack setting. More specifically, the authors study a verification problem where the goal is to certify the adversarial robustness of a regression function under a given attack radius. Since the regression function can be difficult to work with directly (*e.g.*, a deep model), the resulting optimization problem is handled in a conservative but efficient manner: the authors seek the worst-case attack against a worst-case 'candidate' function, where this candidate function is allowed to vary within a class of functions that remain close to the true function. In this work, such closeness is characterized through summary statistics, namely the mean and variance of the function, as well as the behavior of its gradient around a given input. To further maintain tractability, the authors introduce an additional layer of Gaussian smoothing and account for the extra loss induced by this smoothing (unlike the similar literature). While analogous problems are well established in classification through a binary hypothesis testing perspective, the literature is much more limited for regression as critically summarized by the authors. To address this gap, the authors propose a variational calculus approach, rather than a binary hypothesis testing one. With this approach, they are able solve the underlying certification problem with certain guarantees. The paper presents strong theoretical results, including exact reformulations within a family of summary statistics, and supports these with extensive numerical experiments on tasks such as rotated MNIST.

**Compliance With Llm Reviewing Policy:**

Affirmed.

**Final Justification:**

I would like to retain my weak accept score. Overall, I believe the paper is written well and studies an interesting subject. I have some doubts about the applicability of the results given the ad-hoc parameter choices and the discussion on duality over measures.

**Key Questions For Authors:**

Please see some questions in my main review above.

**Limitations:**

Yes, the authors adequately discuss their limitations.

**Strengths And Weaknesses:**

**Strength**
I enjoyed reading this paper. It is written clearly, the mathematical claims are rigorous, the proofs are correct to the best of my understanding. I believe when it comes to the question "if we want to solve problem (7), how can we do that?" the answer is quite sufficiently above the ICML acceptance bar.

**Weaknes**
I am not yet convinced about "why do we want to solve problem (7)". I hope that with an effective rebuttal period the authors can motivate this much better than the current version. More details are listed as follows:

1. First, the motivation for the smoothing step is not sufficiently justified. The authors mention at the beginning of Section 2 that smoothing is common in classification, and then Section 3 immediately proceeds with a smoothed formulation. However, I would like to better understand why Problem (3) should be modeled through Gaussian smoothing in the regression setting. Since the regression output is continuous, smoothing may not play the same role as it does in classification. I can see that smoothing likely helps make the max_{f \in F} component tractable, and it may well be a safe modeling choice in this work, but the current version does not adequately motivate it.
2. There is a cycle in the flow of logic, which needs to be eliminated: (i) In the beginning, the paper motivates that for a trained model that we hold, the main theorem will be on the worst-case function in an ambiguity class that is around the true function. (ii) Then, we say that “suppose we know the mean/variance/gradient information of the true function” and build the ambiguity set. (iii) We solve the problem over this ambiguity set. (iv-->i) Then we claim that the solution will be a verification for any distribution that generates these mean/variance/gradient ambiguity sets. In other words, the optimization problem is solved over a class of functions rather than over the exact trained function itself hence the paper presents the result as a robustness guarantee for the original function, but at that point the guarantee is really for any function consistent with the chosen summary statistics, not directly for the original trained model. This may be problematic for cases where the summary statistics can not be accurately calculated where we end up providing results for an arbitrary family of functions. I think this distinction should be stated much more clearly.
3. The main optimization problem that we are interested in solving resembles the dual of uncertainty quantification problems in distributionally robust optimization (*i.e.*, via Riesz representation). I am curious if this problem is not “accidentally” equivalent to DR-uncertainty quantification. This especially sounds risky now since we know that adversarial robustness can be certified with distributionally robust optimization. I would like to ensure that this problem is significantly different even after dual/distributional reformulations.
4. There are several layers of sufficient conditions to guarantee that certified robustness is almost surely true. I have already discussed above that we take smoothing steps and also look at a class corresponding to the "ambiguity" of functions. Even under such conservative certificates, the paper works with U-statistics and confidence intervals (that said, are we losing the almost surely certificate there?). I would like to therefore understand how much we lose from the true optimality. To this end, can the authors comment on the following scenario: Train a neural network with a specific $l_2$-adversarial robustness guarantee epsilon. Take the trained function and forget about your true epsilon. Now apply the technique in this paper to certify some value of epsilon around a local point (we can choose this local point via worst-case as we can use our past training). How much does this cerified value differs from the true epsilon?


Finally, some minor comments:
- Introduction “these approaches effectively reduces the complex behavior of the function to” —> not clear which function yet (loss, trained function, the function you use to certify, …)
- Introduction “This distinction allows for significantly tighter radii” —> no radius is mentioned yet
- Can the authors first show (3) where F is a singleton (the true function) and motivate what happens to this problem, why and when it is difficult, etc.?
- “with our observed statistics” —> observed from what?
- Experiments: why sigma grid is 0.1, 0.2, 0.5? Also why alpha is 0.35, 0.49? These look ad hoc to me and I would like to understand this further.
- Experiments: why 10 randomly sampled points?
- Appendix page 22: Proof of (a) and Proof of (b) are probably latex typos as I was not able to find (a) and (b).
- Optionally: There is a lot of repetition of Lagrange dual steps in the proofs. Can these be shortened?

---

> ### Author Rebuttal · Authors · 2026-03-30
>
> We sincerely thank Reviewer Tk8Q for the thoughtful review and the positive assessment of our theoretical rigor and mathematical claims. We appreciate your close reading and the sophisticated technical questions you raised. We address your specific questions below.
>
> **Motivation for Gaussian smoothing in regression:**
>
> Response: We thank the reviewer for this helpful point. We agree that, unlike in classification, smoothing in regression is not introduced to turn a discrete decision rule into a continuous one. Rather, it defines the smoothed predictor $g(x)=\mathbb{E}[f(x+e)]$, which is the deployed object we certify. Although the base regressor is real-valued, continuity alone does not make adversarial certification tractable: deep regressors may still be highly non-convex and locally sensitive, making direct worst-case verification of the raw predictor difficult. Gaussian smoothing instead yields a locally averaged predictor whose adversarial shift can be bounded using moment and gradient information. This is why our certificate is centered on $g(z)$, and why smoothing is part of the robustness model rather than merely a technical device for solving the optimization.
>
> **The Logical Cycle / Ambiguity Class:**
> Response: We appreciate the reviewer catching this presentational cycle, which simply stems from inadvertently overloading the notation $f$. In our current draft, we overloaded the notation $f$ to refer to both the specific trained neural network and the arbitrary candidate functions within the ambiguity class. To resolve this, the revised text will explicitly distinguish the specific trained model $f_{\theta}$ from the ambiguity class $\mathcal{F}$ defined by the observed summary statistics. We will briefly clarify that because $f_{\theta} \in \mathcal{F}$, computing the worst-case bound over the entire class $\mathcal{F}$ mathematically guarantees a safe lower-bound certificate for the original model $f_{\theta}$.
>
> **Connection to Distributionally Robust Optimization (DRO):**
>
> Response: We thank the reviewer for this insightful observation. We agree there is a meaningful structural connection to DRO/UQ, since both frameworks use Lagrangian ideas to study worst-case objects. However, they are not equivalent, and our formulation is not a DRO dual in disguise.
>
> The key distinction is that our method remains a function-space variational problem. In the bounded $(E, C, G)+M$ setting, we do not collapse the original problem directly into a finite-dimensional dual. Instead, we first reduce the high-dimensional search over $f: \mathbb{R}^d \rightarrow \mathbb{R}$ to a variational problem over a univariate function $\phi$, and only then solve numerically for the remaining multipliers $(\lambda, \mu, \nu)$ after deriving the optimal clipped likelihood-ratio form. Thus, the dual step is a consequence of the variational analysis, not a replacement for it.
>
> At the primal level, the optimization variables are also different. In standard DRO/UQ, one optimizes over a distribution $Q^{\prime}$ (e.g., in a Wasserstein-$\infty$ ambiguity set). In our formulation, one optimizes over a function $f \in \mathcal{F}$, while all mean/variance/gradient constraints are evaluated under the fixed nominal smoothing distribution $p_z$. Thus the ambiguity in our problem is over functions consistent with local statistics under $p_z$, not over distributions in a transport ball.
>
>
> **The Optimality Gap and Probabilistic Certificates:**
> Response: Trading an absolute guarantee for a high-probability (1−α) bound is unavoidable in sampling-based smoothing (Cohen et al., 2019). About optimality gap: Because calculating the exact "true ϵ" is intractable, we quantified this gap empirically via PGD attacks (Section 7.4), finding a median ratio of 3.48× between the true empirical ϵ and our certified ϵ. We emphasize that this factor is specific to the evaluated architecture and dataset, not a universal constant. We note that our empirical 3.48× gap is highly competitive, especially since simpler binary classification baselines (Cohen et al., 2019) found gaps >2× for nearly half of their samples.
>
> **Minor comments**
> We will clarify wording/notation, explain the singleton $\mathcal{F}$ motivation for (3), specify that observed statistics are Monte Carlo estimates under Gaussian noise, justify the α/σ grids and 10-point setup, and fix Appendix typos.

---

> > ### Author Rebuttal · Reviewer_Tk8Q · 2026-04-03
> >
> > I thank the authors for their response. I have read the rebuttal and I would like to keep my slightly positive score.
> >
> > My reasons for not increasing the score further are:
> >
> > 1. the rebuttal claims several of my points will be clarified later in the paper. I still do not understand why alpha grid is 0.35, 0.49, for example.
> > 2. I am not convinced about the DRO non-equivalence. The fact that there is no finite-dimensional dual should not imply that the problem does not admit a DRO-like dual. There are a lot of infinite-dimensional optimization problems in DRO.
> > 3. I still think the paper is not fully clear about whether the final certificate is for the specific trained regressor or for a larger class of functions consistent with estimated local statistics, and how conservative this relaxation is in practice. The rebuttal does not really answer my previous question on this point.

---

> > > ### Author Response · Authors · 2026-04-05
> > >
> > > # 3
> > > We apologize that our previous response was not sufficiently clear. Following the paradigm established in the seminal work of [5], the certified object is the *smoothed* model
> > >
> > > $$g(x) = \mathbb{E}_{e \sim \mathcal{N}(0, \sigma^2 I)}[f^*(x + e)],$$
> > >
> > > not the base regressor $f^*$: the smoothed model is both the deployed and the certified object.
> > >
> > > Further, the certificate holds for any smoothed model whose base satisfies the required summary statistics. Since $f^*$ satisfies these statistics by construction, $g$ is covered by the guarantee. The relaxation to the ambiguity class $\mathcal F$ does introduce conservatism: the certified radius may be smaller than $g$’s true robust radius. This affects tightness only, not validity. The certificate is a lower bound on $g$'s certified radius, in direct analogy with [5]. We quantify this conservatism empirically: on Rotated MNIST, our certified radius is conservative by a factor of 3.41 on average, and across three synthetic functions (Table 1) by a factor of 1.07–3.65.
> > > # 1
> > > We apologize for the insufficient detail in our initial response; due to character limits, we were necessarily brief. A detailed point-by-point reply follows.
> > > - Intro phrasing: We will clarify "reduces complex behavior" refers to the base function $f(x)$, and change "tighter radii" to "tighter robustness bounds".
> > > - Where $\mathcal{F}$ is a singleton: If one takes $\mathcal{F}=\\{f^\ast\\}$, Eq. (3) reduces to the exact certification problem for the trained model's smoothed predictor. This exact problem is difficult because $g(x)=\mathbb{E}_{e} [f^*(x+e)]$ generally has no closed form for a deep regressor, and the remaining maximization over perturbations is still a hard nonconvex verification problem.
> > >
> > > - "Observed from what?": Monte Carlo samples of the trained base model under Gaussian noise.
> > > - Grid choices ( $\sigma$ and $\alpha$ ): The $\alpha$ grid $\\{0.35,0.49\\}$ evaluates the baseline at both a stable trimming level and its strict theoretical limit ( $\alpha<0.5$ ). The synthetic $\sigma$ grid $\\{0.1,0.2,0.5\\}$ mirrors standard progressions (e.g., [5] evaluated ${0.12,0.25,0.50}$ ). Our primary MNIST evaluation utilizes a 5-point sweep ( $\sigma \in \\{0.06,0.12,0.25,0.50,0.75\\})$.
> > > - 10 random points: Section 7.1 is a proof-of-concept. Ground truth comparison (Appendix J.1) required hundreds of millions of evaluations per test point. 10 points conclusively established the trend. Foundational baselines ( $\alpha$-smoothing, RS-Reg) similarly evaluated only 25 points.
> > > - Appendix typos ((a) and (b)): Thank you for catching this. We will fix this.
> > > # 2
> > > To our knowledge, prior work that adopts a DRO view [1,2,3,4] does not reduce to our approach. Finding the error under a worst-case perturbation $\mathbb{E}\_Q \sup\_{\|x-x'\|\leq r} f(x')$ is equivalent to computing $\sup\_{Q'\in \mathcal{B}\_r(Q)} \mathbb{E}\_{Q'} f(x)$, where $\mathcal{B}\_r(Q)=\\{ Q': W_\infty(Q,Q')\leq r\\}$. [3] and [4] adopt this view to derive a dual problem. However, this does not reduce to our approach because:
> > >
> > > 1. **Bound direction:** Certification requires an upper bound on adversarial risk. The dual of the minimization problems in prior work [1,2,3,4] is a maximization problem, yielding a lower bound — the wrong direction for certification.
> > > 2. **Per-point vs. dataset-level:** This dual is over the entire dataset rather than per-point.
> > > 3. **Dual complexity:** The dual problem lives on the space of measures and is difficult to work with directly as it requires computation of densities with respect to dual variables.
> > >
> > > To address this final point, [1,2] compute a different dual via Strassen's theorem, reducing to a finite-dimensional empirical problem — but this still requires nontrivial combinatorial optimization, and inherits the lower bound issue from point (1).
> > >
> > > Our formulation also differs structurally from the classical UQ primal:
> > > $$\min\_{Q'} \mathbb{E}\_{Q'} f(x) \quad \text{s.t.} \quad \mathbb{E}\_{Q'} \phi\_1(x)=\mu\_1, \quad \mathbb{E}\_{Q'} \phi\_2(x)=\mu\_2, \ldots$$
> > > In the standard UQ primal, the moment constraints are evaluated under the optimization variable $Q'$ itself. In our formulation, the constraints are evaluated under the fixed smoothing distribution which is independent of the dual variable. This structural difference means our problem does not admit a standard UQ dual interpretation.
> > >
> > > In summary, our formulation is best understood as a function-space variational problem: we optimize over functions $f$ consistent with statistics computed under the fixed smoothing distribution $Q$.
> > >
> > >
> > > **[1]** Lower Bounds on Adversarial Robustness from Optimal Transport.
> > > **[2]** Characterizing the Optimal 0-1 Loss for Multi-class Classification with a Test-time Attacker.
> > > **[3]** The Many Faces of Adversarial Risk.
> > > **[4]** Frank and Niles-Weed. Existence and Minimax Theorems for Adversarial Surrogate Risks in Binary Classification.
> > > **[5]** Certified adversarial robustness via randomized smoothing.

---

### Official Review · Reviewer_fZiM · 2026-03-13

**Soundness:** 2
**Presentation:** 3
**Significance:** 3
**Originality:** 3
**Overall Recommendation:** 3
**Confidence:** 3

**Summary:**

This paper extends the traditional randomized smoothing method from classification to regression, and incorporates additional information such as variance and gradients to build a randomized smoothing framework for regression, which is further validated through experiments.

**Compliance With Llm Reviewing Policy:**

Affirmed.

**Final Justification:**

This paper is passable overall. It is not without merit, and I do think its exploration of the problem is of some value. However, the certification it provides is not particularly sound. I therefore remain at a weak reject (3), or at most would consider it borderline (3.5).

**Key Questions For Authors:**

See S&W

**Limitations:**

yes

**Strengths And Weaknesses:**

Strengths:

(1) The theoretical derivations are quite strong. The paper models the problem in multiple ways based on variance, gradients, and related information, and derives several interesting conclusions, most of which are highly consistent with intuition.

(2) The amount of work is substantial, and the authors demonstrate a deep understanding of the research problem.

Weaknesses:

(1) The main text does not contain a conclusion section, which makes the overall structure of the paper incomplete.

(2) The experimental evaluation is insufficient. MNIST is too small a dataset and lacks representativeness.

(3) As the authors mention, the estimation of variance and gradients relies on U-statistics, which reduces the overall contribution of the work. Moreover, randomized smoothing is a certified robustness method, and if it cannot provide a reliable certificate, then even a numerically very close approximation is difficult to accept in this area. Perhaps the problem itself is intractable, or perhaps there are other clever ideas yet to be discovered. Although the gap here is only slight, it falls exactly at the critical point of certified robustness.

---

> ### Author Rebuttal · Authors · 2026-03-30
>
> We sincerely thank Reviewer fZiM for their thoughtful review and for recognizing the strength of our theoretical derivations and the substantial effort in modeling this problem. We address specific weaknesses below.
>
> >Weakness (1): The main text does not contain a conclusion section, which makes the overall structure of the paper incomplete.
>
> Response: We agree and apologize for the omission. For the camera-ready version, we will add a formal conclusion section.
>
> >Weakness (2): The experimental evaluation is insufficient. MNIST is too small a dataset and lacks representativeness.
>
> Response: We respectfully clarify that our evaluation setting is not standard 10-class MNIST digit classification, but the rotated MNIST angle prediction task. This continuous regression task uses a deep E2CNN over a high-dimensional input space (784 pixels), which is more than sufficient to trigger the curse of dimensionality in adversarial robustness. It provides a clean setting for isolating the geometric effect studied in our theory, namely how gradient constraints can overcome the limitations of isotropic variance-based certificates (e.g., certifying all samples at our optimal noise level of $\sigma = 0.75$, where probability-mass methods fail).
>
> Furthermore, establishing the exact empirical tightness of our bounds (Section 7.4) via rigorous PGD optimization required over one million function evaluations per test point (50k samples for the initial smoothed prediction+ 5 restarts × 100 steps × 2k samples/step). Scaling this massive empirical verification pipeline to ImageNet-scale regression is computationally prohibitive and unnecessary to validate the mathematical soundness of our theorems.
>
> >Weakness (3): As the authors mention, the estimation of variance and gradients relies on U-statistics, which reduces the overall contribution of the work. Moreover, randomized smoothing is a certified robustness method, and if it cannot provide a reliable certificate, then even a numerically very close approximation is difficult to accept in this area.
>
> Response: We agree that reliability is the most critical property of a certified defense. Trading a deterministic guarantee for a high-probability $(1−\alpha)$ certificate is an unavoidable necessity in all sampling-based randomized smoothing (e.g., Cohen et al., 2019), as exact verification of deep networks is computationally intractable.
> To ensure our probabilistic certificates remain rigorously reliable, we do not simply plug in raw point estimates of the U-statistics. As detailed in Section 6, we construct confidence intervals around our estimates and explicitly optimize our variational certificate over these intervals to find the most conservative certified radius. This mathematically guarantees that statistical estimation error strictly shrinks, rather than inflates, our robustness bounds. Section 7.4 empirically validates this deliberate conservatism: our optimization yields a median tightness ratio of 3.48x, keeping certificates safely below true empirical failure points. Crucially, our actual empirical violation rate (2%) remains strictly under our theoretical failure allowance (5%), confirming these bounds are robust and safe in practice.

---

> > ### Author Rebuttal · Reviewer_fZiM · 2026-04-03
> >
> > I would like to thank the authors for their response. However, I believe my concerns have not been fully addressed. The experimental results in the current manuscript are still not sufficiently extensive, and I remain skeptical about the reliability of the proposed certification.

---

> > > ### Author Response · Authors · 2026-04-06
> > >
> > > We thank the reviewer for their continued engagement. We would like to clarify that probabilistic certificates are the accepted standard across the scalable certified robustness literature — deterministic certificates for deep networks are computationally intractable, as exact verification is NP-hard [Katz et al., 2017]. All sampling-based randomized smoothing methods, including the foundational Cohen et al. (ICML 2019) and Lécuyer et al. (IEEE S&P 2019), provide certificates that hold with probability $1-\alpha$ rather than deterministically. Our work operates within this same paradigm and should be evaluated against the same standard.
> > >
> > > Within this paradigm, we take particular care to ensure our probabilistic certificates are handled rigorously. We do not plug in raw point estimates of the U-statistics; instead, as detailed in Section 6, we construct confidence intervals around our estimates and optimize our variational certificate over these intervals to find the most conservative certified radius. This guarantees that statistical estimation error strictly shrinks rather than inflates our robustness bounds. Section 7.4 validates this empirically: our actual violation rate (2%) remains strictly below our theoretical failure allowance (5%), confirming that our certificates are reliable and conservative in practice.
> > >
> > > ### Additional experiments
> > >
> > > UTKFace is a large-scale facial age regression dataset with labels in $[0,116]$, representing a scalar, aperiodic regression task on high-dimensional image inputs. Unlike periodic quantities such as angles, age is a non-periodic continuous value with a natural but non-wrapping upper bound. We evaluate our method on this dataset using a pretrained MiVOLO model in inference-only mode. We compare our bounded certifier $(E,C,G)+M$ against $\alpha$-smoothing, using confidence level $0.9$, output tolerance $\varepsilon_y=6$, and Monte Carlo sample size $N=10{,}000$ (and $n_{\mathrm{tr}}=10{,}000$ for $\alpha$-smoothing), motivated by a convergence check indicating $5\mathrm{k}$--$10\mathrm{k}$ is stable and using $10\mathrm{k}$ in final runs. We sweep $ \sigma \in$ $\\{0.06,0.12,0.25,0.5,0.75 \\} $  and $\alpha \in $$\\{0.35,0.49 \\}$ for $\alpha$-smoothing on an initial 20-point run, obtaining best fixed settings $(E,C,G)+M:\ \sigma=0.75$ and $\alpha$-smoothing: $(\sigma,\alpha)=(0.75,0.49)$. We then evaluated the same fixed settings on an additional disjoint 30-point set. While this is not a strict holdout-only evaluation protocol, we report the aggregate over all 50 points to provide the most complete evidence available within the rebuttal timeline.
> > > At these best fixed settings, mean certified radius is $2.207$ vs $0.920$, and median certified radius is $2.089$ vs $0.000$, for $(E,C,G)+M$ and $\alpha$-smoothing, respectively. As shown in Table 1, although $\alpha$-smoothing is larger on some points (14/50), the delta mass is strongly asymmetric in our favor: $\sum_{\Delta_i>0}\Delta_i=68.121$ and $\sum_{\Delta_i<0}\lvert\Delta_i\rvert=3.775$, where $\Delta_i=R_i^{\text{ours}}-R_i^{\alpha\text{-smooth}}$. This gives a gain/loss ratio of $68.121/3.775=18.04\times$. Figure 1 visualizes the corresponding CDF comparison at each method’s best fixed parameter setting.
> > >
> > >
> > > Figure 1 (anonymous link): [CDF of Certified Radii (Best Parameter per Method)](https://ibb.co/RpV64Ykn).
> > >
> > >
> > > **Table 1: UTKFace additional experiment (50 points), best fixed setting per method**
> > > | Setting | Value |
> > > |---|---:|
> > > | Total points | 50 |
> > > | Ours best fixed params | $\sigma=0.75$ |
> > > | $\alpha$-smoothing best fixed params | $\sigma=0.75,\ \alpha=0.49$ |
> > > | Ours mean radius | 2.207 |
> > > | $\alpha$-smoothing mean radius | 0.920 |
> > > | Ours median radius | 2.089 |
> > > | $\alpha$-smoothing median radius | 0.000 |
> > > | Mean delta (ours - alpha) | 1.287 |
> > > | Median delta (ours - alpha) | 2.065 |
> > > | Pointwise wins (ours / alpha / tie) | 36 / 14 / 0 |
> > > | $\sum_{\Delta_i>0}\Delta_i$ | 68.121 |
> > > | $\sum_{\Delta_i<0}\lvert\Delta_i\rvert$ | 3.775 |
> > > | Net sum delta | 64.345 |
> > > | Gain/loss ratio $\left(\frac{\sum_{\Delta_i>0}\Delta_i}{\sum_{\Delta_i<0}\lvert\Delta_i\rvert}\right)$ | $18.04\times$ |
> > >
> > >
> > >
> > >
> > > **[Katz et al., 2017]** Katz, G., Barrett, C., Dill, D., Julian, K., & Kochenderfer, M. Reluplex: An Efficient SMT Solver for Verifying Deep Neural Networks. *CAV 2017*.

---

### Decision · Program_Chairs · 2026-04-30

**Decision:**

Accept (regular)

**Comment:**

This paper studies the problem of certified robustness against L2 perturbations for the case of regression employing randomized smoothing. The authors propose an approach that leverages estimators (means, variance, gradients) to tighten the resulting robustness certificates. The authors provided experiments on estimating rotation angles on mnist (and later also added an experimental UTKFace age-regression).

There is general agreement that this paper is a valuable technical contribution, with sound analysis and strong theoretical results, though with mixed enthusiasm. Reviewers did voice concerns on the limitation of the empirical validation, which was expanded during the rebuttal period. An important point of contention, that limited the excitement about this paper from the reviewers, is the fact that the develop certificates relies on the quality of estimation of the different quantities, and does not -therefore- provide a deterministic certificate. This is natural however, and the discussion of the authors in the paper, in addition to the empirical demonstration of the validity of their finite-size bounds, provides sufficient compelling evidence to this AC. Reviewers had only moderate confidence.

In all, this is a valuable contribution with a clean and rigorous presentation, and I recommend acceptance.